# Training Uncertainty-Aware Classifiers with Conformalized Deep Learning

**Bat-Sheva Einbinder**[*]
Faculty of Electrical & Computer Engineering
(ECE) Technion, Israel
bat-shevab@campus.technion.ac.il

**Yaniv Romano**
Faculty of ECE and of Computer Science
Technion, Israel
yromano@technion.ac.il

**Matteo Sesia**
Department of Data Sciences and Operations
University of Southern California
Los Angeles, California, USA
sesia@marshall.usc.edu

**Yanfei Zhou**
Department of Data Sciences and Operations
University of Southern California
Los Angeles, California, USA
yanfei.zhou@marshall.usc.edu

## Abstract

Deep neural networks are powerful tools to detect hidden patterns in data and leverage them to make predictions, but they are not designed to understand uncertainty and estimate reliable probabilities. In particular, they tend to be overconfident. We begin to address this problem in the context of multi-class classification by developing a novel training algorithm producing models with more dependable uncertainty estimates, without sacrificing predictive power. The idea is to mitigate overconfidence by minimizing a loss function, inspired by advances in conformal inference, that quantifies model uncertainty by carefully leveraging hold-out data. Experiments with synthetic and real data demonstrate this method can lead to smaller conformal prediction sets with higher conditional coverage, after exact calibration with hold-out data, compared to state-of-the-art alternatives.

## 1 Introduction

The predictions of deep neural networks and other complex machine learning (ML) models affect important decisions in many applications [1–6], including autonomous driving, medical diagnostics, or security monitoring. Prediction errors in those contexts can be costly or even dangerous, which makes dependable and explainable uncertainty estimation essential. Unfortunately, deep neural networks are not designed to understand uncertainty and they are easily prone to overfitting; consequently, they may lead to overconfidence [7–9]. Overconfidence is especially problematic in the face of *aleatory* uncertainty [10], which refers to situations in which the outcome to be predicted is intrinsically noisy. Unlike the complementary *epistemic* uncertainty, aleatory uncertainty cannot be eliminated by training a more flexible model or by increasing the sample size. For example, think of prognosticating COVID-19 survival [11, 12], assessing genetic disease predisposition [13–15], or anticipating credit card defaults [16]. In those applications, the outcome of interest is potentially complicated and likely depends on many unmeasured variables. Therefore, no practical model may achieve perfect accuracy, but models that can offer practitioners principled measures of confidence for each individual-level prediction naturally tend to be more useful and trustworthy. Among many existing techniques for estimating uncertainty in ML predictions [7–9], conformal inference [17], stands out for its ability to provide finite-sample guarantees without unrealistic algorithmic simplifications and without strong assumptions about the data generating process or the underlying sources of uncertainty.

---

[*]Authors listed in alphabetical order.

36th Conference on Neural Information Processing Systems (NeurIPS 2022).

Conformal inference is designed to convert the output of any ML model into a *prediction set* of likely outcomes whose size is automatically tuned using *hold-out data*, in such a way that the same procedure applied to future test data will yield prediction sets that are well calibrated in a frequentist sense. In particular, these sets have provable *marginal coverage*; i.e., at the 90% level this means the outcome for a new random test point is contained in the output set 90% of the time. The hold-out data are utilized to evaluate *conformity scores*, or goodness-of-fit scores, whose ordering determines how the sets are to be expanded so that the desired fraction of test points is guaranteed to be covered.

A limitation of conformal inference is that it involves of two distinct phases, training and calibration, that are generally not designed to work together efficiently. The calibration algorithm takes as input pre-trained models that may already be overconfident, and this is sub-optimal because bad habits are harder to correct after they become entrenched. As a result of this uncoordinated two-step approach, conformal predictions may be either unnecessarily conservative or overconfident for certain types of test cases, which can make them unreliable [18, 19] and unfair [20]. We address this challenge by developing a new ML training method that synergetically combines the learning and calibration phases of conformal inference, reducing the overconfidence of the predictions calculated by the output model. The idea is to minimize a new loss function designed to measure the discrepancy in distribution between the conformity scores computed by the current model estimate and those of an imaginary oracle that can leverage perfect knowledge of the data generating process to construct the most informative and reliable possible predictions. As we will demonstrate empirically, applying standard conformal inference (based on independent calibration data) to models trained with our novel algorithm leads to smaller prediction sets with more accurate coverage for all test points. In other words, the "uncertainty-aware" ML models trained with the proposed algorithm tend to work more efficiently within a standard conformal inference framework compared to black-box models, which perhaps should not be surprising. Further, we will also show that the models obtained with our method produce relatively reliable prediction sets even if the post-training conformal calibration step is omitted, although the latter remains necessary in theory to guarantee valid coverage.

**Related work**

Many methods have been developed to mitigate overconfidence in ML [21–34], for example by allowing an *agnostic* output, by suitable post-processing [7, 35–40], or through early stopping [41]. Additional relevant research includes that of [42–46, 46–52], and these will provide us with informative benchmarks. However, unlike conformal inference, these methods have no frequentist guarantees in finite samples, and largely rely on loss functions targeting the accuracy of best-guess predictions, without explicitly addressing uncertainty during training. We build upon conformal inference [53–58], pioneered by [17], which typically deals with off-the-shelf models [17, 19, 59, 60]. Although other very recent works have proposed leveraging ideas from this field to improve training [61–66], this paper is novel as it combines the adaptive conformity scores of [19] with a completely new uncertainty-aware loss function. This departs from [62, 63], which sought to minimize the cardinality of the prediction sets, and from [61, 64–66], which utilized conformal inference ideas for tuning low-dimensional hyper-parameters as opposed to fully guiding the training of all model parameters. Although we focus on classification [19], our method could be repurposed for regression [58, 67] and other supervised tasks [60, 68, 69]. Conformal inference can also be utilized to test hypotheses and calibrate probabilities [70], and our work could be extended to those problems.

## 2 Relevant background on conformal inference

### 2.1 Uncertainty quantification via conformal prediction sets

Consider a data set of i.i.d. (or sometimes simply *exchangeable*) observations $(X_i, Y_i)_{i=1}^{n+1}$ sampled from an arbitrary unknown distribution $P_{XY}$. Here, $X_i \in \mathbb{R}^p$ contains $p$ features for the $i$th sample, and $Y_i \in \{1, \ldots, K\} = [K]$ denotes its label, which we assume to be one of $K$ possible categories. The goal is to train a model on $n$ data points, $(X_i, Y_i)_{i=1}^{n}$, and construct a reasonably small prediction set $\hat{\mathcal{C}}_{n,\alpha} \subseteq [K]$ for $Y_{n+1}$ given $X_{n+1}$ such that, for some fixed level $\alpha \in (0, 1)$,

$$\mathbb{P}\left[Y_{n+1} \in \hat{\mathcal{C}}_{n,\alpha}(X_{n+1})\right] \geq 1 - \alpha. \tag{1}$$

This property is called *marginal coverage* because it treats $(X_i, Y_i)_{i=1}^{n+1}$ as all random. If $\alpha = 0.1$, it ensures $Y_{n+1}$ is contained in the prediction sets 90% of the time. Marginal coverage is practically

feasible but not fully satisfactory, as it is not as reliable and informative as *conditional coverage*:

$$\mathbb{P}\left[Y_{n+1} \in \hat{\mathcal{C}}_{n,\alpha}(x) \mid X_{n+1} = x\right] \geq 1 - \alpha, \qquad \forall x \in \mathbb{R}^p. \tag{2}$$

Conditional coverage would give one confidence that $\hat{\mathcal{C}}_{n,\alpha}(x)$ contains the true $Y$ for any individual data point, which is stronger than (1). For example, imagine a population in which color is a feature and 90% of samples are blue while the others are red. Then, 90% marginal coverage is attained by any prediction sets that contain the true $Y$ for all blue samples but never do for the red ones. Valid coverage can be obtained conditional on a given *protected category* [20], but it is impossible to guarantee (2) more generally without unrealistically strong assumptions [71, 72]. Thus, a typical compromise is to construct prediction sets with marginal coverage and hope they are also reasonably valid conditional on $X$. For multi-class classification, a solution is offered by the conformity scores developed in [19], which are reviewed below as the starting point of our contribution.

## 2.2 Review of adaptive conformity scores for classification

Imagine an *oracle* knowing the conditional distribution of $Y$ given $X$, namely $P_{Y|X}$, and think of how it would construct the smallest possible prediction sets $C_{\alpha}^{\text{oracle}}(x)$ with exact $1 - \alpha$ conditional coverage. For any $x \in \mathcal{X}$ and $y \in [K]$, define $\pi_y(x) = \mathbb{P}[Y = y \mid X = x]$. Then, the oracle would output the smallest subset $S \subseteq [K]$ such that $\sum_{y \in S} \pi_y(x) \geq 1 - \alpha$. In truth, this set may have coverage strictly larger than $1 - \alpha$ due to the discreteness of $Y$; however, exact coverage can be achieved by introducing a little extra randomness [20]. For any nominal coverage level $\tau \in (0, 1)$ and any random noise variable $u \in (0, 1)$, let $\mathcal{S}$ be the function with input $x, u, \pi$, and $\tau$ that computes the set of most likely labels up to (but possibly excluding) the one identified by the above deterministic oracle: $\mathcal{S}(x, u; \pi, \tau) \subseteq [K]$. Note that the explicit expression of $\mathcal{S}$ is written in Appendix A1.1. If $u = U$ is a uniform random variable independent of everything else, it is easy to verify that the following oracle prediction sets have exact conditional coverage at level $\tau = 1 - \alpha$:

$$C_{\alpha}^{\text{oracle}}(x) = \mathcal{S}(x, U; \pi, 1 - \alpha). \tag{3}$$

For example, if $\pi_1(x) = 0.3$, $\pi_2(x) = 0.6$, and $\pi_3(x) = 0.1$, then $C_{0.2}^{\text{oracle}}(x) = \{1, 2\}$ with probability 2/3, and $C_{0.2}^{\text{oracle}}(x) = \{2\}$ with probability 1/3. Of course, this oracle is only a thought experiment because $P_{Y|X}$ is generally unknown. Therefore, to construct practical prediction intervals, $\pi$ must be replaced by an ML model $\hat{\pi}$. Then, conformal inference is needed to ensure the output sets based on the possibly inaccurate ML model at least satisfy marginal coverage (1).

Conformal inference [19] begins by training any classifier on part of the data, indexed by $\mathcal{D}_1 \subset [n]$, to fit an approximation $\hat{\pi}$ of the unknown $\pi$. After substituting $\hat{\pi}$ into the oracle rule $\mathcal{S}$, the hold-out data indexed by $\mathcal{D}_2 = [n] \setminus \mathcal{D}_1$ are leveraged to adjust the prediction sets as to empirically achieve the desired coverage. More precisely, define the following *conformity scores* $W_i$ for all observations in $\mathcal{D}_2$, and for the test point $n + 1$. Intuitively, $W_i$ is the smallest $\tau \in [0, 1]$ such that $\mathcal{S}(X_i, U_i; \hat{\pi}, \tau)$ contains the true $Y_i$, while $U_i$ is a uniform random variable independent of everything else:

$$W_i = W(X_i, Y_i, U_i; \hat{\pi}) = \min\left\{\tau \in [0, 1] : Y_i \in \mathcal{S}(X_i, U_i; \hat{\pi}, \tau)\right\}, \qquad i \in \mathcal{D}_2 \cup \{n + 1\}. \tag{4}$$

These statistics are observed for all $i \in \mathcal{D}_2$, but not for $n + 1$. Define also $\hat{\tau}_{n,\alpha}$ as the $\lceil (1 - \alpha)(1 + |\mathcal{D}_2|) \rceil$ largest element of $\{W_i\}_{i \in \mathcal{D}_2}$. Intuitively, $\hat{\tau}_{n,\alpha}$ is the smallest $\tau \in [0, 1]$ such that $\mathcal{S}(X_i; \hat{\pi}, \tau)$ contains a fraction $1 - \alpha$ of the hold-out data in $\mathcal{D}_2$. Then, the output prediction set for a new $X_{n+1}$ is $\mathcal{C}_{n,\alpha}(X_{n+1}) = \mathcal{S}(X_{n+1}, U_{n+1}; \hat{\pi}, \hat{\tau}_{n,\alpha})$. This has $1 - \alpha$ marginal coverage due to the exchangeability of the calibration and test data [19]. In fact, $Y_{n+1} \notin \mathcal{C}_{n,\alpha}(X_{n_1})$ implies $W_{n+1} > \hat{\tau}_{n,\alpha}$, and by exchangeability the probability of this event is smaller than $\alpha$; see [58]. Unlike alternative approaches based on different scores [17, 18, 53], this solution would yield prediction sets equivalent to those of the oracle [19] if $\hat{\pi} = \pi$. Although generally $\hat{\pi} \neq \pi$, the above prediction sets often achieve relatively high conditional coverage [19] in practice. Our goal is to further improve their empirical performance by training the ML model to be more deliberately aware of uncertainty.

# 3 Methods

## 3.1 The distribution of the adaptive conformity scores

The conformity scores defined in (4) are uniformly distributed conditional on $X = x$, for any $x$, if $\hat{\pi} = \pi$. This property was hinted without proof in [19] and it serves as the starting point of our contribution. Note that all mathematical proofs can be found in Appendix A2.

**Proposition 1.** *The distribution of the conformity scores $W_i$ in (4) is uniform conditional on $X_i$ if $\hat{\pi} = \pi$. That is, $\mathbb{P}[W(X, Y, U; \pi) \leq \beta \mid X = x] = \beta$ for all $\beta \in (0, 1)$, where $(X, Y)$ is a random sample from $P_{X,Y}$, and $U \sim \text{Uniform}[0, 1]$ independent of everything else. Further, $W_i \mid X_i$ is uniform if and only if $\mathcal{S}(X_i; \hat{\pi}, 1 - \alpha)$ has conditional coverage at level $1 - \alpha$ for all $\alpha \in [0, 1]$.*

As we seek accurate conditional coverage, this result suggests training $\hat{\pi}$ as to produce scores that are approximately uniform on hold-out data, at least marginally. Therefore, we will evaluate (4) on hold-out data *while training* $\hat{\pi}$, encouraging the conformity scores to follow a uniform distribution.

## 3.2 An uncertainty-aware conformal loss function

We develop a loss function that approximately measures the deviation from uniformity of the conformity scores defined in (4) by combining classical non-parametric tests for equality in distribution with fast algorithms for smooth sorting and ranking [73, 74]. This loss is combined with the traditional cross entropy as to also promote accurate predictions, and it can be approximately optimized by stochastic gradient descent (it will generally be non-convex). The novel uncertainty-aware component of this loss only sees the hold-out samples through the lens of a non-parametric test applied to the empirical score distribution within a subset of the data. Therefore, it provides little incentive to overfit compared to a traditional loss targeting point-wise predictive accuracy, such as the cross entropy. By contrast, it discourages overconfident predictions which would yield non-uniform scores, as we shall see below. This solution is outlined in Figure A1, Appendix A1.2, and detailed below.

First, the $n$ training samples are partitioned into two subsets, $\mathcal{I}_1$ and $\mathcal{I}_2$ such that $\mathcal{I}_1 \cup \mathcal{I}_2 = [n] = \mathcal{D}_1$. Here we assume the training data are indexed by $\mathcal{D}_1 = [n]$; this notation is slightly different from Section 2, but it is simple and does not introduce ambiguity because the additional calibration data in $\mathcal{D}_2$ remain untouched during training. The training algorithm approximately minimizes a loss function $\ell$ consisting of two additive components, each evaluated on one subset of the data:

$$\ell = (1 - \lambda) \cdot \ell_a(\mathcal{I}_1) + \lambda \cdot \ell_u(\mathcal{I}_2). \tag{5}$$

Above, the hyper-parameter $\lambda \in [0, 1]$ controls the relative weights of the two components. The $\ell_a$ component is evaluated on the data in $\mathcal{I}_1$, and its purpose is to seek high predictive accuracy, as customary. For example, this could be the cross entropy:

$$\ell_a = -\frac{1}{|\mathcal{I}_1|} \sum_{i \in \mathcal{I}_1} \sum_{c=1}^{K} \mathbb{1}\left[Y_i = c\right] \log \hat{\pi}_c(X_i), \tag{6}$$

where $\hat{\pi}$ is the output of the final softmax layer. The novel uncertainty-aware component $\ell_u$ is evaluated on $\mathcal{I}_2$, and its role is to mitigate overconfidence. Concretely, conformity scores $W_i$ are evaluated according to (4) for all $i \in \mathcal{I}_2$, and their empirical distribution is compared to the ideal uniformity expected if $\hat{\pi} = \pi$. Ideally, we would like to quantify this discrepancy by directly applying a powerful non-parametric test, for example by computing the Cramér-von Mises [75, 76] or Kolmogorov-Smirnov [77, 78] test statistics. Concretely, in the latter case,

$$\ell_u = \sup_{w \in [0,1]} \left| \hat{F}_{|\mathcal{I}_2|}(w) - w \right|, \tag{7}$$

where $\hat{F}_{|\mathcal{I}_2|}(\cdot)$ is the empirical cumulative distribution function (CDF) of $W_i$ for $i \in \mathcal{I}_2$: $\hat{F}_{|\mathcal{I}_2|}(w) = (1/|\mathcal{I}_2|) \sum_{i \in \mathcal{I}_2} \mathbb{1}\left[W_i \leq w\right]$. Unfortunately, $\hat{F}_{|\mathcal{I}_2|}(\cdot)$ is not differentiable, which makes the overall loss intractable to minimize. This requires introducing some approximations in $\ell_u$, as explained next.

## 3.3 Differentiable approximations

The empirical CDF of the conformity scores is not differentiable because it involves sorting, which is a non-smooth operation. Further, these scores themselves are not differentiable in the model parameters $\theta$ because they involve ranking and sorting the estimated class probabilities $\hat{\pi}$. In fact, the score (4) can be computed in a closed form,

$$W_i = \hat{\pi}_{(1)}(X_i) + \hat{\pi}_{(2)}(X_i) + \ldots + \hat{\pi}_{(r(Y_i, \hat{\pi}(X_i)))}(X_i) - U_i \cdot \hat{\pi}_{(r(Y_i, \hat{\pi}(X_i)))}(X_i), \tag{8}$$

where $U_i$ is a uniform random variable independent of everything else; see Appendix A1.1 for details. Fortunately, there exist fast approximate algorithms for differentiable sorting and ranking that

work well in combination with standard back-propagation [73, 74]. Note that evaluating $W_i$ in (8) requires accessing elements of $(\hat{\pi}_{(1)}(X_i), \ldots, \hat{\pi}_{(K)}(X_i))$ through a $\theta$-dependent index, $r(Y_i, \hat{\pi}(X_i))$, which is also non-differentiable. Therefore, indexing by $r(Y_i, \hat{\pi}(X_i))$ must be approximated with a smooth linear interpolation; see Appendix A1.2 for further details. In conclusion, the $\ell_u$ loss in (7) is approximated by evaluating a differentiable version of the scores in (4) as described above, and then by replacing their empirical CDF with a differentiable approximation obtained with the same techniques from [74]. This procedure, combined with stochastic gradient descent for fitting the model parameters $\theta$, is summarized in Algorithm 1. Although here we assume $\mathcal{I}_1 = \mathcal{I}_2$ for simplicity, this algorithm can easily accommodate $\mathcal{I}_1 \neq \mathcal{I}_2$. A more technically detailed version of Algorithm 1 is provided in Appendix A1.2, and an open-source software implementation of this method is available online at `https://github.com/bat-sheva/conformal-learning`.

---

**Algorithm 1:** Conformalized uncertainty-aware training of deep multi-class classifiers

---

**Input:** Data $\{X_i, Y_i\}_{i=1}^n$; hyper-parameter $\lambda \in [0, 1]$, learning rate $\gamma > 0$, batch size $M$;
Randomly initialize the model parameters $\theta^{(0)}$;
Randomly split the data into two disjoint subsets, $\mathcal{I}_1, \mathcal{I}_2$, such that $\mathcal{I}_1 \cup \mathcal{I}_2 = [n]$;
Set the number of batches to $B = (n/2)/M$ (assuming for simplicity that $|\mathcal{I}_1| = |\mathcal{I}_2|$);
**for** $t = 1, \ldots, T$ **do**
    Randomly divide $\mathcal{I}_1$ and $\mathcal{I}_2$ into $B$ batches;
    **for** $b = 1, \ldots, B$ **do**
        Evaluate (softmax) conditional probabilities $\hat{\pi}(X_i)$ for all $i$ in batch $b$ of $\mathcal{I}_1 \cup \mathcal{I}_2$;
        Generate a uniform independent random variable $U_i$ for all $i$ in batch $b$ of $\mathcal{I}_2$;
        Evaluate $\tilde{W}_i$ for all $i$ in batch $b$ of $\mathcal{I}_2$, using $U_i$ and a differentiable approximation of (4);
        Evaluate the gradient $\nabla \ell_a(\theta^{(t)})$ of $\ell_a$ in (6) using the data in batch $b$ of $\mathcal{I}_1$;
        Evaluate the gradient $\nabla \tilde{\ell}_u(\theta^{(t)})$ of a differentiable approximation $\tilde{\ell}_u$ of $\ell_u$ in (7) using
          the differentiable scores $\tilde{W}_i$ in batch $b$ of $\mathcal{I}_2$;
        Define $\nabla \tilde{\ell}(\theta^{(t)}) = (1 - \lambda) \cdot \nabla \ell_a(\theta^{(t)}) + \lambda \cdot \nabla \tilde{\ell}_u(\theta^{(t)})$ based on (5);
        Update the model parameters: $\theta^{(t)} \leftarrow \theta^{(t-1)} - \gamma \nabla \tilde{\ell}(\theta^{(t-1)})$.
    **end**
**end**
**Output:** The model $\hat{\pi}$ corresponding to the fitted parameters $\theta^{(T)}$.

---

### 3.4 Theoretical analysis

The uncertainty-aware loss function in Algorithm 1 can be justified theoretically by noting that it is approximately minimized (although possibly non-uniquely) by the imaginary oracle model $\pi$, which yields the smallest possible prediction sets with exact conditional coverage. This analysis focuses on the original version of the loss function defined in (5)–(7), ignoring for simplicity the additional subtleties introduced by the differentiable approximations described in Section 3.3.

**Proposition 2.** *The loss function $\ell \geq 0$ in (5) is bound from above by $\ell^0 + \delta \ell$, where $\ell^0 \geq 0$ attains value zero if $\hat{\pi} = \pi$, and $\delta \ell = \mathcal{O}_\mathbb{P}(1/\sqrt{M})$ as $\mathcal{I}_2 \to \infty$.*

Of course, Algorithm 1 does not minimize (5) exactly because it involves solving a high-dimensional non-convex optimization problem that is difficult to study theoretically. Yet, it is possible to prove at least a weak form of convergence for its stochastic gradient descent, whose solution may not however necessarily approach a global minimum. This analysis is in Appendix A2 for lack of space.

## 4 Numerical experiments

### 4.1 Experiments with synthetic data

The performance of Algorithm 1 is investigated here on synthetic data that mimic a multi-class classification problem in which most samples are relatively easy to classify but a few are unpredictable. Specifically, data are simulated with 100 independent and uniformly distributed features $X = (X_1, \ldots, X_{100}) \in [0, 1]^{100}$ and a label $Y \in [K]$, for $K = 6$. The first feature controls whether the

sample is intrinsically difficult to classify, while the next two features determine the most likely labels; all other features are useless. On average, 20% of the samples are impossible to classify with absolute confidence. This conditional distribution is written explicitly in Appendix A3.1.

The conditional class probabilities $\hat{\pi}$ are estimated as the output of a final softmax layer in a fully connected neural network implemented with PyTorch [79]; see Appendix A3.1 for more information about network architecture and training details. This model is fitted separately with Algorithm 1 and three benchmark techniques including traditional cross entropy minimization and focal loss minimization [80]. Unfortunately, we cannot directly compare to the recent methods of [62, 63] as originally implemented by those authors for lack of openly available computer code. Instead, we consider a hybrid benchmark that combines elements of Algorithm 1 with the main idea of [63], essentially seeking a model that yields small conformal prediction sets, irrespective of conditional coverage; see Appendix A1.3 for details about this benchmark. As early stopping can help mitigate overfitting [41], it is informative to also investigate its effect on each of the aforementioned learning algorithms. For this purpose, we generate an additional validation set of 2000 independent data points and use it to preview the out-of-sample accuracy and loss value at each epoch. Then, the best versions of each model according to these two early stopping criteria are saved during training. After training each model, 10,000 additional independent samples are utilized to calibrate split-conformal prediction sets with 90% marginal coverage, as explained in Section 2.2. All models, including those trained with our method, undergo conformal calibration with these independent data points prior to constructing the prediction sets, as this step is necessary to theoretically guarantee marginal coverage. Further, we ensure the comparisons between different models are always fair by giving each training algorithm access to exactly the same data set, regardless of whether its inner workings involve sample splitting to evaluate separately distinct components of the loss function.

Figure 1 compares the performance of conformal prediction sets obtained with each model for 2000 test points as a function of the number of training samples, averaging over 50 independent experiments. The prediction sets are evaluated in terms of their average size and coverage conditional on $X_1 \le \delta$; i.e., separately for the "hard" samples. For each learning algorithm, the results corresponding to the model achieving the highest conditional coverage among the fully trained and two early stopped alternatives are reported. This allows us to focus on the overall behaviour of different losses while accounting for possible differences in the optimal choices of early-stopping strategies.

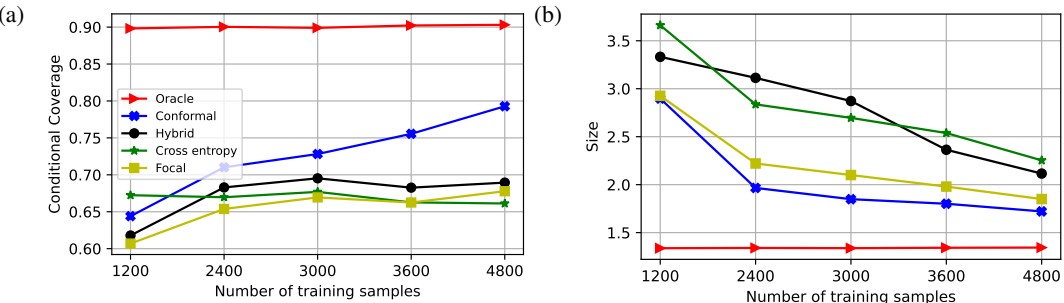

Figure 1: Performance of conformal prediction sets with 90% marginal coverage based on the ideal oracle model and a deep neural network trained with four alternative algorithms. (a): Conditional coverage for intrinsically hard samples. (b): Average size of prediction sets. The results are averaged over 50 independent experiments and the standard errors are below 0.1 (not shown explicitly).

Figure 1 does not visualize marginal coverage because that is guaranteed to be above 90%. The results show our algorithm yields the prediction sets with highest conditional coverage and smallest size, especially when the number of training samples is large. By contrast, the conditional coverage obtained with the cross entropy does not visibly improve as the training data set grows, suggesting systematic over-confidence with hard-to-classify samples. The focal loss improves upon the cross entropy by reducing the average size of the prediction sets, but it does not lead to higher conditional coverage. Finally, the hybrid algorithm increases conditional coverage slightly compared to the cross entropy, but it does not lead to smaller prediction sets. This is likely because the hybrid loss does not introduce very different incentives compared to the cross entropy, as the latter already attempts to maximize the estimated probability of the observed labels, thus effectively seeking small conformal

prediction sets without necessarily high conditional coverage. Note that the focal loss is applied here with hyper-parameter $\gamma = 1$, as we have found larger values to yield lower accuracy; see Figure A2. In this section, our method is applied with the hyper-parameter $\lambda = 0.2$ in (5); additional results obtained with different values of $\lambda$ are in Figures A3–A4, Appendix A3.2.

The improved performance of our method can be understood by looking at the distribution of the corresponding conformity scores for test data, as shown in Figure A6 for the model trained on 2400 samples. Our method leads to scores that are more uniformly distributed than those obtained with the cross entropy, which is indicative of more reliable uncertainty quantification. In fact, our estimated conditional probabilities $\hat{\pi}$ are more similar to the true oracle $\pi$ compared to those obtained by minimizing the cross entropy; see Figure A7. All figures here refer to fully trained models, without early stopping; analogous results with early stopping are presented later. It is worth emphasizing that post-training conformal calibration is always necessary to guarantee marginal coverage, regardless of how the model is trained. At the same time, it is interesting to observe that models trained with our method tend to estimate more reliable probabilities compared to the benchmarks. Therefore, it should not be surprising that our models lead to larger prediction sets with relatively high marginal coverage if we skip the post-training conformal calibration step; see Figure A5.

Figure A8 compares the performance of the prediction sets obtained with each method when covariate shift occurs in the test data. In particular, here we imagine that at test time the uncertainty-controlling feature $X_1$ is sampled uniformly from $[0, a]$ with $a \leq 1$, so that lower values of $a$ correspond to higher proportions of intrinsically hard-to-classify samples. Of course, in this case marginal coverage is no longer guaranteed for the same reason why conditional coverage in Figure 1 is not always controlled. As expected, all models produce smaller set sizes with higher marginal coverage for $a$ closer to 1, consistently with Figure 1, but our method outperforms the benchmarks.

Several additional results are in Appendix A3.2. Figure A9 reports on experiments in which the number $K$ of labels is varied, ranging from 4 to 12. These results show our methods leads to prediction sets with consistently smaller size and typically higher conditional coverage compared to all benchmarks. Figure A10 reports on experiments in which the proportion of hard-to-classify samples is varied, ranging from 0.1 to 0.5. All models lead to higher conditional coverage for larger $\delta$, as implied by the fixed marginal coverage, but our method consistently achieves it with smaller prediction sets. Figures A11–A14 report on experiments with models trained using early stopping based on maximum prediction accuracy on the validation data. Again, our method achieves higher conditional coverage and smaller prediction sets relative to the benchmarks. Figures A15–A18 report analogous results obtained with early stopping based on minimum validation loss. Figures A19–A20 (resp. A21–A22) show the distribution of conformity scores and the estimated class probabilities, as in Figures A6–A7, from models trained with early stopping based on validation predictive accuracy (resp. loss). Finally, Figures A23–A24 (resp. A25–A26) show the distribution of conformity scores and the estimated class probabilities from the focal loss (res. hybrid) models.

## 4.2 Experiments with CIFAR-10 data

Convolutional neural networks guided by the conformal loss are trained on the publicly available CIFAR-10 image classification data set [81] (10 classes), and the models thus obtained are compared to those targeting the three benchmark losses considered before. As these data are not too hard to classify, we make the problem more interesting by randomly applying RandomErasing [82] to some images—a form of corruption that makes images very hard to recognize. The number of training samples is varied from 3000 to 45000. See Appendix A3.3 for details. To measure the performance of conformal prediction sets based on each model, we set aside 5000 calibration and test observations prior to training. All models are calibrated after training, as in the previous section, in order to guarantee valid marginal coverage. The proportions of corrupt images in the calibration and test sets are 0.2, while that in the training set is varied. All models are calibrated as in [19], seeking 90% marginal coverage. Their performance is measured on the test data in terms of the size of the output prediction sets and their coverage conditional on the indicator of corruption. Further, the test accuracy of each model is evaluated based on the misclassification rate of its best-guess label. Figure 2 showcases two example test images, respectively intact and corrupted by RandomErasing, along with their corresponding conditional class probabilities calculated by different models fully trained on 45000 data points. This shows the model trained with our conformal loss is not as overconfident when dealing with the corrupted images as that minimizing the cross entropy.

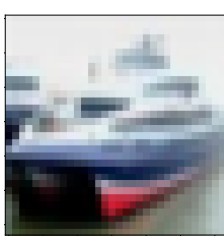

Conformal
{Ship: 1.00}

Hybrid
{Ship: 1.00}

Cross entropy
{Ship: 0.96, Car: 0.04}

Focal
{Car: 0.95, Ship: 0.03}

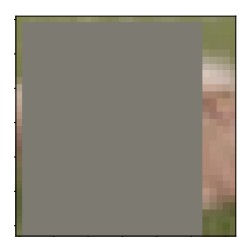

Conformal
{Dog: 0.34, Cat: 0.32}

Hybrid
{Dog: 0.99, Horse: 0.01}

Cross entropy
{Deer: 0.99, Dog: 0.01}

Focal
{Deer: 0.84, Dog: 0.08}

Figure 2: Two example test images from the CIFAR-10 data set, with their top two estimated class probabilities computed by the output softmax layer of convolutional neural networks trained to minimize different loss functions. Left: intact image of a ship. Right: corrupted image of a dog.

Figure A27 summarizes the performance of the prediction sets obtained with each fully trained model, as a function of the number of training samples. Median performance measures are reported over 10 experiments with random data subsets. The conformal loss leads to higher coverage for intrinsically hard images, and smaller prediction sets for the easy ones. As shown in Figure A28, this improvement is associated with higher test accuracy for the models targeting our loss, consistently with their increased robustness to overfitting. As overfitting can also be mitigated by early stopping, we report in Figure A29 the corresponding results obtained with early stopping based on maximum accuracy on a validation data set of size 2000 (or 5000, training with 45000 samples). In this case, all models achieve similar test accuracy, but those trained with our method lead to conformal prediction sets with (slightly) higher conditional coverage and smaller size, especially if trained on many samples and compared to the focal loss. These conclusions are summarized in Table 1 in the case of 45000 training samples, which includes also the results corresponding to models trained with early stopping based on validation accuracy. See Table A1 for results with early stopping based on validation loss. Overall, the models targeting the conformal loss perform best when trained fully or with early stopping based on accuracy, and they tend to achieve higher coverage for the hard images while enabling smaller prediction sets with valid coverage for the easy cases. A similar summary for the models trained with 3000 samples is in Table A2; there, the advantage of the conformal loss compared to the hybrid method is not as marked as in the large-sample experiments. Note that the models in Table 1 have accuracy below 83% due to the corrupted images in the training data; by comparison, 89% accuracy could be achieved by applying the cross-entropy method on the clean data set [83].

In Appendix A3.4, Figures A30 and A31 report on results similar to those in Figures A27 and A29, respectively, but using prediction sets directly constructed based on the probability estimates computed by each model, without conformal calibration. Consistently with the experiments described in the previous section, these results show our models lead to prediction sets with relatively high marginal coverage, especially if the sample size is large. Of course, post-training conformal calibration remains necessary to guarantee the nominal coverage level in finite samples. Figures A32 and A33 visualize some concrete examples of prediction sets obtained with and without post-training conformal calibration, respectively for intact and corrupt images. These results confirm the prediction sets obtained with our method tend to be less overconfident compared to the benchmarks, whether or not post-training conformal calibration is applied to guarantee marginal coverage.

| | Coverage intact/corrupted | | Size intact/corrupted | | Accuracy intact/corrupted | |
| --- | --- | --- | --- | --- | --- | --- |
| | Full | ES (acc) | Full | ES (acc) | Full | ES (acc) |
| Conformal | 0.90/0.90 | 0.90/0.87 | 1.41/5.95 | 1.41/6.09 | 0.81/0.35 | 0.81/0.35 |
| Hybrid | 0.92/0.81 | 0.91/0.86 | 5.54/5.75 | 1.38/5.30 | 0.65/0.40 | 0.83/0.37 |
| Cross Entropy | 0.92/0.84 | 0.92/0.81 | 3.30/4.43 | 1.50/5.05 | 0.68/0.43 | 0.82/0.36 |
| Focal | 0.91/0.83 | 0.93/0.77 | 2.57/3.99 | 1.91/4.25 | 0.68/0.42 | 0.77/0.34 |

Table 1: Conditional coverage and size of conformal prediction sets with 90% marginal coverage on CIFAR-10 data, based on models trained on 45000 data points. The models are trained either fully for many epochs, or with early stopping (ES) based on classification accuracy (acc). The last two columns report the best-guess classification accuracy of the underlying models applied to test data.

Figure A34 reports performance measures as in Figure A27, after fixing the number of training samples to 45000 and varying the corruption proportion. The model trained with the conformal loss leads to smaller prediction sets with higher conditional coverage compared to the benchmarks, and its test accuracy does not decrease as the proportion of corrupted training images increases. Figure A35 demonstrates early stopping mitigates overfitting and allows the benchmarks to maintain relatively high accuracy as the training corruption proportion increases. However, the conformal loss outperforms even if all models are trained with early stopping based on validation accuracy. Figure 3 reports on experiments similar to those in Table 1 but performed under covariate shift. Here, the proportions of corrupt images in the training and calibration data sets are still fixed to 0.2, while the proportion of corrupt images in the test set is varied from 0.2 (no covariate shift) to 1.0 (extreme covariate shift). In these experiments, no model is theoretically guaranteed to achieve valid marginal coverage due to the covariate shift, but the model trained with our method remains approximately valid because it has practically high conditional coverage. In conclusion, these results confirm the model trained with our method can provide reliable uncertainty estimates for both intact and the corrupt images, while the benchmark models tend to be overconfident in the hard-to-classify cases.

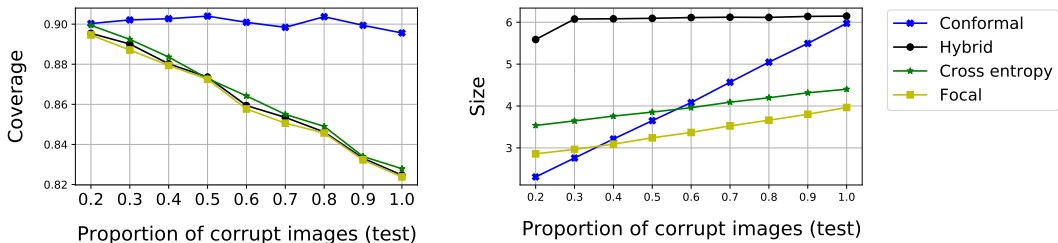

Figure 3: Marginal coverage and average size of conformal prediction sets obtained with models trained by different methods, on the CIFAR-10 data under covariate shift. The results are shown as a function of the proportion of corrupt images in the test set. Other details are as in Table 1.

The distributions of conformity scores obtained with different methods on test data are compared in Figures A36–A39. These results confirm the scores obtained with the conformal loss tend to be more uniform compared to the benchmarks, especially if the training sample size is large. If few training samples are available, the focal loss yields scores that are slightly closer to being uniform, but at the cost of lower accuracy and conditional coverage (Table A2). In Figures A40–A43, the performances of all models are compared separately for each of the 10 possible test labels. These results suggest corrupt images with true labels 4,5, or 6 are most difficult to classify, and their prediction sets have the lowest coverage. However, this issue is mitigated by models fully trained to minimize the proposed conformal loss. Further, the results confirm that models trained with our loss yield more informative prediction sets for the easier unperturbed images, while achieving the desired coverage rate.

### 4.3 Experiments with credit card data

In this section, we analyze a publicly available credit card default data set [84] containing 30000 observations of 23 features and a binary label. Approximately 22% of the labels are equal to 1. The data are randomly divided into 16800 training samples, 4500 calibration samples, and 4500 test samples, while the remaining 4200 samples are utilized to determine early stopping rules. All experiments are repeated 20 times with independent data subsets. The conformal and hybrid losses are implemented using 70% of the training samples for evaluating the cross entropy component. The model architecture is as in Section 4.1; see Appendix A3.5 for further implementation details.

The performances of conformal prediction sets for models trained with different losses are measured in terms of their respective sizes and coverage conditional on the true label being equal to 1. As the samples with label 1 are a minority, constructing prediction sets with valid coverage for their group is an interesting problem that may be relevant in the context of algorithmic fairness [20]. Note that it is possible to *calibrate* conformal prediction sets in such a way as to achieve perfect label-conditional coverage [17, 19], at the cost of higher data usage, but here we focus on *training* uncertainty-aware models as to approximately achieve this goal without explicit label-conditional calibration. Therefore, we apply a slightly modified version of Algorithm 1 in which the empirical distribution of the hold-out

conformity scores is evaluated separately on the samples from each class, and then the sum of these two instances of (7) is utilized as $\ell_u$. The same strategy is also incorporated into the hybrid method.

Table 2 compares the average performances of each model and shows our method achieves the highest conditional coverage. Here, early stopping is applied based on minimum validation loss, as the alternative accuracy criterion achieved lower conditional coverage for all models. Our conformal models have relatively high classification errors on these data, but this can be explained by the high class imbalance. Thus, the error rate in Table 2 may be less informative as a performance metric than the corresponding F-score reported in Figure A44, which is higher (better) for the models trained with our method. Finally, Figure A45 and Table A3 in Appendix A3.6 demonstrate our model leads to conformity scores that are closer to being uniformly distributed on test data, as expected.

| | Coverage | | | | Size all labels/0/1 | | Classification error | |
| | Marginal | | Conditional | | | | | |
| | Full | ES | Full | ES | Full | ES | Full | ES |
|---|---|---|---|---|---|---|---|---|
| Conformal | 0.83 | 0.83 | 0.60 | 0.52 | 1.34/1.31/1.45 | 1.29/1.27/1.39 | 33.05 | 28.52 |
| Hybrid | 0.83 | 0.83 | 0.51 | 0.53 | 1.27/1.24/1.37 | 1.28/1.25/1.38 | 27.00 | 27.52 |
| Cross Entropy | 0.82 | 0.84 | 0.51 | 0.42 | 1.25/1.23/1.33 | 1.24/1.21/1.35 | 26.16 | 24.36 |
| Focal | 0.81 | 0.82 | 0.48 | 0.50 | 1.22/1.20/1.28 | 1.25/1.23/1.32 | 26.56 | 26.30 |

Table 2: Performance of conformal prediction sets with 80% marginal coverage on credit card default data, based on convolutional neural networks targeting different losses. Other details are as in Table 1.

## 5  Discussion

The conformal loss function presented in this paper mitigates overconfidence in deep neural networks and can lead to smaller prediction sets with higher conditional coverage compared to standard benchmarks. This contribution is practically relevant for many applications in which overfitting may occur, but it is especially useful when dealing with noisy data that do not allow very accurate out-of-sample predictions. One limitation of the proposed method is that it is more computationally expensive than its benchmarks. For example, training a conformal loss model on 45000 images in the CIFAR-10 data set took us approximately 20 hours on an Nvidia P100 GPU, while training models with the same architecture to minimize the cross entropy or focal loss only took about 11 hours. Another limitation of the conformal loss is that a relatively large number of training samples appears to be required for significant performance improvements. Nonetheless, the ability of an ML model to more openly admit ignorance when asked to provide an unknown answer is a valuable achievement for which it may sometimes be worth investing additional training resources.

Future research may explore extensions of our method for uncertainty-aware learning to problems beyond multi-class classification or to ML models other than deep neural networks, as well as further applications to real-world data sets. Focusing more closely on image classification, it would be interesting to combine the method proposed in this paper with data augmentation techniques that can further increase predictive accuracy [83], especially if the available training set is limited [85]. Further, it could be useful to combine our method with different random image corruption techniques as well as data augmentation in order to improve robustness to covariate shift and adversarial cases. Such extensions are not straightforward because data augmentation may violate the sample exchangeability assumptions, but recent theoretical advances in conformal inference may open the door to a principled solution [86, 87]. We plan to explore these directions in future work.

Software implementing the proposed method is available online at `https://github.com/bat-sheva/conformal-learning`, with all code needed to reproduce the numerical experiments.

**Acknowledgements**

M. S. and Y. Z. are supported by NSF grant DMS 2210637. M. S. is also supported by an Amazon Research Award. B. E. and Y. R. were supported by the Israel Science Foundation (grant No. 729/21). Y. R. thanks the Career Advancement Fellowship, Technion, for providing research support. The authors also thank four anonymous reviewers for helpful comments and an insightful discussion.

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
