# Appendix to:
# Training Uncertainty-Aware Classifiers with Conformalized Deep Learning

**Bat-Sheva Einbinder**[*][†]    **Yaniv Romano**[†][‡]    **Matteo Sesia**[§]    **Yanfei Zhou**[§]

## A1    Additional methodological details

### A1.1    Conformity scores for multi-class classification

For any $\tau \in [0, 1]$, define as in [8] the *generalized conditional quantile* function:

$$L(x; \pi, \tau) = \min\{k \in \{1, \ldots, K\} \: : \: \pi_{(1)}(x) + \pi_{(2)}(x) + \ldots + \pi_{(k)}(x) \geq \tau\}. \qquad \text{(A1)}$$

Then, recall from that, for any $\tau \in [0, 1]$ and $u \in (0, 1)$, the function with input $x$, $u$, $\pi$, and $\tau$ that computes the set of most likely labels up to (but possibly excluding) the one identified by the deterministic oracle in Section 2.2 is:

$$\mathcal{S}(x, u; \pi, \tau) = \begin{cases} \text{`}y\text{' indices of the } L(x; \pi, \tau) - 1 \text{ largest } \pi_y(x), & \text{if } u \leq V(x; \pi, \tau), \\ \text{`}y\text{' indices of the } L(x; \pi, \tau) \text{ largest } \pi_y(x), & \text{otherwise,} \end{cases} \qquad \text{(A2)}$$

$$V(x; \pi, \tau) = \frac{1}{\pi_{(L(x;\pi,\tau))}(x)} \left[ \sum_{k=1}^{L(x;\pi,\tau)} \pi_{(k)}(x) - \tau \right],$$

where $\pi_{(1)}(x) \geq \pi_{(2)}(x) \geq \ldots \geq \pi_{(K)}(x)$ are the order statistics of $\hat{\pi}_1(x), \ldots, \hat{\pi}_K(x)$. Note that it is typically preferable to skip the randomization step in (A2) if $L(x; \pi, \tau) = 1$, to avoid returning empty prediction sets.

The conformity scores defined in (4) are not differentiable in the model parameters $\theta$ because they involve ranking and sorting the estimated class probabilities $\hat{\pi}$. In fact, $Y_i \in \mathcal{S}(X_i, U_i; \hat{\pi}, \tau)$, where $\mathcal{S}$ is defined as in (A2), if and only if $U_i \geq V(X_i; \hat{\pi}, \tau)$. This means that $Y_i \in \mathcal{S}(X_i, U_i; \hat{\pi}, \tau)$ if and only if $\tau \geq \hat{\pi}_{(1)}(X_i) + \hat{\pi}_{(2)}(X_i) + \ldots + \hat{\pi}_{(r(Y_i, \hat{\pi}(X_i)))}(X_i) - U_i \cdot \hat{\pi}_{(r(Y_i, \hat{\pi}(X_i)))}(X_i)$, where $r(Y_i, \hat{\pi}(X_i))$ denotes the rank of $\hat{\pi}_{Y_i}(X_i)$ among $\hat{\pi}_1(X_i), \ldots, \hat{\pi}_K(X_i)$. Therefore, the conformity scores can be written as in (8):

$$W_i = \hat{\pi}_{(1)}(X_i) + \hat{\pi}_{(2)}(X_i) + \ldots + \hat{\pi}_{(r(Y_i, \hat{\pi}(X_i)))}(X_i) - U_i \cdot \hat{\pi}_{(r(Y_i, \hat{\pi}(X_i)))}(X_i), \qquad \text{(A3)}$$

where $U_i$ is a uniform random variable independent of everything else.

---

[*]Authors listed in alphabetical order.

[†]Department of Electrical and Computer Engineering, Technion, Israel.

[‡]Department of Computer Science, Technion, Israel.

[§]Department of Data Sciences and Operations, University of Southern California, Los Angeles, CA, USA.

36th Conference on Neural Information Processing Systems (NeurIPS 2022).

## A1.2 Further details on the conformal learning algorithm

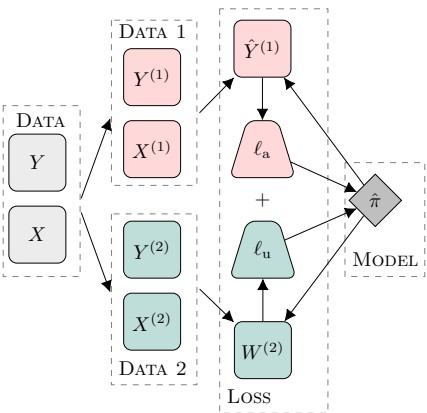

Figure A1: Schematic of the proposed uncertainty-aware deep classification learning algorithm. The training data are split into two subsets. The first subset (in red) is utilized to evaluate a traditional accuracy-based loss function $\ell_{\mathrm{a}}$, such as the cross entropy. The second subset (in green) is utilized to evaluate the deviation from uniformity of the distribution of the model's conformity scores $W$, which is approximated by a differentiable loss function $\ell_{\mathrm{u}}$. The overall additive loss is minimized through gradient descent. The output of the final soft-max layer in the trained model can be interpreted as a more reliable estimate of conditional class probabilities.

Here we explain in detail the implementation of the differentiable loss function $\tilde{\ell}_{\mathrm{u}}$ in Algorithm 1. The function $\tilde{\ell}_{\mathrm{u}}$ takes as input the estimated probabilities for the $i$-th data point as output by the soft-max layer of the model, namely $[\hat{\pi}_1(X_i), \hat{\pi}_2(X_i), \ldots, \hat{\pi}_K(X_i)]$, as well as the corresponding label, $Y_i$. The estimated probabilities are soft-sorted, yielding an array $[\tilde{\pi}_{(1)}(X_i), \tilde{\pi}_{(2)}(X_i), \ldots, \tilde{\pi}_{(K)}(X_i)]$ that approximates the order statistics $[\hat{\pi}_{(1)}(X_i), \hat{\pi}_{(2)}(X_i), \ldots, \hat{\pi}_{(K)}(X_i)]$. Similarly, the estimated probabilities are also soft-ranked, yielding an array $[\tilde{r}(1, \hat{\pi}(X_i)), \tilde{r}(1, \hat{\pi}(X_i)), \ldots, \tilde{r}(K, \hat{\pi}(X_i))]$ that takes continuous values in the interval $[1, K]$ and approximates the true label ranks $[\hat{r}(1, \hat{\pi}(X_i)), \hat{r}(1, \hat{\pi}(X_i)), \ldots, \hat{r}(K, \hat{\pi}(X_i))]$.

To evaluate the differentiable conformity scores $\tilde{W}_i$, we begin by computing a soft approximation of the estimated conditional CDF of the observed label; this is achieved by taking the cumulative sum of $[\tilde{\pi}_{(1)}(X_i), \tilde{\pi}_{(2)}(X_i), \ldots, \tilde{\pi}_{(K)}(X_i)]$. Then, we approximate the value of the CDF at the true label $Y_i$ by soft-indexing the differentiable approximation of the CDF obtained above, obtaining an approximation $\tilde{\pi}_{(1)}(X_i) + \tilde{\pi}_{(2)}(X_i) + \ldots + \tilde{\pi}_{(\tilde{r}(Y_i, \tilde{\pi}(X_i)))}(X_i)$ of $\hat{\pi}_{(1)}(X_i) + \hat{\pi}_{(2)}(X_i) + \ldots + \hat{\pi}_{(\hat{r}(Y_i, \hat{\pi}(X_i)))}(X_i)$. This soft-indexing operation needs to operate with a continuous-valued index and simply consists of a smooth linear interpolation of the target array. Next, we apply soft-sorting to also compute the estimated probability of the observed label, $\tilde{\pi}_{(\tilde{r}(Y_i, \tilde{\pi}(X_i)))}(X_i)$. Thus, the differentiable approximation of the conformity score in (A3) is given by:

$$\tilde{W}_i = \tilde{\pi}_{(1)}(X_i) + \tilde{\pi}_{(2)}(X_i) + \ldots + \tilde{\pi}_{(r(Y_i, \tilde{\pi}(X_i)))}(X_i) - U_i \cdot \tilde{\pi}_{(r(Y_i, \tilde{\pi}(X_i)))}(X_i), \tag{A4}$$

where $U_i$ is a uniform random variable independent of everything else.

Finally, we need a differentiable approximation of the Kolmogorov-Smirnov deviation from uniformity of the differentiable conformity scores $\tilde{W}_i$ obtained above. For this purpose, we compute a smooth version of the empirical CDF of $\tilde{W}_i$ for all $i \in \mathcal{I}_2$; this is achieved by viewing the CDF as a sum of $|\mathcal{I}_2|$ indicator functions, each of which is approximated smoothly using two sigmoid functions. Then, we can obtain a differentiable approximation of the Kolmogorov-Smirnov distance from the uniform distribution by simply taking the maximum distance between the above approximate CDF and the uniform CDF. This procedure is summarized in Algorithm A1, which is a more technical version of Algorithm 1. For further implementation details, we refer our open-source software available online at `https://github.com/bat-sheva/conformal-learning`.

**Algorithm A1:** Conformalized uncertainty-aware training of deep multi-class classifiers

---

**Input:** Data $\{X_i, Y_i\}_{i=1}^n$; hyper-parameter $\lambda \in [0, 1]$, learning rate $\gamma > 0$, batch size $M$;

Randomly initialize the model parameters $\theta^{(0)}$;

Randomly split the data into two disjoint subsets, $\mathcal{I}_1, \mathcal{I}_2$, such that $\mathcal{I}_1 \cup \mathcal{I}_2 = [n]$;

Set the number of batches to $B = (n/2)/M$ (assuming for simplicity that $|\mathcal{I}_1| = |\mathcal{I}_2|$);

**for** $t = 1, \ldots, T$ **do**

    Randomly divide $\mathcal{I}_1$ and $\mathcal{I}_2$ into $B$ batches;

    **for** $b = 1, \ldots, B$ **do**

        Evaluate (softmax) conditional probabilities $\hat{\pi}(X_i)$ for all $i$ in batch $b$ of $\mathcal{I}_1 \cup \mathcal{I}_2$;

        Evaluate the gradient $\nabla \ell_a(\theta^{(t)})$ of $\ell_a$ in (6) using the data in batch $b$ of $\mathcal{I}_1$;

        **for** $i$ *in batch $b$ of $\mathcal{I}_2$* **do**

            Generate a uniform independent random variable $U_i$;

            Soft-sort the estimated probabilities output by the soft-max layer;

            Sum the soft-sorted probabilities to approximate the CDF of $Y \mid X_i$;

            Approximate the rank of $Y_i$ based on the estimated probabilities using soft-ranking;

            Approximate the CDF of the labels at the observed $Y_i$ with soft-indexing;

            Approximate the probability of the observed $Y_i$ with soft-indexing;

            Evaluate differentiable conformity scores $\tilde{W}_i$ with (A4).

        **end**

        Approximate the CDF of $\tilde{W}_i$ for all $i$ in batch $b$ of $\mathcal{I}_2$ using a sum of sigmoid functions;

        Compute $\tilde{\ell}_u$ as the max distance between the above smooth CDF and the uniform CDF;

        Evaluate the gradient $\nabla \tilde{\ell}_u(\theta^{(t)})$ of $\tilde{\ell}_u$;

        Define the total gradient $\nabla \tilde{\ell}(\theta^{(t)}) = (1 - \lambda) \cdot \nabla \ell_a(\theta^{(t)}) + \lambda \cdot \nabla \tilde{\ell}_u(\theta^{(t)})$ based on (5);

        Update the model parameters: $\theta^{(t)} \leftarrow \theta^{(t-1)} - \gamma \nabla \tilde{\ell}(\theta^{(t-1)})$.

    **end**

**end**

**Output:** The model $\hat{\pi}$ corresponding to the fitted parameters $\theta^{(T)}$.

---

### A1.3   Implementation of the hybrid training algorithm

This section explains the implementation of the hybrid benchmark method applied in Section 4. This benchmark is inspired by the recent proposal of [11], although it may not be able to replicate its performance exactly because a software implementation of the latter has not yet been made available. This benchmark is based on a loss function designed to incentivize the trained model to produce the smallest possible conformal prediction sets with the desired coverage (e.g., 90% if $\alpha = 0.1$). The hybrid training procedure is similar to Algorithm 1, in the sense that it relies on analogous soft-sorting, soft-ranking, and soft-indexing algorithms to evaluate a differentiable approximation $\tilde{W}_i$ of the conformity score $W_i$ in (8). However, unlike Algorithm 1, this hybrid method does not compare the distribution of $\tilde{W}_i$ on hold-out data to the ideal uniform distribution. Instead, for all data points $X_i$ in the second training subset $\mathcal{I}_2$, approximate conformity scores $\tilde{W}_i$ are computed and then soft-sorted. This gives us an approximation of the $1 - \alpha$ empirical quantile of $\tilde{W}_i$, which can be used to compute a differentiable approximation of the size of the corresponding conformal prediction set (more precisely, we find the rank of the label at which the empirical CDF of $\tilde{W}_i$ first crosses the $1 - \alpha$ threshold). Then, a differentiable loss function $\tilde{\ell}_s$ is defined as a the average size of these smoothed conformal prediction sets. This plays the role of $\tilde{\ell}_u$ in Algorithm 1, as outlined in Algorithm A2.

**Algorithm A2:** Hybrid conformalized training of deep multi-class classifiers

**Input:** Data $\{X_i, Y_i\}_{i=1}^n$; hyper-parameter $\lambda \in [0,1]$, learning rate $\gamma > 0$, batch size $M$, desired marginal coverage level $\alpha \in (0,1)$;

Randomly initialize the model parameters $\theta^{(0)}$;

Randomly split the data into two disjoint subsets, $\mathcal{I}_1, \mathcal{I}_2$, such that $\mathcal{I}_1 \cup \mathcal{I}_2 = [n]$;

Set the number of batches to $B = (n/2)/M$ (assuming for simplicity that $|\mathcal{I}_1| = |\mathcal{I}_2|$);

**for** $t = 1, \ldots, T$ **do**

    Randomly divide $\mathcal{I}_1$ and $\mathcal{I}_2$ into $B$ batches;

    **for** $b = 1, \ldots, B$ **do**

        Evaluate (softmax) conditional probabilities $\hat{\pi}(X_i)$ for all $i$ in batch $b$ of $\mathcal{I}_1 \cup \mathcal{I}_2$;

        Generate a uniform independent random variable $U_i$ for all $i$ in batch $b$ of $\mathcal{I}_2$;

        Evaluate $\tilde{W}_i$ for all $i$ in batch $b$ of $\mathcal{I}_2$, using $U_i$ and a differentiable approximation of (4);

        Compute a differentiable approximation $\tilde{S}_i$ of the size of the conformal prediction set for $X_i$ at level $1 - \alpha$, for all $i$ in batch $b$ of $\mathcal{I}_2$;

        Define $\tilde{\ell}_s(\theta^{(t)}) = (1/m) \sum_{i \in \mathcal{I}_2} |S_i|$, where $|\cdot|$ is the set size;

        Evaluate the gradient $\nabla \ell_a(\theta^{(t)})$ of $\ell_a$ in (6) using the data in batch $b$ of $\mathcal{I}_1$;

        Evaluate the gradient $\nabla \tilde{\ell}_s(\theta^{(t)})$ of a differentiable approximation $\tilde{\ell}_s$ of $\ell_s$ in (7) using the differentiable scores $\tilde{W}_i$ in batch $b$ of $\mathcal{I}_2$;

        Define $\nabla \tilde{\ell}(\theta^{(t)}) = (1-\lambda) \cdot \nabla \ell_a(\theta^{(t)}) + \lambda \cdot \nabla \tilde{\ell}_s(\theta^{(t)})$ based on (5);

        Update the model parameters: $\theta^{(t)} \leftarrow \theta^{(t-1)} - \gamma \nabla \tilde{\ell}(\theta^{(t-1)})$.

    **end**

**end**

**Output:** The model $\hat{\pi}$ corresponding to the fitted parameters $\theta^{(T)}$.

## A2 Additional theoretical results

### A2.1 Convergence analysis of Algorithm 1

To facilitate the exposition of our analysis, we begin by introducing some helpful notations. Define $\mathcal{D}_1 = (X_i, Y_i)_{i \in \mathcal{I}_1}$ and $\mathcal{D}_2 = (X_i, Y_i)_{i \in \mathcal{I}_2}$: the data subsets utilized for computing the two components of the loss in (5), $\ell_a$ and $\ell_u$ respectively. Let $\mathbf{Z}'$ and $\mathbf{Z}''$ denote randomly chosen batches of $\mathcal{D}_1$ and $\mathcal{D}_2$ at time $t$, respectively, while $\mathbf{U}''$ denotes the corresponding batch of conformity scores $W_i$ at time $t$. The state of Algorithm 1 at time $t$ is determined by $\mathbf{U}''$ and $\zeta^{(t)} = (\mathbf{Z}', \mathbf{Z}'', \theta^{(t)})$, where $\theta^{(t)}$ is the vector containing the current values of all weights of the deep neural network. Denote also the gradient of the differentiable approximation $\tilde{\ell}$ of the loss $\ell$ in (5) at time $t$ as $g^{(t)} = \nabla \tilde{\ell}(\mathbf{Z}', \mathbf{Z}'', \mathbf{U}'')$.

With this notation in place, one may say the stochastic gradient descent in Algorithm 1 aims to minimize the expectation of $\tilde{\ell}(\mathbf{Z}', \mathbf{Z}'', \mathbf{U}'') = (1-\lambda) \cdot \ell_a(\mathbf{Z}') + \lambda \cdot \tilde{\ell}_u(\mathbf{Z}'', \mathbf{U}'')$ conditional on the data $(X_i, Y_i)_{i \in [n]}$. Note that $\tilde{\ell}$ is a deterministic function of $\mathbf{Z}', \mathbf{Z}'', \mathbf{U}''$ and it is differentiable in $\theta^{(t)}$. Define $J^{(t)}$ as the expected value of $\tilde{\ell}(\mathbf{Z}', \mathbf{Z}'', \mathbf{U}'')$ conditional on $\zeta^{(t)}$:

$$J^{(t)} = \mathbb{E}\left[ \tilde{\ell}(\mathbf{Z}', \mathbf{Z}'', \mathbf{U}'') \mid \zeta^{(t)} \right],$$

where the expectation is taken over the independent uniform random vector $\mathbf{U}''$, and let $\nabla J^{(t)}$ indicate its gradient with respect to $\theta^{(t)}$. In practice, at each step Algorithm 1 updates $\theta^{(t)}$ in the direction of an unbiased estimate $g^{(t)}$ of $\nabla J^{(t)}$, based on an independent random realization of $\mathbf{U}''$:

$$g^{(t)} = \nabla \tilde{\ell}(\mathbf{Z}', \mathbf{Z}'', \mathbf{U}''). \tag{A5}$$

This setup makes it possible to prove Algorithm 1 will tend to approach a regime of small $\nabla J^{(t)}$ after sufficiently many gradient updates, under suitable regularity conditions. This analysis is inspired by [3] and [10], as well as by the approach taken for an analogous convergence result in [9].

**Proposition A1.** *Assume there exists a finite Lipschitz constant $L > 0$ such that, for all parameter configurations $\theta'$, $\theta''$ and all possible values of the data batches $\mathbf{Z}'$, $\mathbf{Z}''$,*

$$\|\nabla \mathbb{E}\left[ \tilde{\ell}(\mathbf{Z}', \mathbf{Z}'', \mathbf{U}'') \mid \mathbf{Z}', \mathbf{Z}'', \theta' \right] - \nabla \mathbb{E}\left[ \tilde{\ell}(\mathbf{Z}', \mathbf{Z}'', \mathbf{U}'') \mid \mathbf{Z}', \mathbf{Z}'', \theta'' \right]\|_2 \leq L \|\theta' - \theta''\|_2. \tag{A6}$$

*Assume also the variance of $g^{(t)}$ in (A5) is uniformly bounded by some $\sigma^2 \in \mathbb{R}$:*

$$\mathbb{E}\left[\|g^{(t)} - J^{(t)}\|_2^2 \mid \zeta^{(t)}\right] \leq \sigma^2, \qquad \forall t \leq T.$$

*Then, for any initial state $\zeta^{(1)}$ of the learning algorithm and $\Delta = (2/L)\sup\left(J^{(1)}\right)$,*

$$\frac{1}{T}\sum_{t=1}^{T}\mathbb{E}\left[\|\nabla J^{(t)}\|_2^2 \parallel \mid \zeta^{(1)}\right] \leq \frac{L^2\Delta}{T} + \left(\mu_0 + \frac{\Delta}{\mu_0}\right)\frac{L\sigma}{\sqrt{T}}.$$

In plain words, Proposition A1 says the squared norm of the gradient of the loss function $\| \nabla J^{(t)} \|_2^2$ decreases on average at rate $T^{-1/2}$ as $T \to \infty$. This can be interpreted as a weak form of convergence, although it does not imply that $\theta^{(t)}$ will reach a global minimum or any other fixed point.

### A2.2  Mathematical proofs

*Proof of Proposition 1.* By definition of the conformity score function $W(\cdot)$ in (4), for any $\alpha \in (0, 1)$,

$$\mathbb{P}[W(X, Y, U; \pi) > 1 - \alpha \mid X = x] = \mathbb{P}[\min\{t \in [0, 1] : Y \in \mathcal{S}(x, U; \pi, t)\} > 1 - \alpha]$$
$$= \mathbb{P}[Y \notin \mathcal{S}(x, U; \pi, 1 - \alpha)] = \alpha.$$

Above, the second equality follows directly from the fact that $\mathcal{S}(x, U; \pi, t)$, defined in (A2), is by construction increasing in $t$, and therefore $Y \notin \mathcal{S}(x, U; \pi, 1 - \alpha)$ if and only if $\min\{t \in [0, 1] : Y \in \mathcal{S}(x, U; \pi, t)\} > 1 - \alpha$. $\qquad\square$

*Proof of Proposition 2.* The proof consists of showing that $\ell_{\mathrm{a}}$ and $\ell_{\mathrm{u}}$ are separately minimized by $\hat{\pi} = \pi$, although only approximately in the latter case. Without loss of generality, we will assume that $\mathcal{I}_1 = \mathcal{I}_2 = M = n/2$, to simplify the notation.

The first part of the proof is standard and proceeds as follows. Recall that the cross entropy loss $\ell_{\mathrm{a}}$ defined in (6) is

$$\ell_{\mathrm{a}} = -\frac{1}{M}\sum_{i \in \mathcal{I}_1}\sum_{c=1}^{K} Y_{i,c}\log\hat{\pi}_c\left(X_i\right),$$

where $Y_{i,c} = \mathbb{1}\left[Y_i = c\right]$ and $\hat{\pi}_c\left(X_i\right) = \hat{\mathbb{P}}\left[Y_i = c \mid X = x\right]$. Then, the expected value of $\ell_{\mathrm{a}}$, taken over the randomness in the data indexed by $\mathcal{I}_1$, is

$$\mathbb{E}\left[\ell_{\mathrm{a}}\right] = -\frac{1}{M}\sum_{i \in \mathcal{I}_1}\sum_{c=1}^{K}\mathbb{E}\left[Y_{i,c}\log\hat{\pi}_c\left(X_i\right)\right]$$
$$= -\frac{1}{M}\sum_{i \in \mathcal{I}_1}\sum_{c=1}^{K}\mathbb{E}\left[\mathbb{E}\left[Y_{i,c}\log\hat{\pi}_c\left(X_i\right) \mid X_i\right]\right]$$
$$= -\frac{1}{M}\sum_{i \in \mathcal{I}_1}\sum_{c=1}^{K}\mathbb{E}\left[\mathbb{E}\left[Y_{i,c} \mid X_i\right]\log\hat{\pi}_c\left(X_i\right)\right]$$
$$= -\frac{1}{M}\sum_{i \in \mathcal{I}_1}\mathbb{E}\left[\sum_{c=1}^{K}\pi_c\left(X_i\right)\log\hat{\pi}_c\left(X_i\right)\right]$$
$$= \frac{1}{M}\sum_{i \in \mathcal{I}_1}\mathbb{E}\left[H\left(\pi\left(X_i\right), \hat{\pi}\left(X_i\right)\right)\right]$$
$$= \mathbb{E}[H\left(\pi\left(X\right), \hat{\pi}\left(X\right)\right)],$$

where $H(\pi(X), \hat{\pi}(X))$ is the cross entropy of the conditional distribution $\hat{\pi}$ relative to the conditional distribution $\pi$ given $X$. Note that the last equality above simply follows from the underlying assumption that the data consist of i.i.d. observations from some unknown distribution $P_{XY}$.

This implies that $\mathbb{E}\left[\ell_a\right]$ is minimized by $\hat{\pi} = \pi$, because the cross entropy can be written as

$$H(\pi\left(X\right), \hat{\pi}\left(X\right)) = H(\pi\left(X\right)) + D_{KL}(\pi\left(X\right) \parallel \hat{\pi}\left(X\right)),$$

where $H(\pi\left(X\right))$ is the (constant) entropy of $\pi\left(X\right)$ and $D_{KL}$ denotes the Kullback–Leibler divergence, which is always non-negative and exactly equal to 0 if $\hat{\pi} = \pi$.

For the second part of the proof, recall that $\ell_u$ was defined in (7) as:

$$\ell_u = \mathcal{K}\left(\mathbf{W}\right),$$

where $\mathbf{W} \in \mathbb{R}^M$ are the conformity scores on $\mathcal{I}_2$, and $\mathcal{K}$ is the Kolmogorov–Smirnov distance from the uniform distribution:

$$\mathcal{K}\left(\mathbf{W}\right) = \sup_{w \in [0,1]} \left|\hat{F}_M\left(w\right) - w\right|.$$

Above, $\hat{F}_M$ is the empirical CDF of the scores. Note that $\mathcal{K}\left(\mathbf{W}\right)$ can be bound from above by

$$
\begin{aligned}
\mathcal{K}\left(\mathbf{W}\right) &= \sup_{w \in [0,1]} \left|\hat{F}_M\left(w\right) - w\right| \\
&= \sup_{w \in [0,1]} \left|\hat{F}_M\left(w\right) - F\left(w\right) + F\left(w\right) - w\right| \\
&\leq \sup_{w \in [0,1]} \left|\hat{F}_M\left(w\right) - F\left(w\right)\right| + \sup_{w \in [0,1]} \left|F\left(w\right) - w\right|,
\end{aligned}
$$

where $F$ is the true CDF of the conformity scores and the last step above is the triangle inequality. The first term on the right-hand-side above can be bound by the DKW inequality [2]:

$$\mathbb{P}\left[\sup_{w \in [0,1]} \left|\hat{F}_M\left(w\right) - F\left(w\right)\right| \geq \epsilon\right] \leq 2\exp^{-2M\epsilon^2}, \qquad \forall \epsilon > 0.$$

Therefore, we have obtained that

$$\ell_u \leq \sup_{w \in [0,1]} \left|F\left(w\right) - w\right| + \mathcal{O}_{\mathbb{P}}\left(\frac{1}{\sqrt{M}}\right).$$

According to Proposition 1, the conformity scores follow a uniform distribution if $\hat{\pi} = \pi$, and therefore in that case $\sup_{w \in [0,1]} \left|F\left(w\right) - w\right| = 0$. This completes the proof. $\qquad\square$

*Proof of Proposition A1.* The proof is essentially the same as that of Supplemental Theorem S1 in [9], but we nonetheless report here all details here completeness. First, note that a first-order Taylor expansion applied to the gradient update of Algorithm 1 yields

$$J^{(t+1)} \leq J^{(t)} + \langle \nabla J^{(t)}, \theta^{(t+1)} - \theta^{(t)} \rangle + \frac{L}{2}\mu^2 \|g^{(t)}\|_2^2.$$

Define $\delta^{(t)} = g^{(t)} - \nabla J^{(t)}$. Then, as $-\mu g^{(t)} = \theta^{(t+1)} - \theta^{(t)}$, the above inequality can be written as

$$
\begin{aligned}
J^{(t+1)} &\leq J^{(t)} - \mu\langle \nabla J^{(t)}, g_t \rangle + \frac{L}{2}\mu^2 \|g^{(t)}\|_2^2 \\
&= J^{(t)} - \mu\|\nabla J^{(t)}\|_2^2 - \mu\langle \nabla J^{(t)}, \delta_t \rangle + \frac{L}{2}\mu^2\left(\|\nabla J^{(t)}\|_2^2 + 2\langle \nabla J^{(t)}, \delta_t \rangle + \|\delta_t\|_2^2\right) \\
&= J^{(t)} - \left(\mu - \frac{L}{2}\mu^2\right)\|\nabla J^{(t)}\|_2^2 - \left(\mu - L\mu^2\right)\langle \nabla J^{(t)}, \delta_t \rangle + \frac{L}{2}\mu^2\|\delta^{(t)}\|_2^2.
\end{aligned}
$$

Summing the above inequalities over $t = 1, \ldots, T$, and noting that $J^{(t)} \geq 0$ for all $t$, we obtain:

$$
\begin{aligned}
\left(\mu - \frac{L}{2}\mu^2\right)\sum_{t=1}^{T}\|\nabla J^{(t)}\|_2^2 &\leq J^{(1)} - J^{(t+1)} - \left(\mu - L\mu^2\right)\sum_{t=1}^{T}\langle \nabla J^{(t)}, \delta^{(t)} \rangle + \frac{\mu^2 L}{2}\sum_{t=1}^{T}\|\delta^{(t)}\|_2^2 \\
&\leq J^{(1)} - \left(\mu - L\mu^2\right)\sum_{t=1}^{T}\langle \nabla J^{(t)}, \delta^{(t)} \rangle + \frac{\mu^2 L}{2}\sum_{t=1}^{T}\|\delta^{(t)}\|_2^2.
\end{aligned}
$$
(A7)

Recall that the estimated gradients $g^{(t)}$ are unbiased, i.e. $\mathbb{E}\big[g^{(t)} \mid \zeta^{(t)}\big] = \nabla J^{(t)}$. Therefore, taking the conditional expectation of both sides of (A7) given $\zeta^{(1)}$ leads to

$$
\begin{aligned}
\mathbb{E}\Big[\langle \nabla J^{(t)}, \delta^{(t)}\rangle \mid \zeta^{(1)}\Big] &= \mathbb{E}\Big[\mathbb{E}\Big[\langle \nabla J^{(t)}, \delta^{(t)}\rangle \mid \zeta^{(1)}, \zeta^{(t)}\Big] \mid \zeta^{(1)}\Big] \\
&= \mathbb{E}\Big[\mathbb{E}\Big[\langle \nabla J^{(t)}, \delta^{(t)}\rangle \mid \zeta^{(t)}\Big] \mid \zeta^{(1)}\Big] \\
&= \mathbb{E}\Big[\langle \nabla J^{(t)}, \mathbb{E}\Big[\delta^{(t)} \mid \zeta^{(t)}\Big]\rangle \mid \zeta^{(1)}\Big] \\
&= \mathbb{E}\Big[\langle \nabla J^{(t)}, 0\rangle \mid \zeta^{(1)}\Big] = 0.
\end{aligned}
$$

Replacing this result into (A7), and leveraging the assumption that $\mathbb{E}\big[\|\delta^{(t)}\|_2^2 \mid \zeta^{(t)}\big] \leq \sigma^2$, leads to

$$
\left(\mu - \frac{L}{2}\mu^2\right) \sum_{t=1}^{T} \mathbb{E}\Big[\|\nabla J^{(t)}\|_2^2 \mid \zeta^{(1)}\Big] \leq J^{(1)} + \frac{\mu^2 L}{2} T \sigma^2.
$$

Then, multiplying both sides above by $2/[LT(2\mu - L\mu^2)]$ results in

$$
\frac{1}{TL} \sum_{t=1}^{T} \mathbb{E}\Big[\|\nabla J^{(t)}\|_2^2 \mid \zeta^{(1)}\Big] \leq \frac{2J^{(1)}}{TL(2\mu - L\mu^2)} + \frac{\sigma^2 \mu}{2 - L\mu} \leq \frac{\Delta}{T(2\mu - L\mu^2)} + \frac{\sigma^2 \mu}{2 - L\mu}.
$$

Finally, choosing $\mu = \min\left\{\frac{1}{L}, \frac{\mu_0}{\sigma\sqrt{T}}\right\}$ for some $\mu_0 > 0$ gives the desired result:

$$
\frac{1}{T} \sum_{t=1}^{T} \mathbb{E}\Big[\| \nabla J^{(t)} \|_2^2\| \zeta^{(t)}\Big] \leq \frac{L^2 \Delta}{T} + \left(\mu_0 + \frac{\Delta}{\mu_0}\right)\frac{L\sigma}{\sqrt{T}}.
$$

$\square$

## A3 Additional details and results from numerical experiments

### A3.1 Details about experiments with synthetic data

The conditional data-generating distribution of $Y$ given $X$ is given by:

$$
\mathbb{P}[Y \mid X] = \begin{cases}
\left(\frac{1}{K/2}, \frac{1}{K/2}, \frac{1}{K/2}, 0, 0, 0\right), & \text{if } X_1 \leq \delta, X_2 < 0.5, \\
\left(0, 0, 0, \frac{1}{K/2}, \frac{1}{K/2}, \frac{1}{K/2}\right), & \text{if } X_1 \leq \delta, X_2 \geq 0.5, \\
(1, 0, 0, 0, 0, 0), & \text{if } X_1 > \delta, 0 \leq X_3 \geq \frac{1}{K}, \\
(0, 1, 0, 0, 0, 0), & \text{if } X_1 > \delta, \frac{1}{K} \leq X_3 \geq \frac{2}{K}, \\
\vdots & \vdots \\
(0, 0, 0, 0, 0, 1), & \text{if } X_1 > \delta, \frac{K-1}{K} \leq X_3 \geq 1,
\end{cases} \tag{A8}
$$

Above, we set $\delta = 0.2$, so that 20% of the samples are impossible to classify with absolute confidence.

The model trained to predict $Y \mid X$ is a fully connected neural network implemented in PyTorch [7], with 5 layers of width 256, 256, 128 and 64 and ReLU activations; the conditional class probabilities $\hat{\pi}$ are output by a final softmax layer. For both our new method and the hybrid method, the batch size is 750 and the hyper-parameter $\lambda$ controlling the relative weights of the two components of the loss is 0.2. Our method (resp., the hybrid method) is applied using $5/6$ of the data to evaluate the cross entropy and $1/6$ for the conformity score (resp., conformal size) loss. The models minimizing the cross entropy and focal loss are trained via stochastic gradient descent for 3000 epochs with batch size 200 and initial learning rate 0.01, decreased by a factor 10 halfway through training; see below for details about early stopping. The hybrid loss model is trained via stochastic gradient descent for 4000 epochs with learning rate 0.01 decreased by a factor 10 halfway through training. The conformal uncertainty-aware model is trained via Adam [5] for 4000 epochs with learning rate 0.001 decreased by a factor 10 halfway through training.

### A3.2 Results with synthetic data

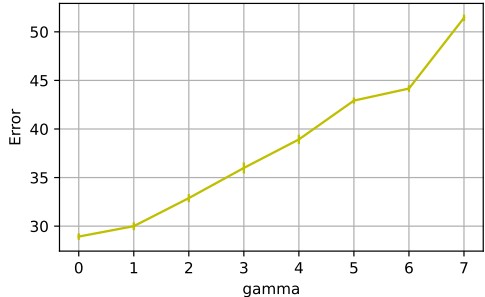

Figure A2: Best-guess misclassification rate on test data for a deep classifier trained to minimize the focal loss, with early stopping based on validation prediction accuracy. The results are averaged over 50 independent experiments and shown as a function of the loss function hyper-parameter $\gamma$. The size of the training sample is 2400. Other details are as in Figure 1.

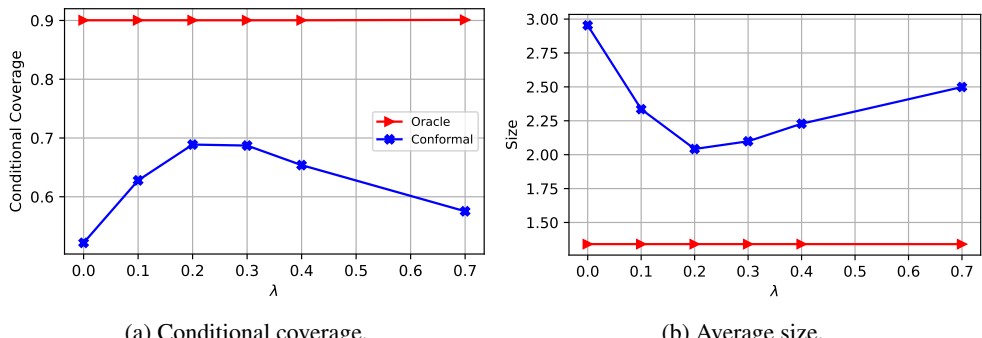

(a) Conditional coverage.

(b) Average size.

Figure A3: Conditional coverage (a) and average size (b) of conformal prediction sets obtained with our method in the numerical experiments with synthetic data of Figure 1, compared to the ideal results produced by a perfect oracle. Early stopping based on validation accuracy is employed. The results are shown as a function of the hyper-parameter $\lambda$ in (5). The number of training data points is 2400.

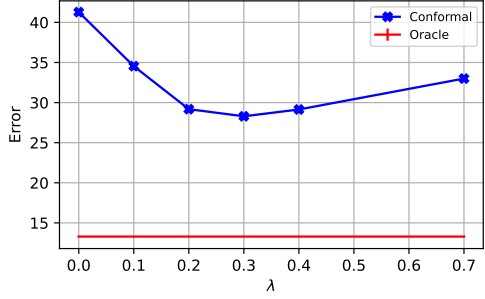

Figure A4: Average accuracy of best-guess predictions computed by models trained with our method in the numerical experiments with synthetic data of Figure 1, compared to the ideal results produced by a perfect oracle. Early stopping based on validation accuracy is employed. Other details are as in Figure A3.

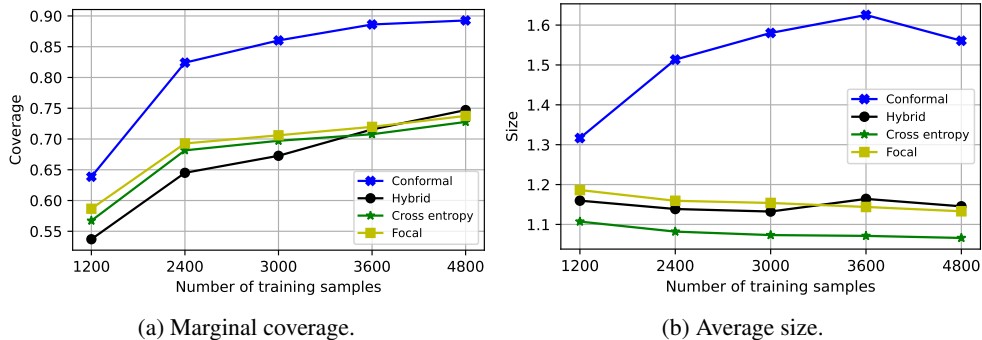

(a) Marginal coverage.

(b) Average size.

Figure A5: Marginal coverage (a) and average size (b) of conformal prediction sets obtained with different learning method in the numerical experiments of Figure 1, without post-training conformal calibration. The nominal target level is 90%. Other details are as in Figure 1.

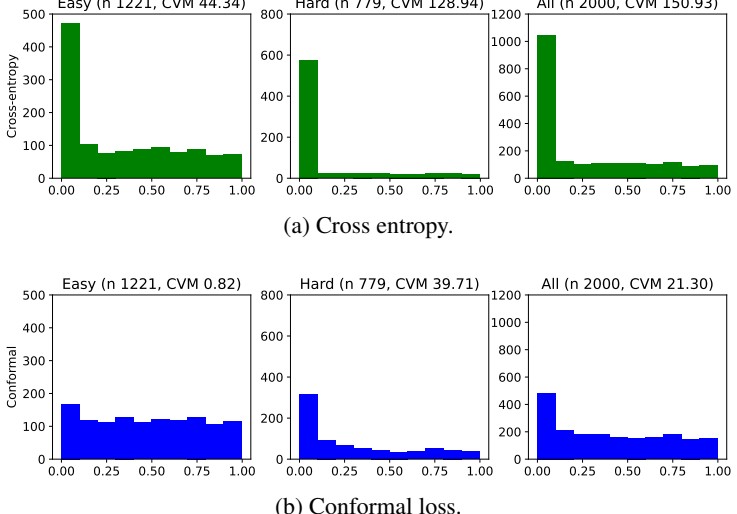

(a) Cross entropy.

(b) Conformal loss.

Figure A6: Histograms of conformity scores on synthetic test data obtained with deep classification models trained to minimize the cross entropy (a) or the proposed proposed conformal loss (b). The scores are shown separately based on the intrinsic difficulty of the test samples. Left: samples with $X_1 > 0.2$ (easy). Center: samples with $X_1 \leq 0.2$ (hard). Right: all samples. These test data contain approximately $40\%$ of hard samples. The numbers displayed after the "CVM" acronym are the values of the Cramer von Mises [1, 12] statistic for testing the uniformity in distribution of the conformity scores; smaller values indicate a more uniform distribution. Other details are as in Figure 1.

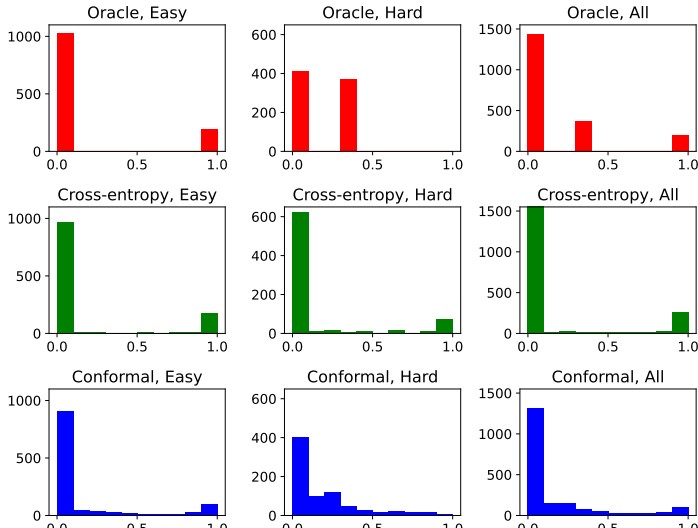

Figure A7: Histograms of conditional class probabilities ($\mathbb{P}[Y = 2 \mid X]$) computed on test synthetic data by the true oracle model (top) or estimated by a deep classification network minimizing the cross entropy (middle) or the proposed conformal loss (bottom). Other details are as in Figure A6.

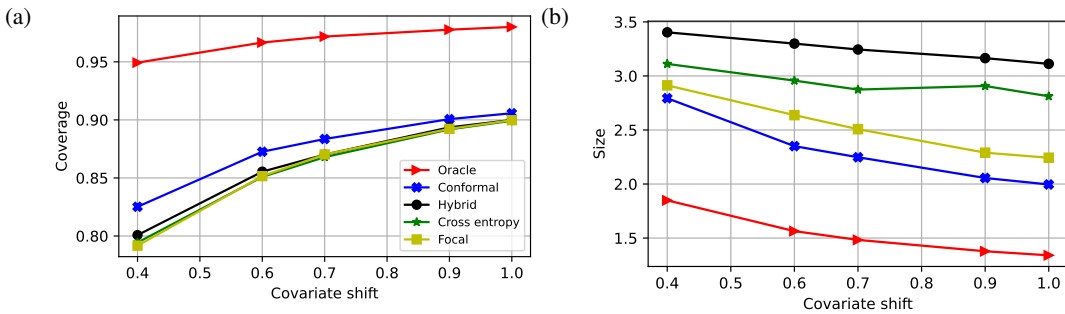

Figure A8: Performance of conformal prediction sets with 90% marginal coverage based on the ideal oracle model and a deep neural network trained with four alternative algorithms. The models are fully trained with 2400 samples, and the results are shown as a function of the amount of covariate shift. Covariate shift=1 corresponds to no covariate shift (20% of hard-to-classify samples), while lower values of covariate shift correspond to test sets with more numerous hard-to-classify samples compared to the training and calibration data sets. Other details are as in Figure 1.

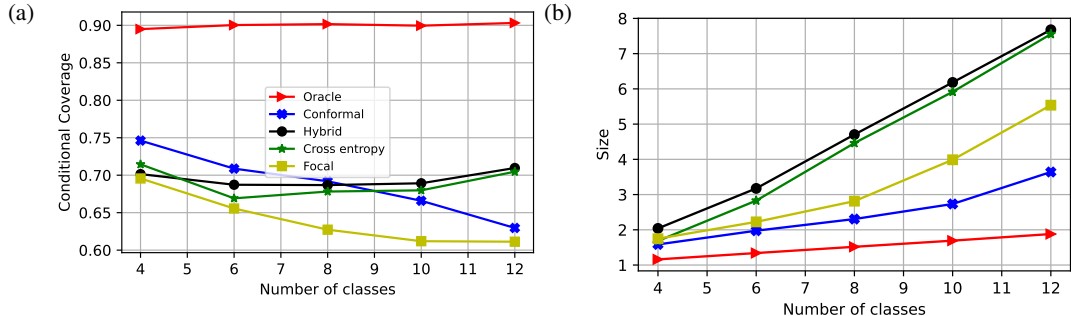

Figure A9: Performance of conformal prediction sets with 90% marginal coverage based on the ideal oracle model and a deep neural network trained with four alternative algorithms. The models are fully trained with 2400 samples, and the results are shown as a function of the number $K$ of possible classes. Other details are as in Figure 1.

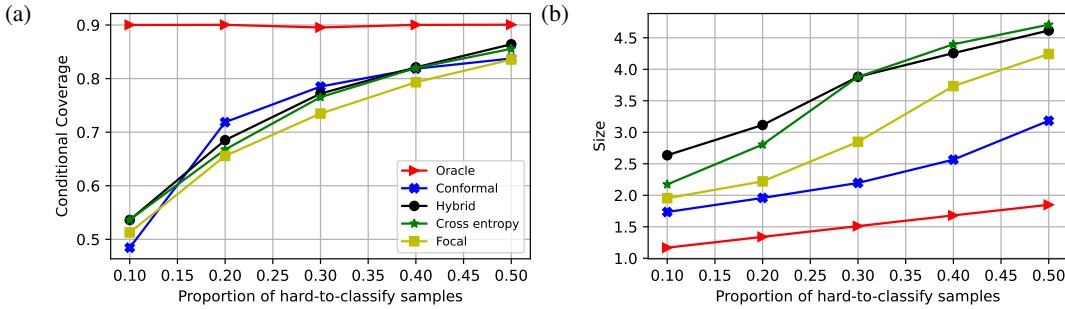

Figure A10: Performance of conformal prediction sets with 90% marginal coverage based on the ideal oracle model and a deep neural network trained with four alternative algorithms. The models are fully trained with 2400 samples, and the results are shown as a function of the proportion $\delta$ of hard-to-classify samples in the data. Other details are as in Figure 1.

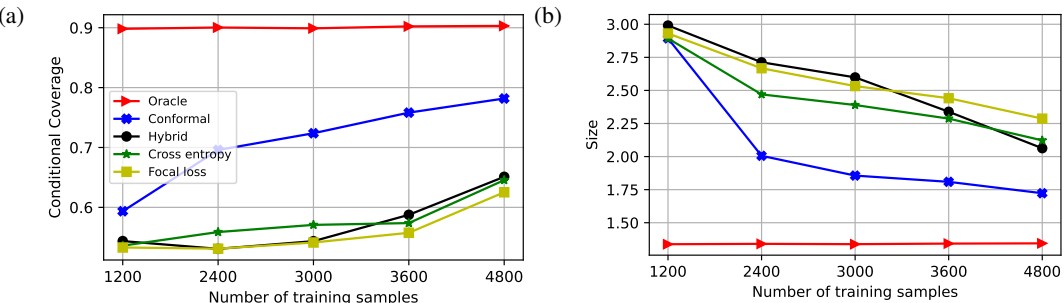

Figure A11: Performance of conformal prediction sets with 90% marginal coverage based on the oracle model and a deep neural network trained with different algorithms. The models are trained with 2400 samples and early stopping based on maximum accuracy on validation data. Other details are as in Figure 1.

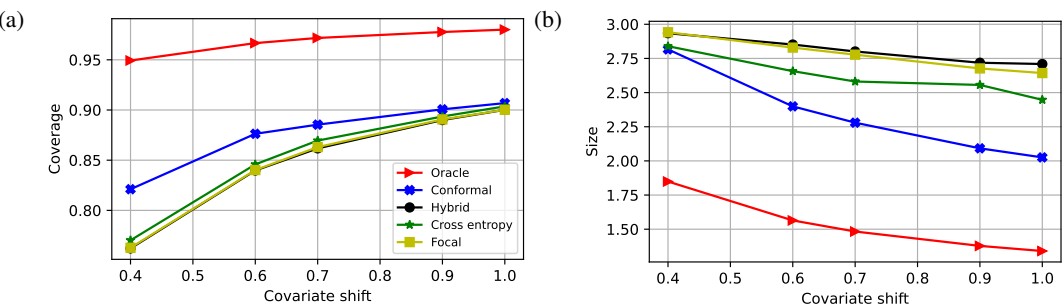

Figure A12: Performance of conformal prediction sets with 90% marginal coverage based on the ideal oracle model and a deep neural network trained with four alternative algorithms using 2400 training samples. The results are shown as a function of the amount of covariate shift. Early stopping based on validation accuracy is employed. Other details are as in Figure A8.

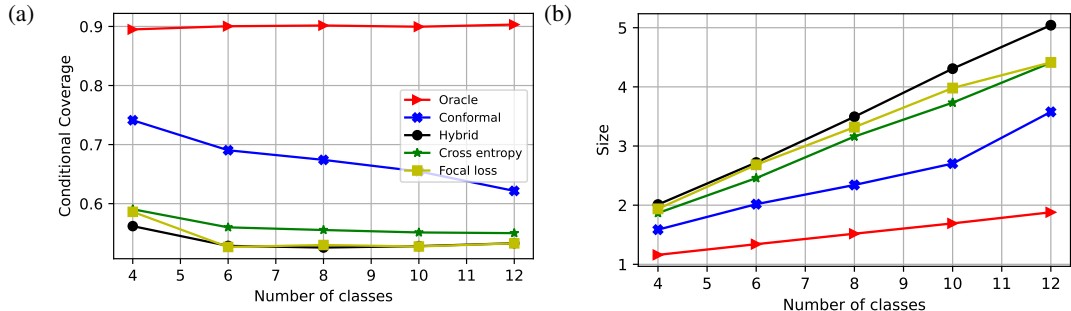

Figure A13: Performance of conformal prediction sets with 90% marginal coverage based on the oracle model and a deep neural network trained with different algorithms. The models are trained with 2400 samples and early stopping based on maximum accuracy on validation data. Other details are as in Figure A9.

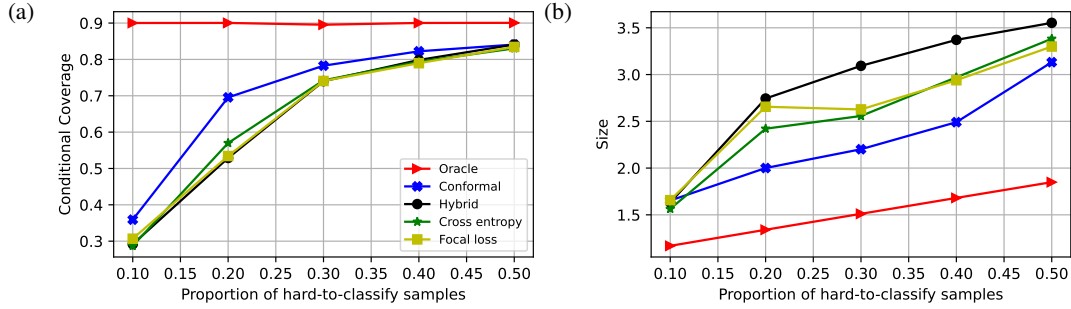

Figure A14: Performance of conformal prediction sets with 90% marginal coverage based on the oracle model and a deep neural network trained with different algorithms. The models are trained with 2400 samples and early stopping based on maximum accuracy on validation data. Other details are as in Figure A10.

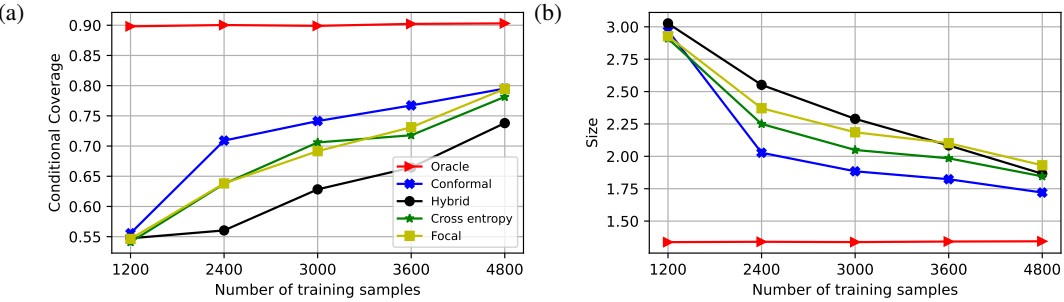

Figure A15: Performance of conformal prediction sets with 90% marginal coverage based on the oracle model and a deep neural network trained with different algorithms. The models are trained with 2400 samples and early stopping based on minimum validation loss. Other details are as in Figure 1.

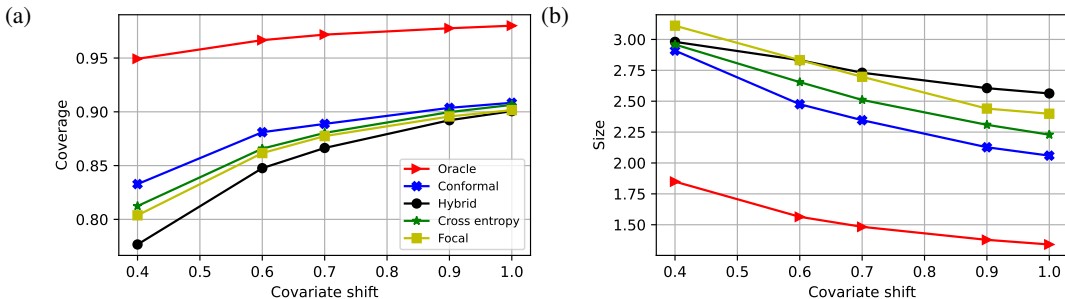

Figure A16: Performance of conformal prediction sets with 90% marginal coverage based on the ideal oracle model and a deep neural network trained with four alternative algorithms using 2400 training samples. The results are shown as a function of the amount of covariate shift. Early stopping based on minimum validation loss is employed. Other details are as in Figure A8.

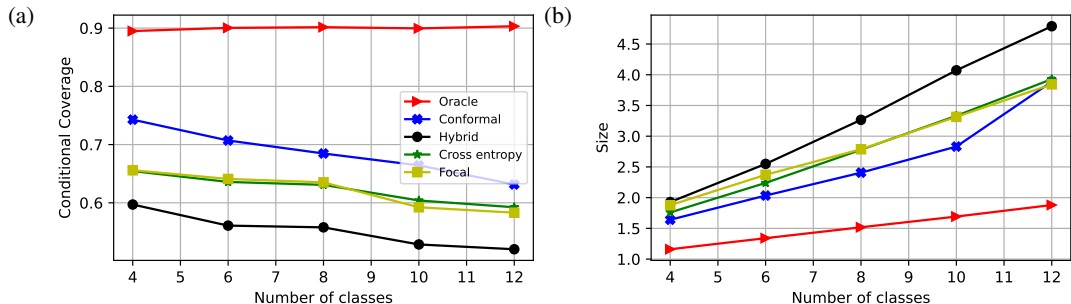

Figure A17: Performance of conformal prediction sets with 90% marginal coverage based on the oracle model and a deep neural network trained with different algorithms. The models are trained with 2400 samples and early stopping based on minimum validation loss. Other details are as in Figure A9.

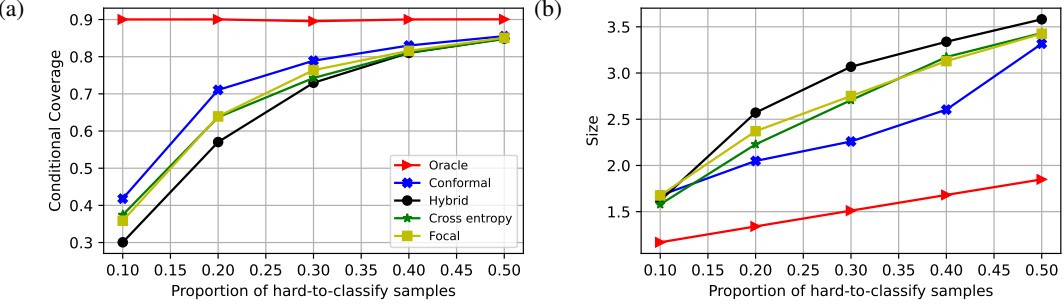

Figure A18: Performance of conformal prediction sets with 90% marginal coverage based on the oracle model and a deep neural network trained with different algorithms. The models are trained with 2400 samples and early stopping based on minimum validation loss. Other details are as in Figure A10.

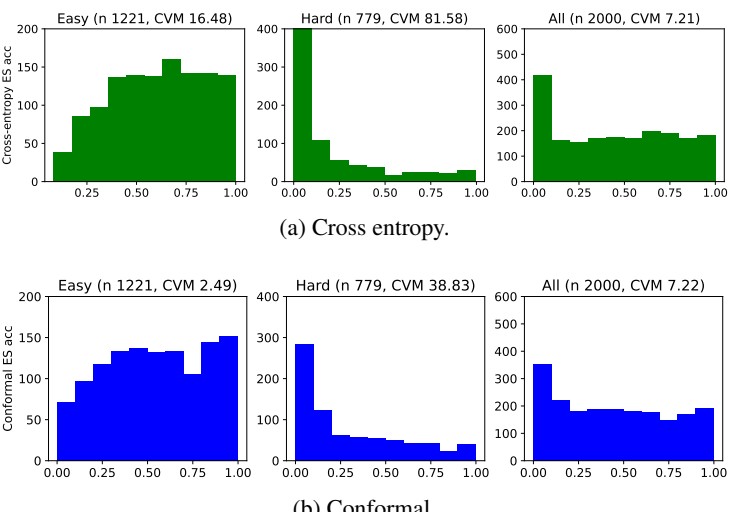

(a) Cross entropy.

(b) Conformal.

Figure A19: Histograms of conformity scores on synthetic test data obtained with deep classification models trained to minimize different losses. The models are trained with early stopping based on maximum validation accuracy. Other details are as in Figure A6.

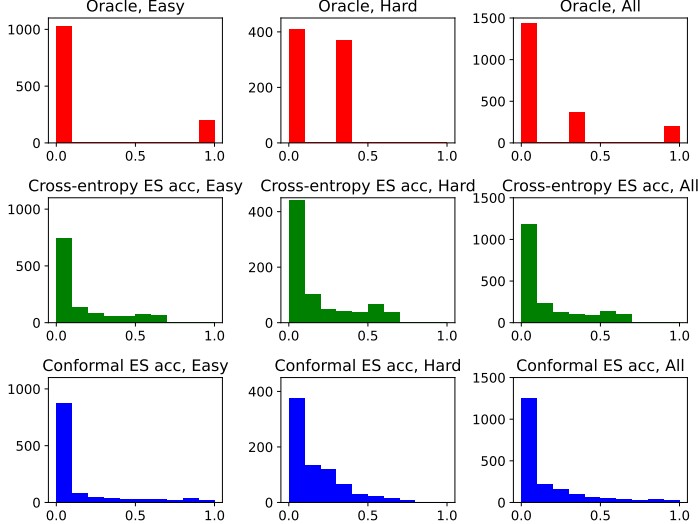

Figure A20: Histograms of conditional class probabilities ($\mathbb{P}[Y = 2 \mid X]$) computed on test synthetic data by the true oracle model or estimated by a deep classification network minimizing different loss functions. The models are trained with early stopping based on maximum validation accuracy. Other details are as in Figure A7.

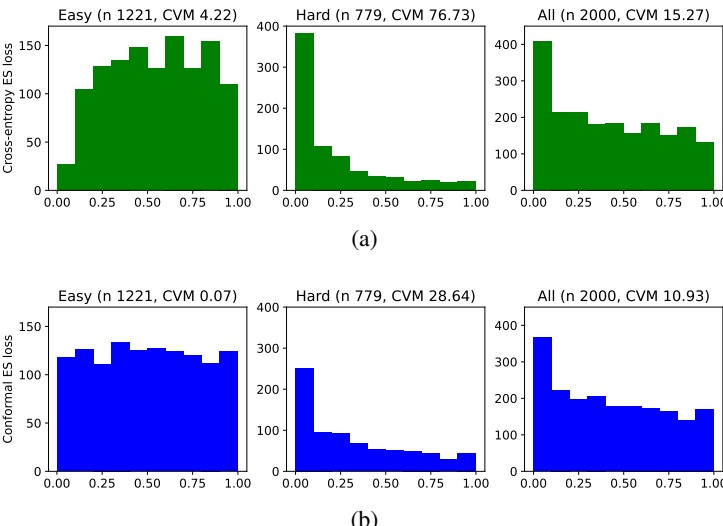

Figure A21: Histograms of conformity scores on synthetic test data obtained with deep classification models trained to minimize different losses. The models are trained with early stopping based on minimum validation loss. Other details are as in Figure A6.

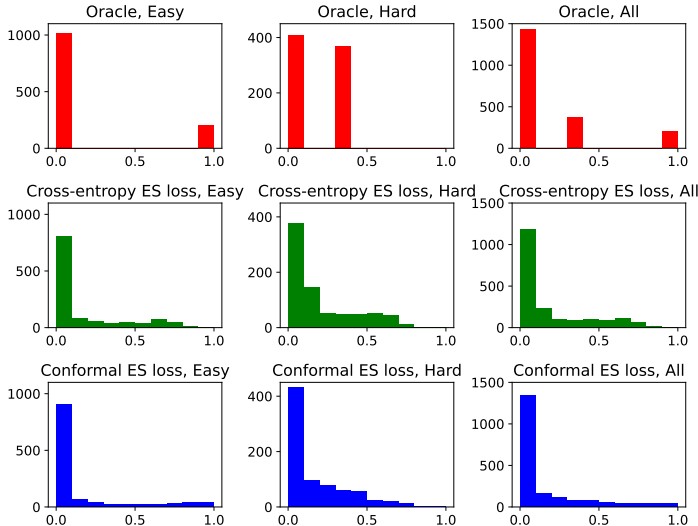

Figure A22: Histograms of conditional class probabilities ($\mathbb{P}[Y = 2 \mid X]$) computed on test synthetic data by the true oracle model or estimated by a deep classification network minimizing different loss functions. The models are trained with early stopping based on minimum validation loss. Other details are as in Figure A7.

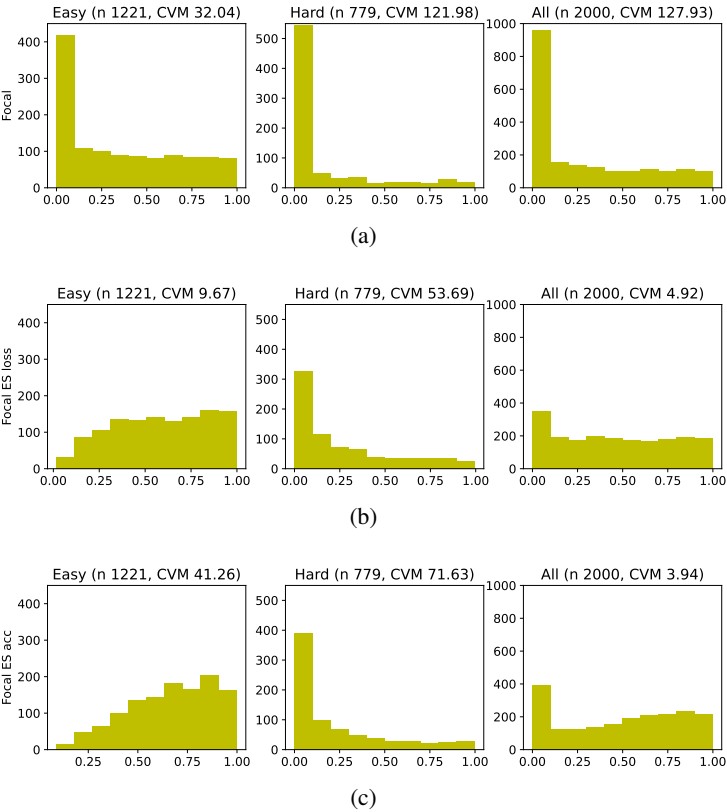

Figure A23: Histograms of conformity scores on synthetic test data obtained with deep classification models trained to minimize the focal loss. Top: fully trained model. Center: early stopping (ES) based on minimum validation loss (ES loss). Bottom: early stopping based on maximum validation accuracy (ES acc). Other details are as in Figure A6.

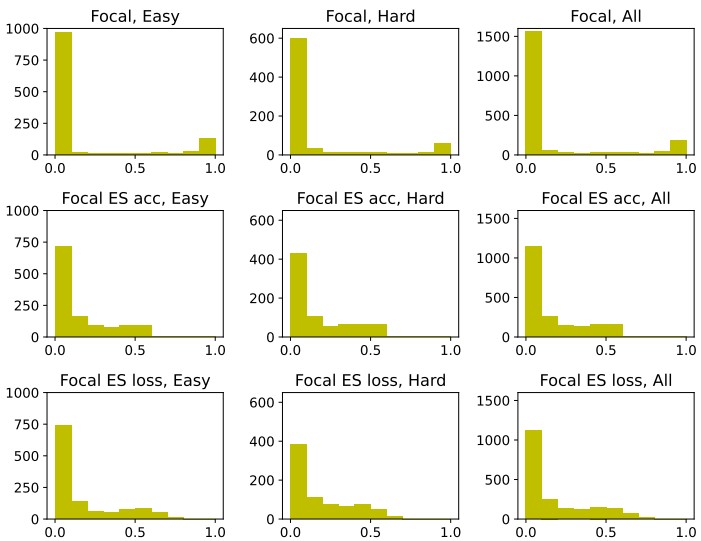

Figure A24: Histograms of conditional class probabilities ($\mathbb{P}[Y = 2 \mid X]$) computed on test synthetic data by a deep classification network minimizing the focal loss. Top: fully trained model. Center: early stopping based on minimum validation loss. Bottom: early stopping based on maximum validation accuracy. Other details are as in Figure A7.

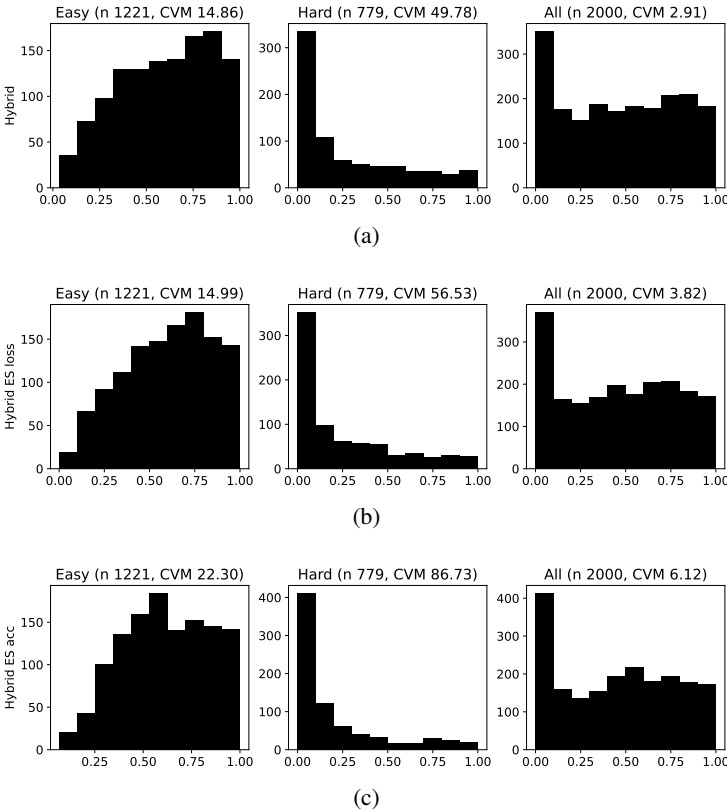

Figure A25: Histograms of conformity scores on synthetic test data obtained with deep classification models trained to minimize the hybrid loss. Top: fully trained model. Center: early stopping based on minimum validation loss. Bottom: early stopping based on maximum validation accuracy. Other details are as in Figure A6.

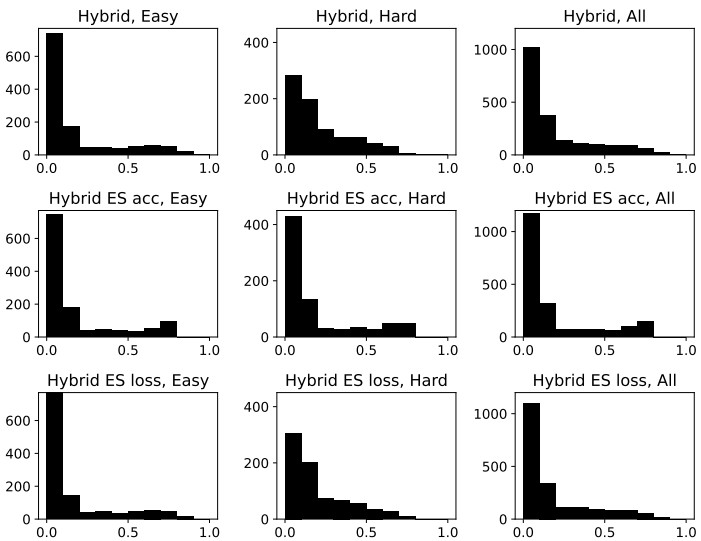

Figure A26: Histograms of conditional class probabilities ($\mathbb{P}[Y = 2 \mid X]$) computed on test synthetic data by a deep classification network minimizing the hybrid loss. Top: fully trained model. Center: early stopping based on minimum validation loss. Bottom: early stopping based on maximum validation accuracy. Other details are as in Figure A7.

### A3.3 Details about experiments with CIFAR-10 data

All images undergo standard normalization pre-processing. Then, the RandomErasing [13] method implemented in PyTorch [7] is applied with scale parameter 0.8 and ratio 0.3, and the fraction of corrupted images is varied within $\{0.01, 0.025, 0.05, 0.1, 0.2\}$. All models are trained on $\{3000, 10000, 16500, 27500, 45000\}$ samples. A ResNet-18 architecture for Cifar-10 is utilized, after modifying the original ResNet-18 network for ImageNet [4] as in `https://github.com/kuangliu/pytorch-cifar`: the kernel size of the first convolution layer is changed to 3x3, striding is removed, and the first maxpool operation is omitted. For the *conformal loss*, the hyper-parameter $\lambda$ controlling the relative weights of the uniform matching and cross entropy components is set to be 0.1. Then, the loss is approximately minimized using the Adam optimizer and a batch size of 768. The learning rate is initialized to 0.001 and then decreased by a factor 10 halfway through the training. The number of epochs is 2000 when training with 45000 samples; this is increased to 3200, 3500, 4000, and 5000 when training with 27500, 16500, 10000, and 3000 samples, respectively, as to reach a reasonably stationary state in each case. For the *cross entropy* loss, which is equivalent to the above loss with $\lambda = 0$, the batch size is 128 and the optimizer is stochastic gradient descent, which we have observed to work better in this case. The number of epochs is 1000 in the case of largest samples size, and 1500 with smaller samples. The learning rate is initialized to 0.1 and then decreased by a factor 10 halfway through the training. For the *focal loss*, we follow [6] and utilize stochastic gradient descent with batches of size 128. The main focal loss hyper-parameter is set equal to 3. The number of epochs is 1000 with 45000 samples, and similarly to the case of the cross entropy, it is increased to 1500 for smaller sample sizes. The learning rate is initialized to 0.1 and then decreased by a factor 10 halfway through the training. For the *hybrid loss*, the size hyper-parameter is set equal to 0.2. The optimizer is Adam with batch size 768. The number of epochs is 3000 with 45000 samples, and 3500, 3800, 5000, and 5000 when training with 27500, 16500, 10000, and 3000 samples, respectively. The learning rate is initialized to 0.001 and then decreased by a factor 10 halfway through the training. These hyper-parameters were tuned to separately maximize the performance of each method.

### A3.4 Results with CIFAR-10 data

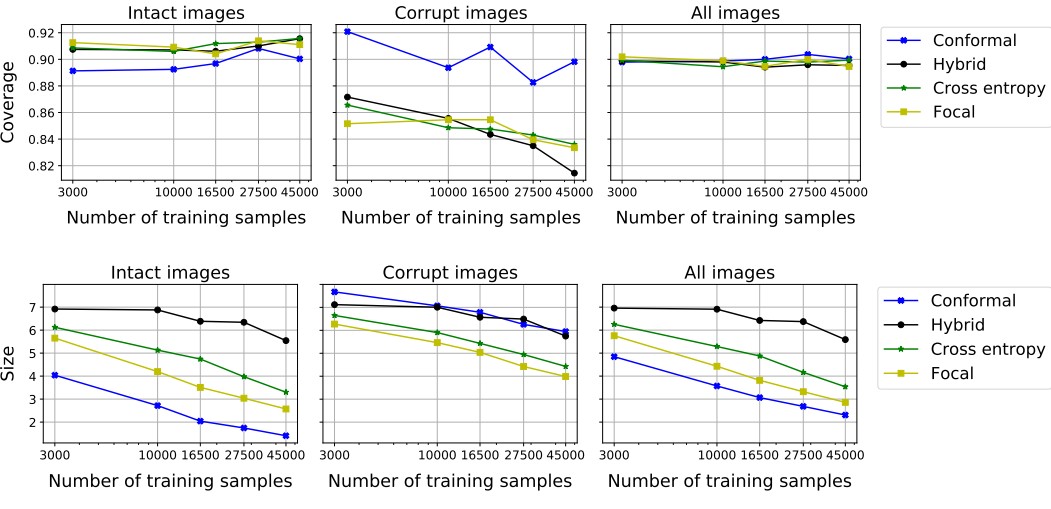

Figure A27: Performance of conformal prediction sets with 90% marginal coverage on CIFAR-10 data, based on convolutional neural networks trained with different loss functions. Top: conditional coverage based, separately for intact and corrupt images. Bottom: size of the prediction sets. The proportion of corrupted images in the training data is 20%

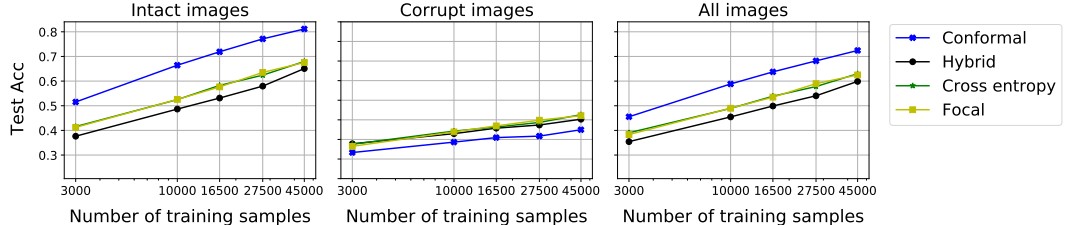

Figure A28: Test accuracy (Acc) of classification models trained on CIFAR-10 data, based on convolutional neural networks trained with different loss functions. Other details are as in Figure 1.

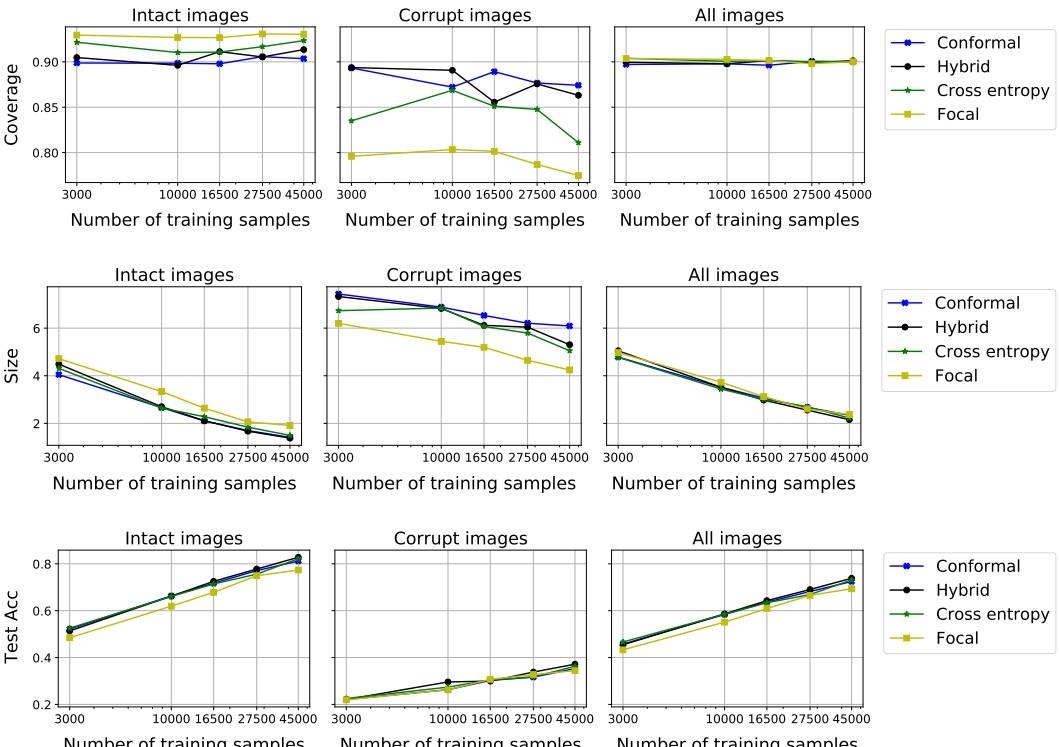

Figure A29: Performance of conformal prediction sets with 90% marginal coverage on CIFAR-10 data, based on convolutional neural networks trained with different loss functions. The models are trained with early stopping based on validation classification accuracy. Top: conditional coverage, separately for intact and corrupt images. Middle: size of the prediction sets. Bottom: test accuracy. Other details are as in Figure A27.

| | Coverage intact/corrupted | | | Size intact/corrupted | | | Accuracy intact/corrupted | | |
|---|---|---|---|---|---|---|---|---|---|
| | Full | ES (acc) | ES (loss) | Full | ES (acc) | ES (loss) | Full | ES (acc) | ES (loss) |
| Conformal | 0.90/0.90 | 0.90/0.87 | 0.94/0.79 | 1.41/5.95 | 1.41/6.09 | 1.86/5.12 | 0.81/0.35 | 0.81/0.35 | 0.79/0.29 |
| Hybrid | 0.92/0.81 | 0.91/0.86 | 0.92/0.83 | 5.54/5.75 | 1.38/5.30 | 1.67/5.37 | 0.65/0.40 | 0.83/0.37 | 0.80/0.32 |
| Cross Entropy | 0.92/0.84 | 0.92/0.81 | 0.92/0.82 | 3.30/4.43 | 1.50/5.05 | 1.51/4.95 | 0.68/0.43 | 0.82/0.36 | 0.82/0.36 |
| Focal | 0.91/0.83 | 0.93/0.77 | 0.94/0.81 | 2.57/3.99 | 1.91/4.25 | 2.08/4.58 | 0.68/0.42 | 0.77/0.34 | 0.76/0.35 |

Table A1: Performance of conformal prediction sets with 90% marginal coverage on CIFAR-10 data, based on models trained on 45,000 data points. The models are either trained fully for many epochs, or trained with early stopping (ES) based on different criteria: highest classification accuracy (acc) or lowest loss (loss). Other details are as in Table 1.

|  | Coverage intact/corrupted | | | Size intact/corrupted | | | Accuracy intact/corrupted | | |
| --- | --- | --- | --- | --- | --- | --- | --- | --- | --- |
|  | Full | ES (acc) | ES (loss) | Full | ES (acc) | ES (loss) | Full | ES (acc) | ES (loss) |
| Conformal | 0.89/0.92 | 0.90/0.89 | 0.93/0.78 | 4.04/7.67 | 4.05/7.43 | 4.49/6.31 | 0.52/0.23 | 0.52/0.22 | 0.50/0.20 |
| Hybrid | 0.91/0.87 | 0.90/0.89 | 0.95/0.72 | 6.92/7.12 | 4.49/7.33 | 5.4/5.96 | 0.38/0.28 | 0.51/0.22 | 0.46/0.17 |
| Cross Entropy | 0.91/0.87 | 0.92/0.84 | 0.95/0.71 | 6.13/6.65 | 4.31/6.73 | 5.24/5.73 | 0.42/0.28 | 0.53/0.23 | 0.44/0.17 |
| Focal | 0.91/0.85 | 0.93/0.80 | 0.96/0.70 | 5.65/6.27 | 4.72/6.20 | 5.61/5.77 | 0.41/0.26 | 0.48/0.22 | 0.44/0.18 |

Table A2: Performance of conformal prediction sets with 90% marginal coverage on CIFAR-10 data, based on models trained on 3,000 data points. Other details are as in Table A1.

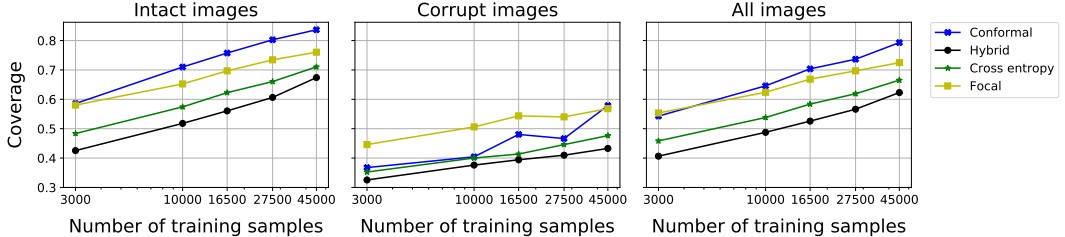

Figure A30: Marginal coverage of conformal prediction sets obtained with different learning method in the numerical experiments of Figure A27, without post-training conformal calibration. The target coverage level is 90%. Other details are as in Figure A27.

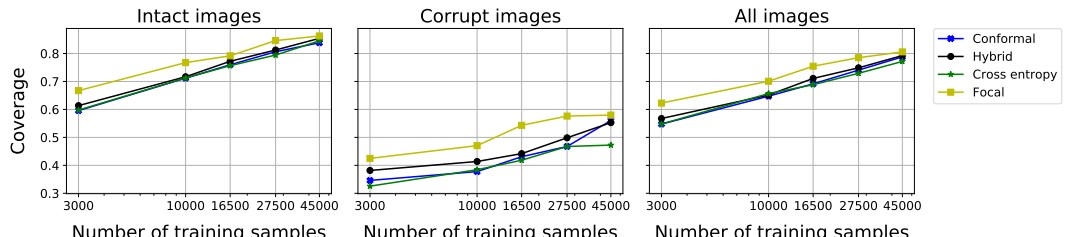

Figure A31: Marginal coverage of conformal prediction sets obtained with different learning method in the numerical experiments of Figure A29, without post-training conformal calibration. The target coverage level is 90%. Other details are as in Figure A29.

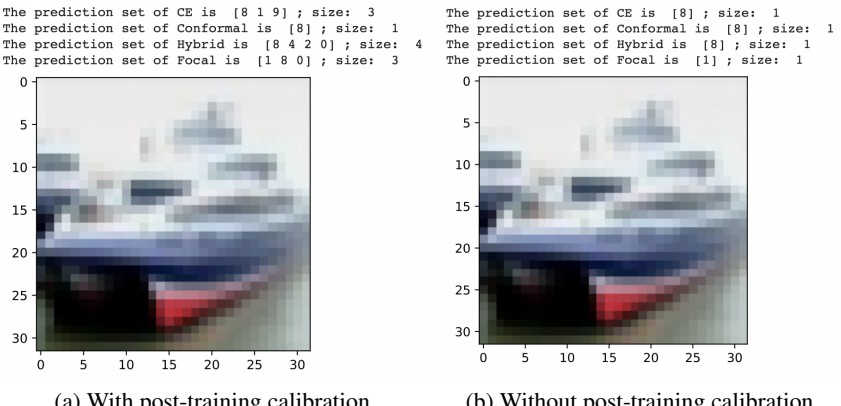

(a) With post-training calibration.          (b) Without post-training calibration.

Figure A32: Examples of prediction sets for an intact image, based on models fully trained with different methods, with and without post-training conformal calibration. These experiments are based on the CIFAR-10 data utilized in Table 1. The nominal coverage level is 0.9.

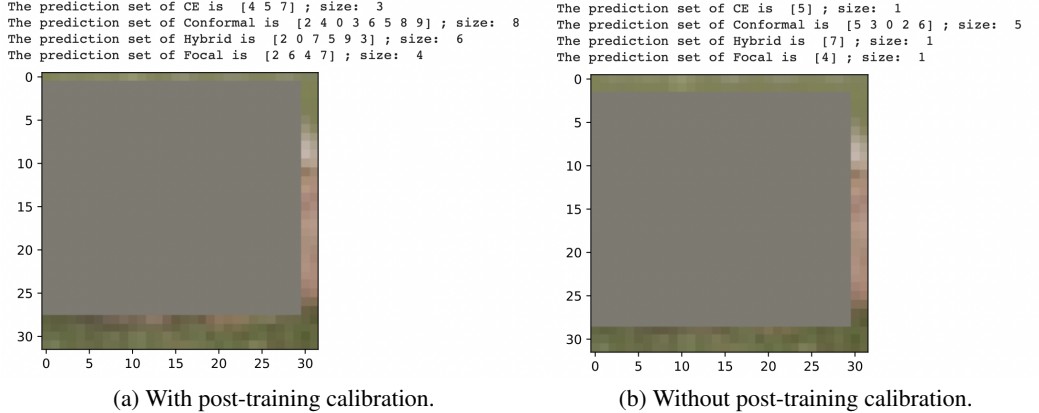

(a) With post-training calibration.       (b) Without post-training calibration.

Figure A33: Examples of prediction sets for a corrupt image, based on models fully trained with different methods, with and without post-training conformal calibration. Other details are as in Figure A32.

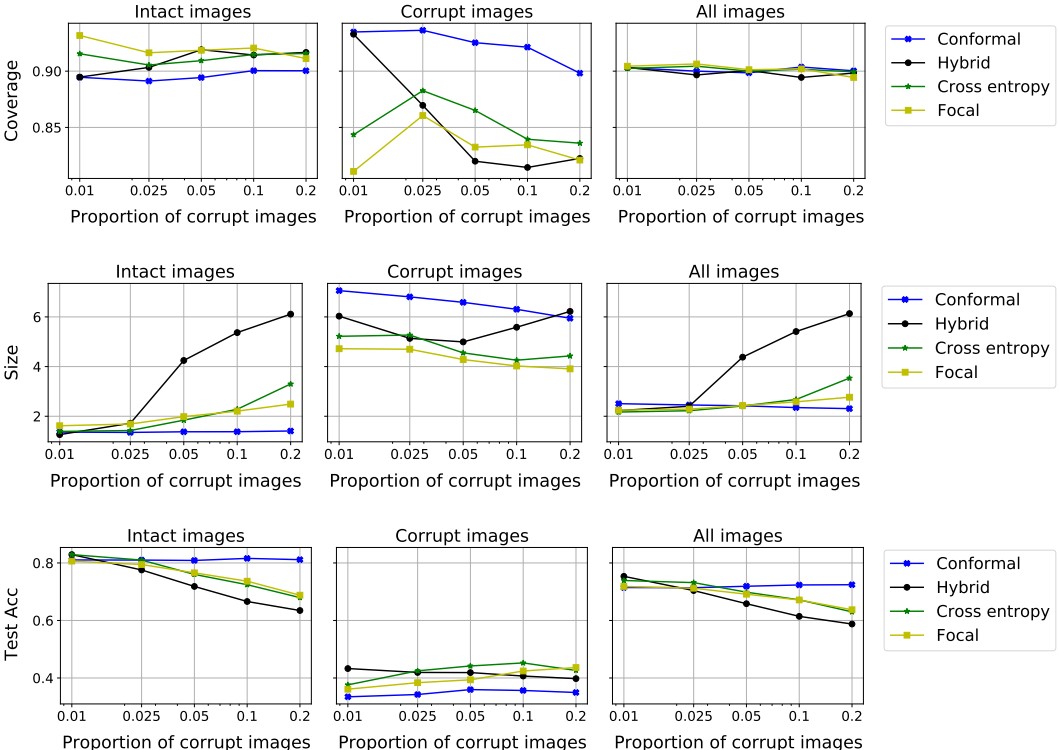

Figure A34: Performance of conformal prediction sets with 90% marginal coverage on CIFAR-10 data, based on convolutional neural networks trained with different loss functions. The results are shown as a function of the proportion of corrupt images in the training data. Top: conditional coverage, separately for intact and corrupt images. Middle: size of the prediction sets. Bottom: test accuracy. Other details are as in Figure A27.

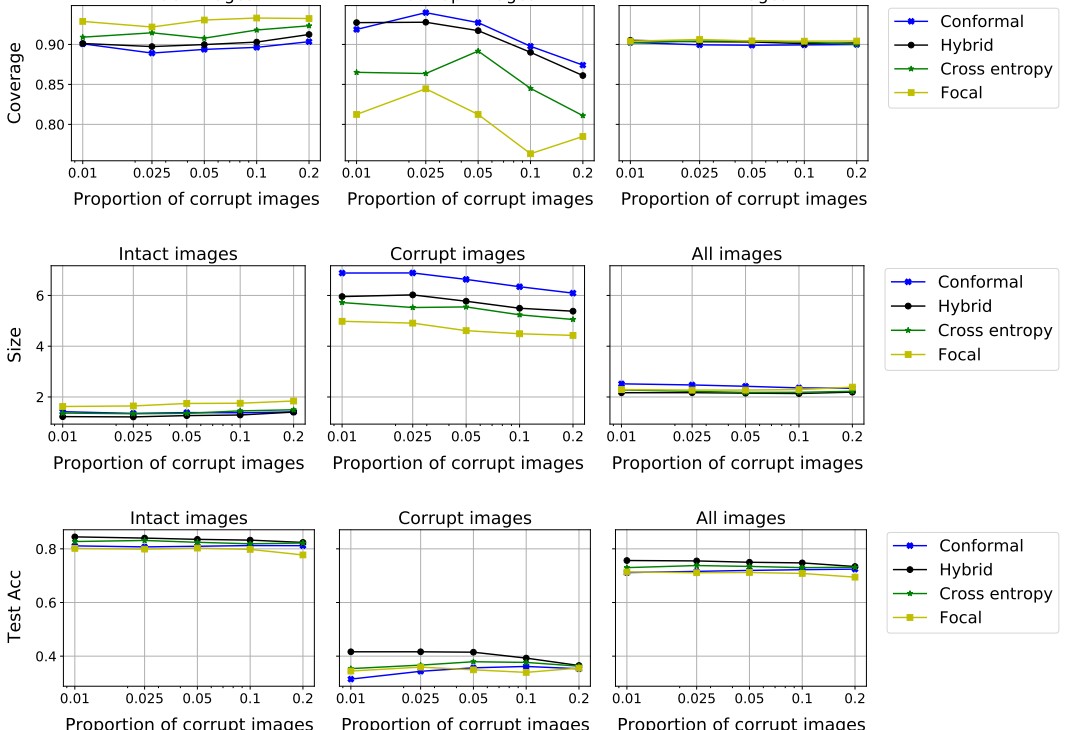

Figure A35: Performance of conformal prediction sets with 90% marginal coverage on CIFAR-10 data, based on convolutional neural networks trained with different loss functions. The models are trained with early stopping based on validation prediction accuracy. Other details are as in Figure A34.

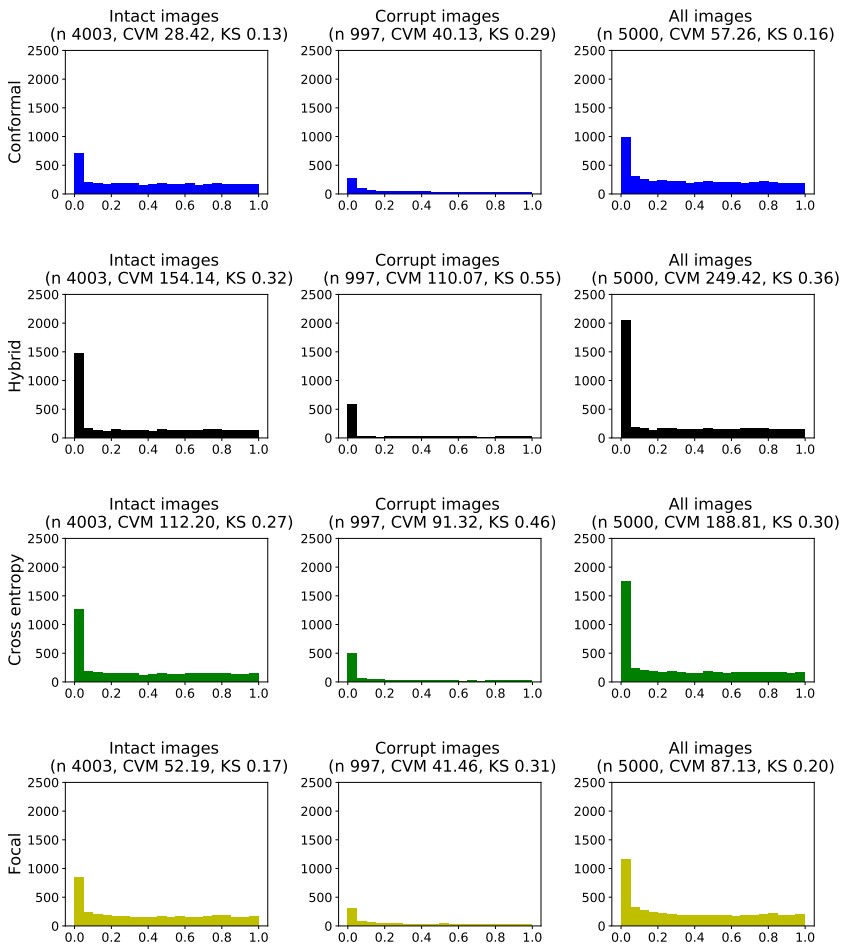

Figure A36: Histograms of conformity scores evaluated on test data and based on convolutional neural network models fully trained with different loss functions on 45000 images from CIFAR-10. The scores are reported separately for intact and corrupt images. The statistics in parenthesis at the top of each facet indicate the number of test samples, and the values of the Cramér-von Mises and Kolmogorov-Smirnov statistics for testing uniformity. Other details are as in Figure A27.

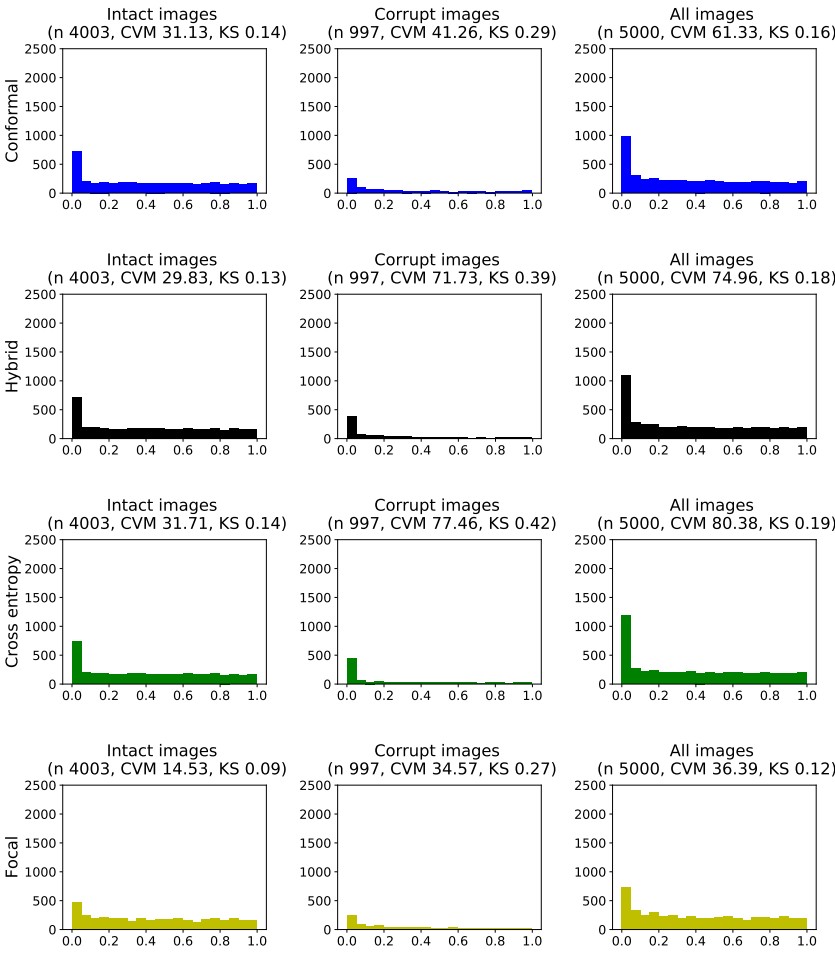

Figure A37: Histograms of conformity scores evaluated on test data and based on convolutional neural network models trained on 45000 images from CIFAR-10 with different loss functions and early stopping based on validation accuracy. Other details are as in Figure A36.

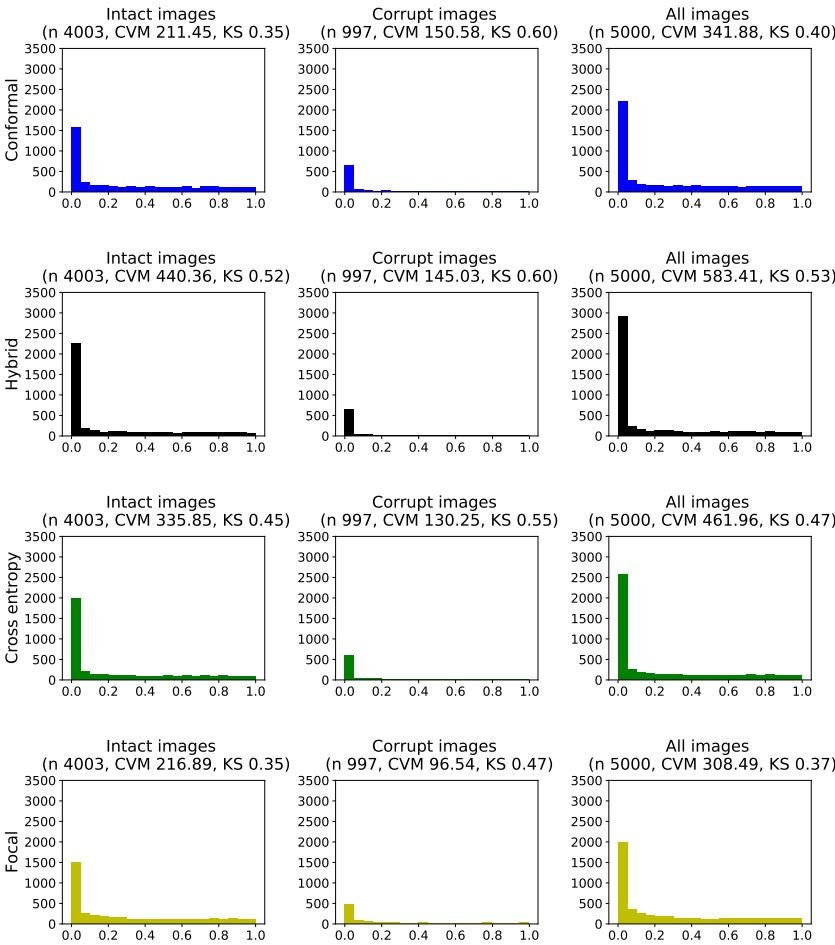

Figure A38: Histograms of conformity scores evaluated on test data and based on convolutional neural network models fully trained with different loss functions on 3000 images from CIFAR-10. Other details are as in Figure A36.

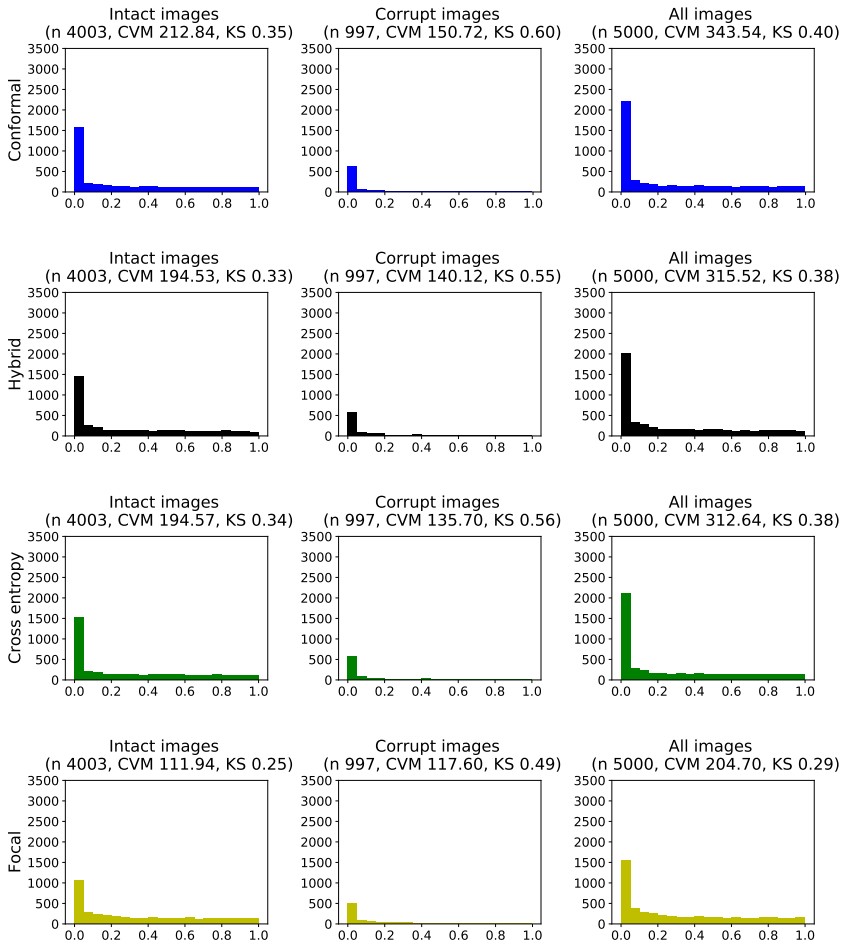

Figure A39: Histograms of conformity scores evaluated on test data and based on convolutional neural network models trained on 3000 images from CIFAR-10 with different loss functions and early stopping. Other details are as in Figure A37.

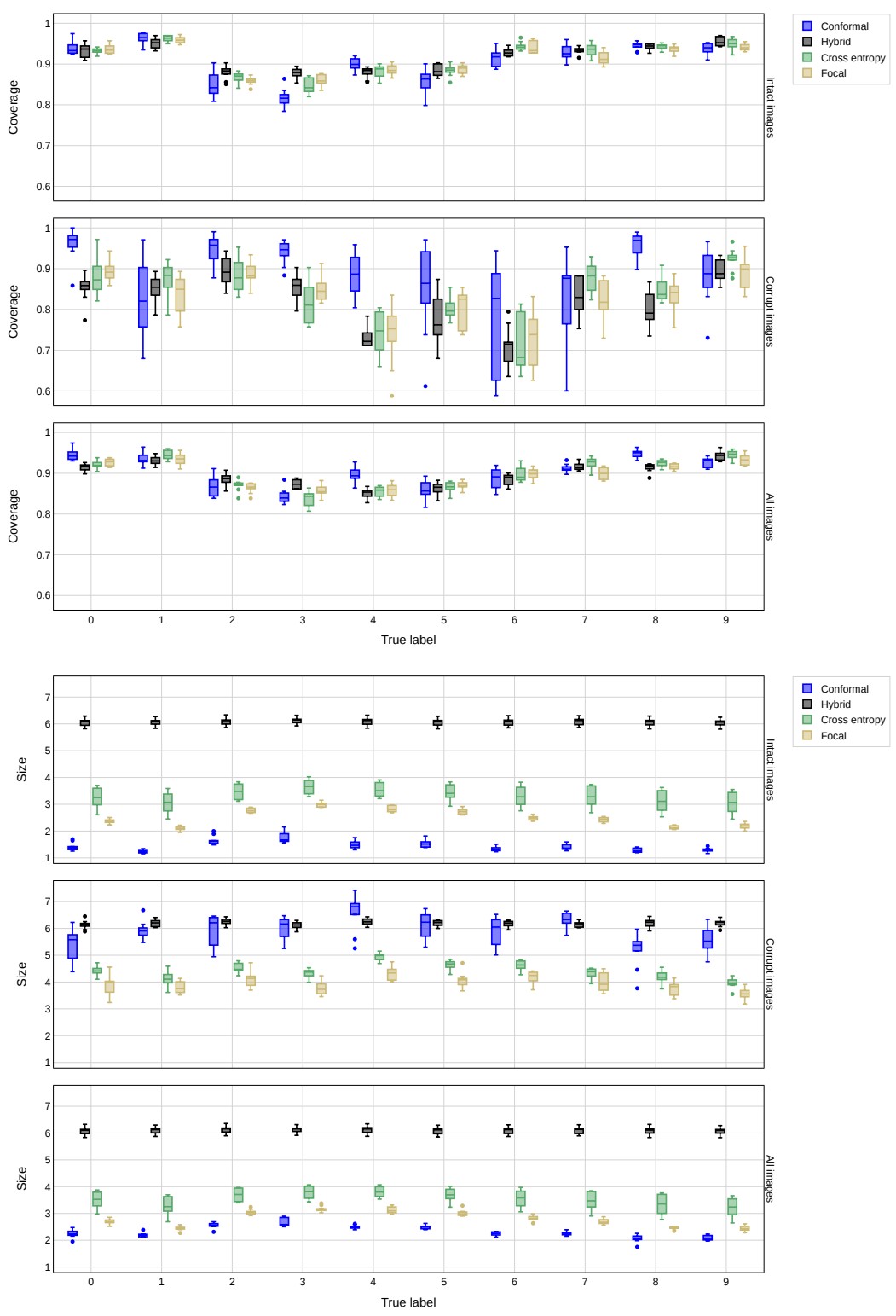

Figure A40: Coverage (top) and size (bottom) of conformal prediction sets with 90% marginal coverage on CIFAR-10 data, based on convolutional neural networks trained with different loss functions. The performance is reported separately for each true label. The models are fully trained on 45,000 images. Other details are as in Figure A27.

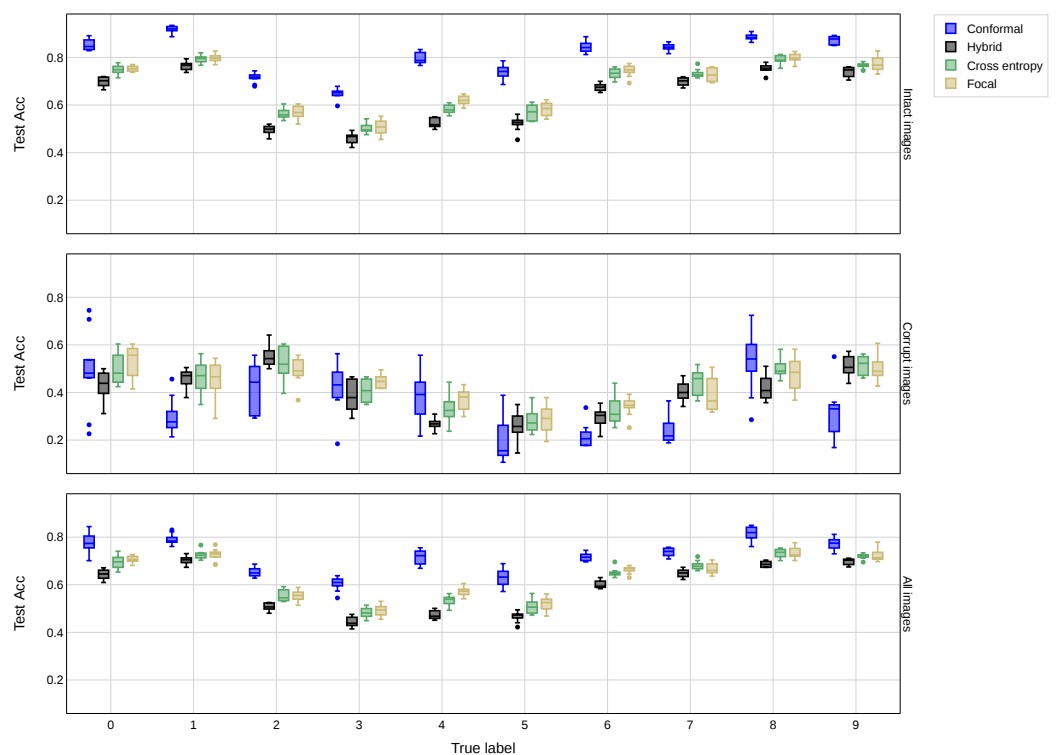

Figure A41: Test accuracy of convolutional neural networks trained with different loss functions. The performance is reported separately for each true label. The models are fully trained on 45,000 images. Other details are as in Figure A40.

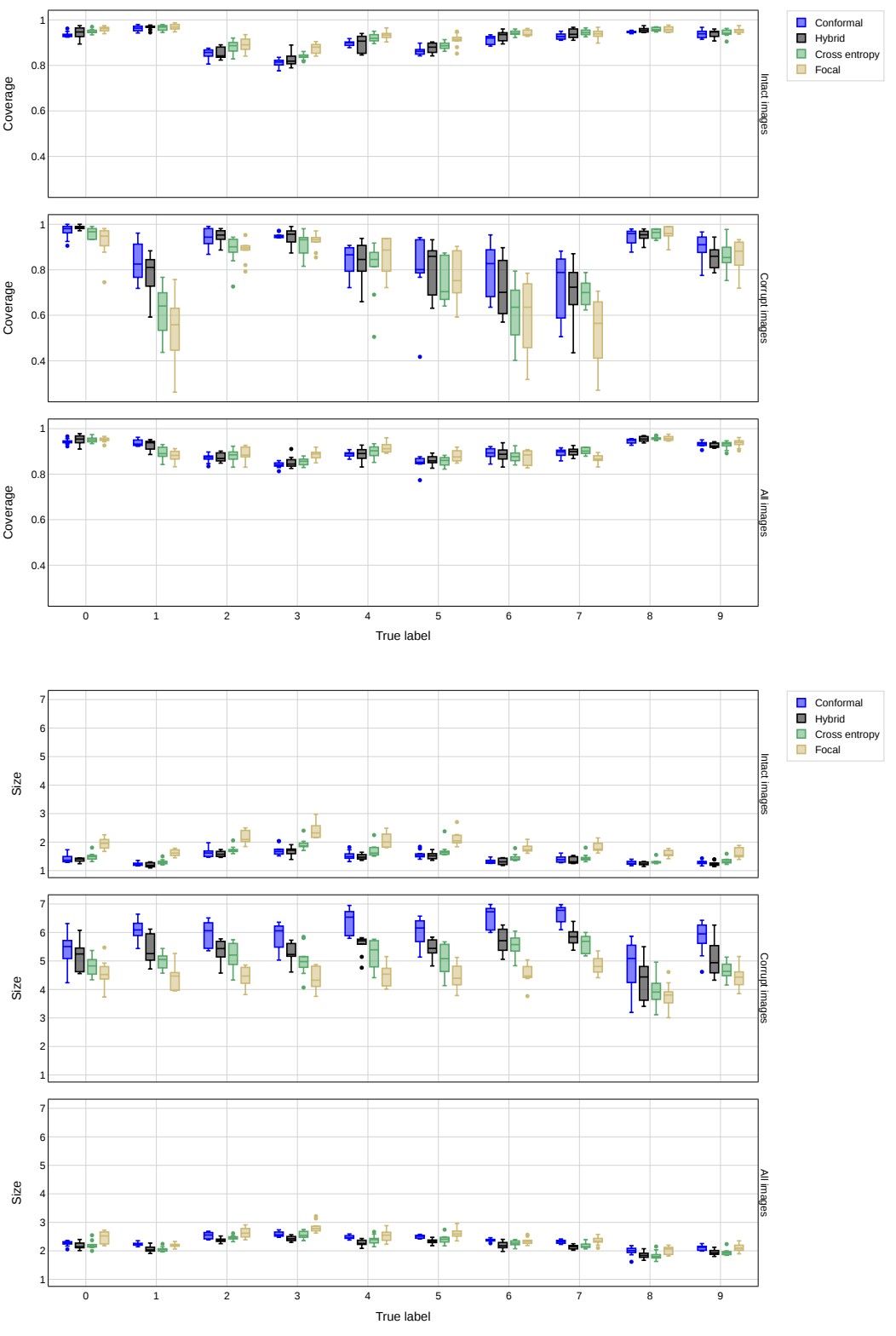

Figure A42: Coverage (top) and size (bottom) of conformal prediction sets with 90% marginal coverage on CIFAR-10 data, based on convolutional neural networks trained with different loss functions. The performance is reported separately for each true label. The models are trained on 45,000 images with early stopping based on validation accuracy. Other details are as in Figure A27.

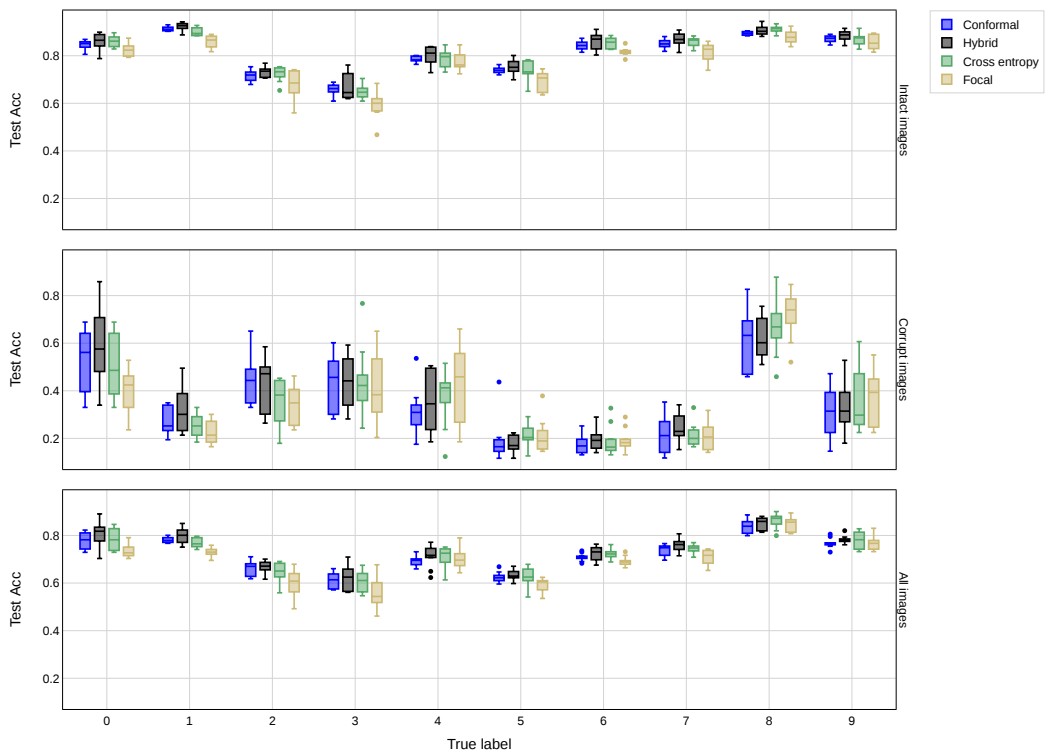

Figure A43: Test accuracy of convolutional neural networks trained with different loss functions. The performance is reported separately for each true label. The models are trained on 45,000 images with early stopping based on validation accuracy. Other details are as in Figure A42.

### A3.5 Details about experiments with credit card data

The models based on the conformal and hybrid losses are trained for 6000 and 4000 epochs, respectively, using the Adam optimizer with batch size of 2500. The initial learning rate of 0.0001 is decreased by a factor 10 halfway through training. The hyper-parameter $\lambda$ controlling the relative weight of the cross entropy component is set equal to 0.1 for both losses. The models minimizing the cross entropy and focal loss are trained via Adam for 3000 epochs with batch size 500 and learning rate 0.0001, also decreased by a factor 10 halfway through training. These hyper-parameters were tuned to separately maximize the performance of each method.

### A3.6 Results with credit card data

| | Coverage | | | | Size all labels/0/1 | | Classification error | | Distance from uniformity of conformity scores | |
|---|---|---|---|---|---|---|---|---|---|---|
| | Marginal | | Conditional | | | | | | | |
| | Full | E.S. | Full | E.S. | Full | E.S. | Full | E.S. | Full | E.S. |
| Conformal | 0.83 | 0.83 | 0.60 | 0.52 | 1.34/1.31/1.45 | 1.29/1.27/1.39 | 33.05 | 28.52 | 5.14/5.27/32.38 | 1.08/10.97/43.05 |
| Hybrid | 0.83 | 0.83 | 0.51 | 0.53 | 1.27/1.24/1.37 | 1.28/1.25/1.38 | 27.00 | 27.52 | 24.58/2.11/96.30 | 24.16/4.20/84.72 |
| Cross Entropy | 0.82 | 0.84 | 0.51 | 0.42 | 1.25/1.23/1.33 | 1.24/1.21/1.35 | 26.16 | 24.36 | 40.09/3.10/115.30 | 5.26/15.76/115.86 |
| Focal | 0.81 | 0.82 | 0.48 | 0.50 | 1.22/1.20/1.28 | 1.25/1.23/1.32 | 26.56 | 26.30 | 64.51/8.47/132.94 | 42.056/3.22/117.43 |

Table A3: Performance of conformal prediction sets with 80% marginal coverage on credit card default data, including also statistics measuring the distance from uniformity of the conformity scores evaluated on test data. Other details are as in Figure A3.

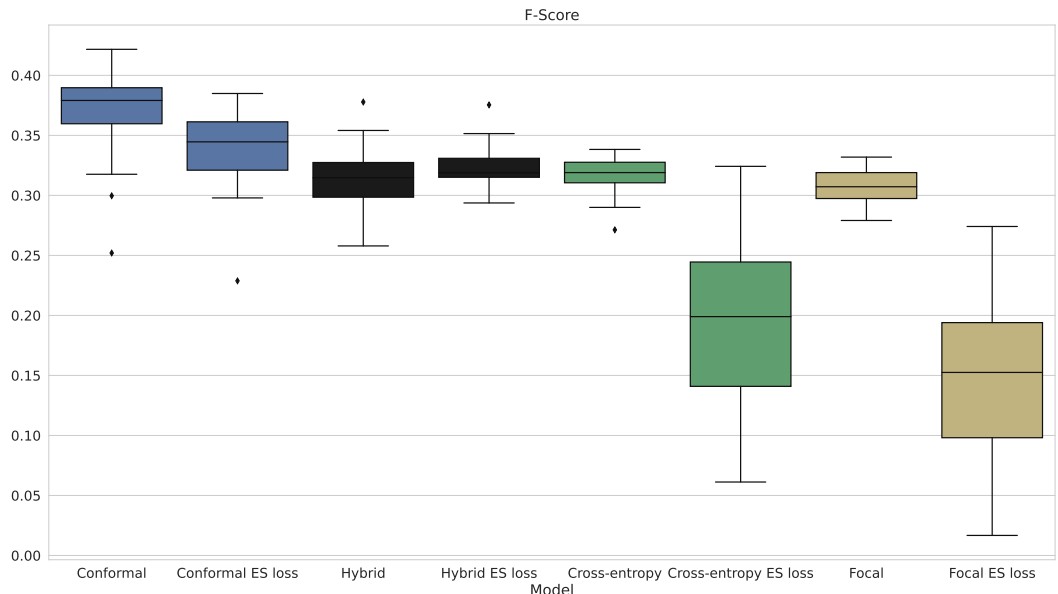

Figure A44: Empirical distribution over 20 independent experiments of the classification F-scores obtained with models trained with different methods on the credit card data set of Table 2.

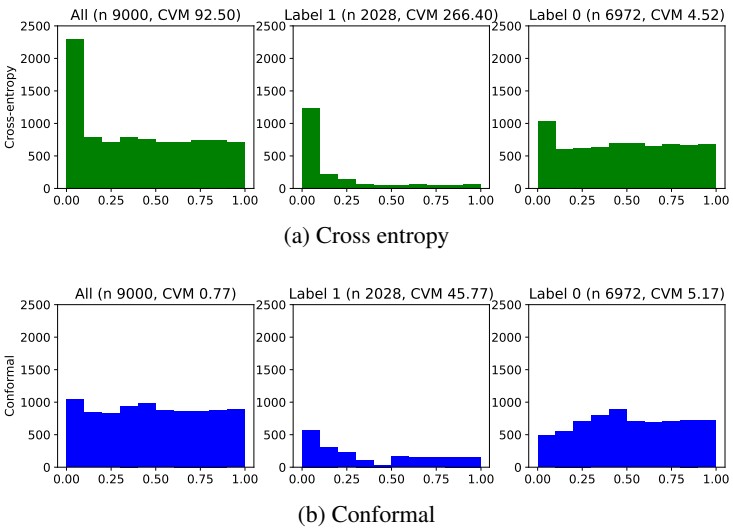

Figure A45: Histograms of conformity scores computed with models trained with different algorithms in simulations with credit card default data. The scores are evaluated on test data separately for samples with label 0 and 1. (a): Models trained with the cross entropy loss. (b): Models trained with the proposed conformal loss.