# OpenReview forum: "Training Uncertainty-Aware Classifiers with Conformalized Deep Learning"
_NeurIPS.cc/2022/Conference — NeurIPS 2022 Accept_

### Official Review · Reviewer_m7U1 · 2022-07-04

**Rating:** 5
**Confidence:** 2
**Soundness:** 3 good
**Presentation:** 2 fair
**Contribution:** 2 fair

**Summary:**

The paper combines ideas from conformal prediction and differentiable ranking/sorting to develop a loss function that improves the predictive uncertainty of a classifier by training it to make predictions with a desired conformity score. The paper reports competitive coverage and conformal prediction set sizes on synthetic data, CIFAR10 and a credit card default dataset.

**Questions:**

I'm overall not too confident in my review due to being unfamiliar with the related literature and commonly used experimental setups and results, but I'd appreciate the following questions being clarified:

* Are you indeed using a two-level procedure even with your conformal prediction loss? If yes, is it possible to use the loss as a one-level approach and if so, how does it perform? I had a look at the appendix and did not find anything there, but might have missed it.
* Do you have an explanation for where the need for early stopping arises? I can imagine that with an added regularizer this can happen, but the bad test accuracy for the cross-entropy baseline on CIFAR10 seems extremely strange to me, I don't think I have ever seen the test accuracy of a properly trained Resnet go down over time on that dataset. Is it due to the data corruption? That being said, it would be nice to have results trained on clean data as a baseline to see that the implementation is reasonable.
* How do different choices of $\lambda$ affect performance?

Other (minor) notes:
* I'm a bit uncomfortable with some statements in the introduction. In particular, neural nets trained with proper scoring rules should be calibrated in the infinite data limit. Intrinsically noisy data is not problematic, as this would be reflected in the training set and lead to less confident predictions. The problem in benchmarking on datasets such as CIFAR10 is the opposite, namely that they are highly curated and there is hardly any noise in the data, but once the underlying data distribution shifts the predictions become overconfident. I'm probably making some statements here myself that are not perfectly accurate either, but I'd encourage have another pass over the first paragraph of the introduction and ensure that all statements are technically precise and correct.
* Exchangeability of the data seems to be a core assumption, it would be nice to have an evaluation of the method under some shift of the test distribution on the real data, e.g. using different degrees of corruption as in (Ovadia et al., 2019. Can you trust your model's uncertainty? Evaluating predictive uncertainty under dataset shift. In NeurIPS).
* Table captions should go above the tables (unless the style instructions have been changed).

**Limitations:**

The computational overhead (roughly doubled training time in the experiments) is discussed explicitly. I did not see discussion around the reliance of the proposed loss on a post-hoc conformalization step.

**Strengths And Weaknesses:**

Strengths:
* The method is novel as far as I can tell (although I am not familiar at all with the related literature on frequentist approaches for uncertainty estimation) and appears to be technically sound.
* The technical setup is well-structured and clear for the most part (although see exceptions).
* The proposed method seems to perform the best on the synthetic data and is competitive on real data.

Weaknesses:
* I found section 3.3 quite vague and think the paper would benefit from recapping the specific techniques it uses from the referenced papers. As it stands, the text is not self-contained and I don't think I'd be able to implement the method without working through other works. The algorithm box does not help in this regard as it is quite wordy rather than technical and concise.
* I was fairly surprised that the proposed method appears to require a post-hoc calibration step (lines 199-201, 259, 305), even though the introduction criticizes this two-step approach as a limitation of conformal learning.
* The empirical results on the real data seem to be somewhat of a mixed bag with many close scores. Reporting results without early stopping seems unnecessary to me since the classification performance is significantly better with early stopping (although I am surprised by the need for it; see questions).
* There is no ablation study on the hyperparameter for the weight of the conformal loss. I'd imagine that this parameter allows for trading off more accurate vs better calibrated predictions, but it would be helpful to see this confirmed experimentally.

---

> ### Author Response · Authors · 2022-08-01
> **Response to comments by Reviewer m7U1 (Part 1, continued below)**
>
> Weakness (1). As mentioned in our discussion with Reviewer 9Dxm, we thought it might be a bit overwhelming to lay out all the technical details of our method implementation in the paper, and therefore we focused on explaining the key ideas. We also thought it would be unnecessary to write all implementation details in the paper because our code has been made available to the reviewers (and it will be published with the paper). In case you missed it, the code can be found in the “code” folder of the supplementary archive. Therefore, the reviewers and the readers do not need to figure out how to implement what is described in the paper solely by reading the main text. Everything is already implemented for them in the accompanying code. The code is well organized and documented, so it is not difficult to see what it does and modify it as desired. Simple usage examples are also provided in the “examples” sub-directory.
> Regarding the existing fast soft-sorting and ranking techniques which we utilize to implement our method, they are indeed a bit too complex to explain in a self-contained way within this paper. However, we cite the relevant papers and we refer the interested readers to those. For many readers however, it might be unnecessary to gain a full understanding of the inner workings of those techniques. The authors of [74] have made a really nice and user-friendly software package publicly available, and that is how we apply fast soft-sorting and ranking.
> That being said, we will follow the suggestion and include more implementation details (as well as a longer, more technical version of Algorithm 1) in the supplement if given the opportunity to revise the paper. We are also working on an even more carefully documented release of the software, which will be made available as an update through GitHub.
>
> Weakness (2). The limitation of conformal inference discussed in the introduction of our paper is not that a post-hoc calibration step is required to mathematically guarantee valid marginal coverage. The limitation discussed in the introduction is that the typical approach in the literature consists of carrying out the training phase in a way that is completely unaware of the subsequent post-hoc calibration. As a result of this lack of coordination, the output prediction sets may have unnecessarily low conditional coverage. To criticize this standard two-step approach as inefficient is not the same as to suggest that all possible two-step approaches are inefficient. To the contrary, the contribution of this paper is precisely that of developing a more coherent and coordinated two-step training/calibration conformal prediction framework.
>
> As demonstrated in this paper, our new training method leads to models whose predicted probabilities are naturally better calibrated and which lead to smaller prediction sets with higher conditional coverage when applied in combination with post-hoc conformal calibration. As also discussed in our response to reviewer Xxp1, post-hoc conformal calibration is relatively less crucial for models trained with our method compared to models trained through standard means. To illustrate this point, we have added several figures (NewFigures 5-9) in the new supplementary material (response_figures.pdf) which summarize the results of new numerical experiments conducted without post-hoc calibration. These results demonstrate that prediction sets computed by models trained with our method are closer to having valid coverage compared to models trained by other means, even if no post-hoc conformal calibration is performed. These new results will be integrated into the main paper if we are given the opportunity to revise it. While such observations speak positively as to the performance of our method, we do not believe they would justify skipping the post-hoc calibration step, and indeed we never advocate to do so. To the contrary, we believe it is generally a good idea to perform post-hoc calibration regardless of how the model is trained because that is the only currently available solution to obtain rigorous finite-sample mathematical guarantees. Our goal in this paper is simply to provide methods to train models that can be later calibrated as smoothly and efficiently as possible.

---

> > ### Author Response · Authors · 2022-08-01
> > **Response to comments by Reviewer m7U1 (Part 2, continued below)**
> >
> > Weakness (3). We don’t quite agree that our results on real data are a “mixed bag”. We actually think they are quite informative and clearly encouraging, but it is true that they may take a moment to parse. Uncertainty is a subtle concept to quantify and communicate, and it has to be measured carefully along different dimensions. First, as discussed with reviewer 9Dxm, conditional coverage and prediction set size need to be interpreted together. In fact, the trivial prediction sets which always include all possible labels would have perfect 100% conditional coverage but would also be completely useless. Second, the room for improvement in conditional coverage within the CIFAR-10 data set (Table 1) is small, because the benchmark methods already do quite well (see our discussion of epistemic vs aleatory uncertainty with reviewer 3Bsi). Yet, our method fully achieves the desired 90% conditional coverage in Table 1 without an excessive increase in the size of the prediction sets. Third, our results on the credit card data set are even more clearly positive: we achieve 60% conditional coverage while the best alternative benchmark only achieves 53%; again, this improvement is obtained without an excessive increase in the size of the prediction sets. These results are far from being mixed, and they are consistent with the picture drawn from the simulated data experiments.
> > Regarding the comparison of different methods with and without early stopping, this is actually very important and informative. Early stopping is a well-known technique which can serve as an implicit form of regularization and can often mitigate overfitting and lead to 1-guess predictions with higher test accuracy. Therefore, it is important to include it in our experiments for a fair comparison with all benchmarks. At the same time, early stopping is not a satisfactory solution for overconfidence. In fact, early stopping is typically implemented in such a way as to approximately maximize 1-guess prediction accuracy, not to mitigate overconfidence. This fact is consistent with our observations that early stopping does not seem to be as useful in combination with our method, and it can make the conditional coverage of models trained via cross-entropy even worse. In this sense, it is not true that “Reporting results without early stopping seems unnecessary [...] since the classification performance is significantly better with early stopping”. As shown in Tables 1 and 2, early stopping leads to (mostly) higher 1-guess accuracy for our benchmark methods, but at the same time it often makes their overconfidence problems even worse (lower conditional coverage). We hope that this clarifies the confusion. In any case, we are grateful for your question because these are important discussions which deserve more space in the manuscript. We will be glad to include them in a shorter format if given the opportunity to revise the paper.
> >
> > Weakness (4). We have indeed studied the effect of different values of the hyper-parameter lambda in our loss function, but we omitted some results in the interest of space. We appreciate the opportunity given to us here to provide some additional information about the effect of this hyper-parameter. Let us recall that the parameter lambda controls how closely our loss function resembles the traditional cross-entropy. If lambda=0, our loss function reduces to the standard cross-entropy; with that choice, we would expect our method to yield models with reasonably accurate 1-guess predictions, but without mitigating overconfidence. Towards the other end of the spectrum (lambda=1), the cross-entropy loss component of our loss function effectively disappears. This makes the learning progress much slower (because the novel conformity score function is noisier and harder to optimize) and it tends to lead to models that are not very accurate (high coverage but uninformatively large prediction sets). Empirically, we found that values of lambda between 0.1 and 0.3 work best in practice (see details in the appendix), but in principle this hyper-parameter could be tuned using hold-out data, possibly the same data used by other methods for early stopping).
> > To provide some concrete evidence in support of the above intuitive discussion, we have added additional figures in the new supplementary material “response_figures.pdf”; see NewFigures 2–4 therein. Those figures report on experiments with the synthetic data sets of Section 4.1 in which our hyper-parameter lambda is varied between 0 and 0.7 (we do not go above 0.7 because the training often becomes too slow and ineffective beyond that point). These results indicate that values of lambda between 0.2 and 0.3 lead to the smallest prediction sets with the highest conditional coverage. Recall from Section A3.1 that the experiments reported in our paper are based on lambda=0.2. We will include the new figures in the supplementary material if given the opportunity to revise it.

---

> > > ### Author Response · Authors · 2022-08-01
> > > **Response to comments by Reviewer m7U1 (Part 3, continued below)**
> > >
> > > Questions. “I'm overall not too confident in my review due to being unfamiliar with the related literature and commonly used experimental setups and results”. We appreciate the honesty, but we would like to reassure you that we found your questions to be quite insightful, and we are grateful for the opportunity they give us to provide further clarifications.
> > >
> > > Question (1). This question is related to a similar question reviewer 3Bsi. Our guiding theoretical result (Proposition 1) says that the conformity scores of a well-calibrated model should be uniformly distributed when evaluated on independent (or hold-out) data. The “hold-out” part of this statement is crucial. Having uniformly distributed conformity scores on the training data is perfectly consistent with the model being grossly overfitted and overconfident, and thus it says nothing about whether the predictions for future test points will be more or less well calibrated. This is why we prepare the model to produce well-calibrated predictions for future test points by training it to produce approximately uniform conformity scores for hold-out data. Please see our longer answer to reviewer 3Bsi and other related responses to other reviewers for further details.

---

> > > > ### Author Response · Authors · 2022-08-01
> > > > **Response to comments by Reviewer m7U1 (Part 4, continued below)**
> > > >
> > > > Question (2). As mentioned in our answer above to your earlier comment, early stopping is a well-known technique which can serve as an implicit form of regularization and can often mitigate overfitting and lead to 1-guess predictions with higher test accuracy. However, early stopping is not a satisfactory solution for overconfidence because it is typically implemented in such a way as to approximately maximize 1-guess prediction accuracy, not to mitigate overconfidence. This fact is consistent with our observations that early stopping does not seem to be as useful in combination with our method, and it can make the conditional coverage of models trained via cross-entropy even worse.
> > > >
> > > > Regarding the performance of different methods on the CIFAR-10 data, this is related to a similar question by reviewer 3Bsi. The reason why we do not reach 90%+ accuracy with Resnet18 on CIFAR-10 is that we do not utilize, for simplicity, all the different data augmentation techniques applied in the paper mentioned by the reviewer. As prior works have demonstrated empirically, data augmentation can be a very effective strategy to boost predictive accuracy in image classification tasks. However, as it involves the very definition of the available data set, we see it as a somewhat orthogonal issue to that of designing the loss function and learning algorithm, which is the problem we consider here. There is no particular reason why the conformalized training ideas discussed in this paper could not in the future be applied in combination with data-augmentation techniques, and in fact we suspect that it might help to do so. However, it would not be completely straightforward to add image augmentation in this paper. First, it would excessively lengthen the manuscript and complicate the comparison with different methods, because it would result in lots of different combinations of training algorithms and data augmentation techniques to consider. Second, the theory behind conformal inference typically assumes all data points to be exchangeable with one another, which is not necessarily true after data augmentation. This issue can likely be addressed in an effective and theoretically rigorous way, but the topic deserves sufficient attention and care that it is best left to future work. We will emphasize this exciting opportunity for further research if given the opportunity to revise this manuscript.
> > > >
> > > > Regarding why we corrupted some images in CIFAR-10, instead of working only with the original clean images, is that this is a data set with very low aleatory uncertainty (see our response to reviewer Xxp1 for a full discussion of different sources of uncertainty). As explained in the introduction, overconfidence is an especially concerning problem in applications with significant aleatoric uncertainty, and our method is specifically designed to address that issue. Therefore, the clean CIFAR-10 data set would not offer the most interesting application, because uncertainty isn’t really a huge concern there and there is very little room for improvement. That being said, we have explored the effect of different proportions of corrupt images in the training data on the performances of different methods in Figure A27. We have also performed additional experiments on the CIFAR-10 to investigate the effect of covariate shift (changes in the proportion of corrupt images) in the test set; see our answer to the related comment by Reviewer 9Dxm for further details. The results of these additional experiments are included in NewFigure 10, within the new supplementary document “response_figures.pdf”. These results show that our method is more robust to covariate shift compared to all alternative benchmarks, consistently with the results in Figure A5 pertaining synthetic data. We will insert this figure into the paper if given the opportunity to revise it. Unfortunately, we did not have sufficient time to carry out the particular experiments you suggested within the limited time frame allowed for this rebuttal period, because that would require re-training all models on clean CIFAR-10 data, which is not something we have already done. In any case, we hope that this response can already answer your question.
> > > >
> > > > Question (3). Please see our answer to your comment about “weakness 4”, which also answers this question.

---

> > > > > ### Author Response · Authors · 2022-08-01
> > > > > **Response to comments by Reviewer m7U1 (Part 5, continued below)**
> > > > >
> > > > > Minor note (1). Neural nets trained with proper scoring rules may very well be calibrated in some imaginary infinite-data limit, but unfortunately this does not mean they are calibrated in practical applications with finite data sets. They are typically not even close to being well-calibrated; they are often very poorly calibrated. Think of Figure 1, and most other results in our paper: it’s not easy to get high conditional coverage, which means the models are not well calibrated. Even when neural networks happen to be approximately calibrated in some marginal “average” sense, they are usually not calibrated at all for many different types of test cases (poor conditional coverage). The difficulty of training neural networks (or other machine learning models) that are well-calibrated in practice is notorious, which is why there is a rich literature on all sorts of regularization techniques, early stopping strategies, post-hoc calibration methods, and conformal inference to try to address this problem.
> > > > >
> > > > > We also find ourselves at disagreement on the statement that “Intrinsically noisy data is not problematic, as this would be reflected in the training set and lead to less confident predictions.”. If only that were true! Deep neural networks were not originally designed to work with intrinsically noisy data, and indeed the comments of other reviewers seem consistent with our impression that aleatory uncertainty has not always received sufficient attention in deep learning. The practical overconfidence of deep neural networks is a particularly serious problem with intrinsically noisy data even if there is no covariate shift, as our numerical experiments and applications demonstrate (compare for example Tables 1 and 2). Again, the confusion here might arise because the reviewer is thinking more about theoretical limits of infinite data. Regardless of whether deep neural networks could in theory become naturally well-calibrated using any (reasonable) loss function in some abstract infinite-data limit, we have to deal with the fact that they are not really doing well enough from that regard in the real world (see Table 2, for example). The goal of this paper is not to suggest that some theoretical flaw in traditional loss functions fundamentally prevents them from learning well-calibrated models in some infinite-data-limit abstraction. Our goal is to provide a concrete and directly applicable solution to the very real ill-calibration problems that anyone can easily observe in many applications.
> > > > >
> > > > > Finally, regarding covariate shift, we do agree: of course covariate shift makes calibration even more challenging. See for example our Figure A5, or the new figures added in this rebuttal, as discussed above. However, covariate shift is not the only problem. We hope that at this point we have made our case that it is generally not easy to train well-calibrated deep neural networks, with or without covariate shift, but our method can be useful for that purpose under both settings.
> > > > >
> > > > > Minor note (2). As discussed in previous answers to related comments, Figure A5 reports on the results of numerical experiments based on synthetic data with covariate shift. Additionally, we have carried out additional experiments based on the CIFAR-10 data under different degrees of image corruption in the test set (covariate shift). These results are included in NewFigure 10, within the new supplementary document “response_figures.pdf”. These figures show that our method is more robust to covariate shift compared to all alternative benchmarks, consistently with the results in Figure A5 pertaining synthetic data. We will insert this figure into the paper if given the opportunity to revise it.
> > > > > It is interesting to spend a few words to explain why it should be unsurprising that our method performs relatively well under these forms of covariate shift even though data exchangeability is violated. The reason is that our method practically achieves higher conditional coverage on these data sets compared to the benchmarks, and methods with perfect conditional coverage are theoretically immune to covariate shift.

---

> > > > > > ### Author Response · Authors · 2022-08-01
> > > > > > **Response to comments by Reviewer m7U1 (Part 6, last)**
> > > > > >
> > > > > > Limitation.  Regarding the use of post-hoc calibration, this issue is related to a comment by Reviewer Xxp1. Our method is specifically designed to train models that are approximately calibrated on their own, which tends to make the post-hoc conformal calibration step relatively less crucial compared to models trained through standard means. To illustrate this point, we have added several figures (NewFigures 5-9) in the supplementary material (response_figures.pdf) which summarize the results of new numerical experiments conducted without post-hoc calibration, on both real and synthetic data. These results demonstrate that prediction sets computed by models trained with our method are much closer to having valid coverage compared to models trained by other means, even if no post-hoc conformal calibration is performed. These new results will be integrated into the main paper if we are given the opportunity to revise it. While such observations speak positively as to the performance of our method, we do not believe they would justify skipping the post-hoc calibration step, and indeed we never advocate to do so. To the contrary, we believe it is generally a good idea to perform post-hoc calibration regardless of how the model is trained because that is the only currently available solution to obtain rigorous finite-sample mathematical guarantees. Our goal in this paper is simply to provide methods to train models that can be later calibrated as smoothly and efficiently as possible. If given the opportunity to revise this manuscript, we will indicate more clearly in the introduction that post-hoc conformal calibration remains a good idea with any model, including those trained with our method.

---

> > ### Comment · Reviewer_m7U1 · 2022-08-06
> > **Thank you for the response**
> >
> > Thank you for your detailed response. I particularly appreciate the clarification on the post-hoc recalibration and the additional results demonstrating the reduced reliance on this second step. I’m happy to facilitate a consensus between the reviewers and will increase my score.
> >
> > A couple more specific comments on the response:
> >
> > * **W1**: I appreciate that under the hood these differentiable sorting methods may be complex and that the reader does not need to understand their inner workings if software packages supporting them exist. However, I would imagine that based on the API and abstractions of such a software package it would be possible to provide an intuition to the reader of what the core functionality necessary is. A deep learning paper won’t discuss the inner workings of PyTorch’s autodiff module to calculate gradients, but will give an idea of e.g. the innovations on a new layer type/architecture etc that would fit into the module library. I also appreciate that you intend to release code accompanying the paper, however the core artifact of a publication is still the paper itself, so the paper should give at least a good idea of the implementation (even if it is not possible to reproduce all results without the code, but most readers won’t attempt to do this, so they shouldn’t have to refer to the code to understand the paper).
> > * **W3**/**Q2**: Thank you for the clarifications, I had not noticed that you weren’t using data augmentation, I can see that this may lead to a need for early stopping. I would suggest mentioning as a limitation that the method can’t be used with data augmentation out-of-the-box.

---

> > > ### Author Response · Authors · 2022-08-09
> > > **Thank you!**
> > >
> > > Dear Reviewer m7U1,
> > >
> > > Thank you for taking the time to consider our long response, and for your good suggestions of adding further technical details about differentiable sorting as well as mentioning the possible complications related with using our method with data augmentation.
> > > We will follow your advice in the next round of revision, or while preparing the camera-ready manuscript if our paper is accepted.
> > >
> > > Sincerely,
> > > The auhtors

---

### Official Review · Reviewer_9Dxm · 2022-07-11

**Rating:** 6
**Confidence:** 3
**Soundness:** 3 good
**Presentation:** 3 good
**Contribution:** 3 good

**Summary:**

The authors propose a method of combining the learning and calibration phases
of conformal inference, reducing overconfidence in the ML model, and giving
smaller prediction sets. They minimize a new loss with minimizes the
discrepancy in conformity scores between the current model and an unknown
oracle.


**Questions:**

- I am not sure there is enough information in the current version of the paper in order to implement the algorithm. Section 3.3 is easy to intuitively understand, but it leaves me confused as to where to go in order to actually implement what is written in the paper. Could a more detailed explanation of the exact equation needed to reproduce $\ell_u$ be included?

# Summary

Overall, I think the contribution is solid, but there can be some improvements made as reflected in my comments and questions.

**Limitations:**

The authors have highlighted the computational complexity as a limitation. I am not aware of any adverse societal impacts of their work.

**Strengths And Weaknesses:**

# Strengths

- The method id well justified and comes equipped with in depth analysis
- The experiments verify the claims made within the text and also provide ablations on the proposed components.
- The experiments are thorough.

# Weaknesses

- Equation three seems to give incomplete infromation. $u$ and $\tau$ are
defined but there is no mention of how they contribute to the function
$\mathcal{S}$. It is then defined in the appendix, but A2 in the appendix takes
$t$ (not $\tau$ but I guess they are the same?) as an argument and equation 3 takes $1 - \alpha$ leaving $\tau$ unused.
This needs to be cleared up.

- The covariate shift experiments occur on the synthetic data. It would be nice
to see if these results hold for larger datasets as well. For example, what
would happen if a model were trained with regular CIFAR10 data with no
corruptions in either the training or calibration set, and then tested with the
random erase corruptions or CIFAR10-C [1] test set.

- L237: states that figure A6 shows typically higher conditional coverage as
compared to all benchmarks. It appears there is actually a clear trend which
shows that the proposed model performs worse on larger numbers of classes, yet
there is no mention of this. Can the authors provide any explanation for this?

# References

[1] https://github.com/hendrycks/robustness

---

> ### Author Response · Authors · 2022-08-01
> **Response to comments by Reviewer 9Dxm**
>
> Weakness (1). Thank you for pointing out that some of the notation in Section 2.2 can be clarified. We agree that right now it relies too heavily on Appendix A1, and even so there are some discrepancies which may be confusing. We will be happy to clear this up and make Section 2.2 self contained if given the opportunity to revise this manuscript.
>
> Weakness (2). We are glad to read you found the experiments demonstrating increased robustness to covariate-shift interesting (e.g., Figure A5). Your suggestion to include additional covariate-shift experiments involving the CIFAR-10 data set is also very interesting. It is regrettable that we don’t have enough time to carry out those experiments within the short time window allowed for this rebuttal phase, but we have managed to perform new experiments that are very similar to what you suggested. The approach we followed was faster to implement than what you suggested because it didn’t involve re-training the models we saved, but it is very close in spirit. What we did is that we applied the pre-trained models from our paper to a modified CIFAR-10 test set in which the percentage of corrupted images is varied as a control parameter. The results are shown in NewFigure 10, within the new supplementary document “response_figures.pdf”. These results show that our method is more robust to covariate shift compared to all alternative benchmarks, consistently with the results in Figure A5 pertaining synthetic data. We will insert this figure into the paper if given the opportunity to revise it. We hope this answers your question satisfactorily.
>
> Weakness (3). Figure A6(a) shows that the conditional coverages obtained with our method and with the focal loss decrease as the number of classes increases, while those of the other benchmark methods remain more or less constant. Meanwhile, Figure A6(b) shows that the corresponding size of the prediction sets increases very rapidly for all methods except ours. These results need to be interpreted together, because conditional coverage does not mean much by itself without taking the size of the prediction sets into consideration. In fact, the trivial prediction sets which always include all possible labels would have perfect 100% conditional coverage but would also be completely useless. Now, that being said, do we have a good theoretical understanding of why different methods seem to find different trade-offs between conditional coverage and size of the prediction sets depending on the number of possible labels? Not really: this is an interesting question which may need to be investigated in more depth by future work. We will better highlight this subtlety in the paper if given the opportunity to revise it; thank you for pointing it out.
>
> At the same time, does this curious phenomenon reflect a weakness of our method? Hardly so, we argue. According to Figure A6, the relative drop in conditional coverage that we experience with 12 labels is not huge (64% vs 70% of hybrid and cross-entropy, out of target 90%), but difference in the size of the prediction sets is very large (3.75 vs 7.75 of hybrid and cross-entropy, out of 12 possible labels). It is reasonable to imagine that a prediction set with less than 4/12 labels could be much more informative than one with almost 8/12, even if the conditional coverage is a little lower. Of course, it would be nice if we could always outperform all benchmarks with respect to every meaningful metric, but the relevant metrics are often competing with one another, and therefore that is quite an unrealistic goal. In light of this discussion, we hope the reviewer can agree that, by any reasonable holistic measure, our method does quite well in Figure A6 (even though we do not beat the benchmarks as thoroughly there as in other settings; e.g., Figure A5).
>
> Question (1). We thought it might be a bit overwhelming to lay out all the technical details of our method implementation in the paper, and therefore we focused on explaining the key ideas. We also thought it would be unnecessary to write all implementation details in the paper because all of our code has been made available to the reviewers (and it will be published with the paper). In case you missed it, the code can be found in the “code” folder of the supplementary archive. Therefore, the reviewers and the readers do not need to figure out how to implement what is described in the paper solely by reading the main text. Everything is already implemented for them in the accompanying code. The code is well organized and documented, it is not difficult to see what it does and modify it as desired. Simple usage examples are also provided in the “examples” sub-directory.  That being said, we would be happy to include more implementation details to the supplement if given the opportunity to revise it. We are also working on an even more carefully documented release of the software, which will be made available as an update through GitHub.

---

> > ### Comment · Reviewer_9Dxm · 2022-08-05
> > **About weakness [2]**
> >
> > Thank you or your response and extra experiment on Weakness(2). My main point in asking this question was to see how the model behaved when seeing corruptions which were not present in the training set. Section 4.2 says that the training set contains corrupted images too.
> >
> > If I am not mistaken, the pretrained models you have can be compared on the CIFAR-10 test set in this way without any retraining, correct?

---

> > > ### Author Response · Authors · 2022-08-06
> > > **Authors reply**
> > >
> > > There is a good reason why all models in our experiments are trained on data containing at least a few corrupted images. How could any model otherwise learn how to properly deal with aleatory uncertainty, after being trained on clean CIFAR-10 data which contain virtually no aleatory uncertainty? The strength of our method is that it can be more effective than its benchmarks in recognizing uncertainty in the training data and learning how to account for it at test time. However, despite its relatively good performance, our method is not "magic". It cannot learn about patterns that are never observed in the data, and neither could any other realistic alternative approach! This is why we think it is more informative to carry out experiments in which we train our method on a mixture of clean and corrupted images, and then we apply it to test data involving similar mixtures with varying proportions. These are in a certain sense simpler experiments than you suggest, but they are better fitted to the main point of this paper and they are still highly non-trivial, as demonstrated by the clearly less than ideal performance of the existing benchmarks.
> > >
> > > Analogous reasoning explains why we did not try to apply our models, nor any of the benchmarks, to CIFAR10-C test data. The CIFAR10-C images are affected by a completely different set of possible corruptions, including manipulation of contrast, brightness, sharpness/blur, level of noise, etc. If we were to apply our pre-trained models (fitted on CIFAR10 with masking corruptions) to CIFAR10-C images, the results would speak more as to the robustness of these models to major distributional shifts than to their ability to learn about previously observed uncertainty. Of course, uncertainty estimation and robustness to distributional shifts are related issues, but they are also clearly distinct insofar as this paper goes. This paper focuses on uncertainty estimation and conditional coverage, not on robustness to completely new and previously unseen types of data. If we wanted to obtain models that are more robust to completely new types of images, we would most likely need even more sophisticated models trained on much more diverse types of data (e.g., image classification data with lots of different types of corruption) [1]. We think this is a very interesting direction for future work, and we would be happy to mention it in a paper revision.
> > >
> > > [1] D. Hendrycks, N. Mu, E. D. Cubuk, B. Zoph, J. Gilmer, and B. Lakshminarayanan. "AugMix: A Simple Data Processing Method to Improve Robustness and Uncertainty." In International Conference on Learning Representations. 2019.

---

> > > > ### Comment · Reviewer_9Dxm · 2022-08-07
> > > > **Thanks.**
> > > >
> > > > I see that the strongest case for the model is using it as used in the experiments in the paper. But generalizing to other corruptions may actually be a side effect of the outlined training procedure. Without experiments, one can never know. Even if they were to fail, they would provide more insight as to the full extent of the method and what can be worked on in the future.
> > > >
> > > > Anyway, I still think the method is useful and novel which is reflected in my original score.
> > > >
> > > > Thanks

---

> > > > > ### Author Response · Authors · 2022-08-07
> > > > > **Thanks!**
> > > > >
> > > > > Thank you for the discussion!
> > > > > We will think about how to address the important follow-up problem you suggested.

---

### Official Review · Reviewer_3Bsi · 2022-07-11

**Rating:** 7
**Confidence:** 4
**Soundness:** 3 good
**Presentation:** 3 good
**Contribution:** 2 fair

**Summary:**

This paper proposes a differentiable loss function for training conformal predictors. In conformal prediction, a popular scoring function for classification sets is the APS method of Romano et. al. [1], as if the learned conditional distribution $\hat \pi_y(x) \approx  \pi_y(x) := \mathbb{P}(Y = y \mid X = x)$ is exact, then the constructed prediction sets are the smallest randomized prediction sets with the desired conditional coverage, i.e., satisfying $\mathbb{P}(Y_{n+1} \in C(X_{n+1}) \mid X_{n+1} = x) \geq 1 - \alpha$. Typically, the way to go about solving this problem is to first learn $\hat \pi_y(x)$, and then plug it into the APS method (which is then calibrated using normal conformal techniques). Unfortunately, as the authors point out, predictions based on poorly pre-trained $\hat \pi_y(x)$ are hard to correct once $\hat \pi_y(x)$ is fixed. This paper links the two steps of conformal prediction into one joint loss, with a focus on improving conditional coverage via better recovering the oracle behavior of the true $\pi_y(x)$ combined with APS. This joint objective is composed of the normal cross-entropy classification loss, together with a regularizer that encourages conformal scores on held-out data to be uniformly distributed (as they would be if $\hat \pi_y(x) = \pi_y(x)$). The method is empirically validated on both synthetic and real datasets, though the gains are fairly minor depending on the setting.

[1] Classification with Valid and Adaptive Coverage. https://arxiv.org/abs/2006.02544.

**Questions:**

- Low 80% top-1 accuracy seems pretty low for a ResNet-18 on CIFAR-10 (see, e.g., https://github.com/kuangliu/pytorch-cifar). This makes me question if the cross-entropy baseline is indeed well trained? Can you explain this difference?

**Limitations:**

The authors have adequately addressed the limitations and potential societal impact of their work.

**Strengths And Weaknesses:**

=== Strengths ===

- The paper is well-written and well-evaluated. The method is also well-motivated. In particular, I appreciate the focus on conditional coverage rather than set size, as set size can be gamed when only subject to marginal coverage constraints.

- The proposed algorithm is simple (at least when relying on pre-existing differentiable sorting algorithms). Proposition 1 yields good intuition (and also interestingly appears somewhat related to the fact that the distribution of $Z = F_{Y|X}(Y)$ is uniform for continuous r.v.s $Y$ with CDF $F_{Y|X}$, but here by construction for discrete $y$).

- Though I have some questions about the empirical effectiveness on real data (see below), at least under certain settings the method can lead to smaller prediction sets with better conditional coverage---two important and impactful qualities for real-world deployment of conformal algorithms.

=== Weaknesses ===

- When factoring in early stopping, the empirical gains on real datasets vs. baselines (i.e., Hybrid) appear minor at best (e.g., the best seems to be a +4% gain in corrupted coverage on CIFAR-10 when comparing fully-trained conformal with early-stopped hybrid?). The raw (top-1) accuracy also seems to be negatively on the credit card default task (whereas standard conformal methods that don't modify the base conformal score don't affect the top-1 accuracy).

- This is addressed in the paper, but the proposed method is quite expensive to train (2x compared to cross-entropy loss, as reported in Section 5). As a result, I'm not sure how well this would scale to larger models than a ResNet18. Given the somewhat small gains in efficiency/conditional coverage, the impact of this approach seems likely to be somewhat limited. (That said, I imagine that the "Hybrid method" of [1] is similarly slow, as it also involves a differentiable sort.)

- One thing which bothers me is the data inefficiency of the proposed approach. It seems rather wasteful to only use labeled data in $\mathcal I_2$ for regularization. It seems if the main goal is to simply get a better estimate of $\pi_y(x)$ by reproducing its behaviour when plugged into conformal APS (e.g., uniform score distr.), what if we just use that data to train a bigger, modern model (e.g., see models in [2]), optionally with better regularization (e.g., calibration objective in [3])? Or, since the method also requires validation of $\lambda$, early stopping on the same validation set (as done in the experiments, which improves baselines substantially) also seems to be a fair data-wise comparison.

- Likewise, from the paper experiments, benefits become more pronounced with more data, however, one could argue that uncertainty estimation is particularly important for tasks in which we don't have much data (and hence base models are poorly trained). Note that this data inefficiency is also shared by [1], but the objective in [1] can dependent on properties of the set $C$ that may be orthogonal to conditional coverage (and recovering $\pi_y(x)$ may not yield the oracle).

=== Minor ===

- Notation: In several places $|\mathcal I_*|$ should be used in place of $\mathcal{I}_*$ (e.g., L165).

[1] Learning Optimal Conformal Classifiers. https://arxiv.org/abs/2110.09192.

[2] Revisiting the Calibration of Modern Neural Networks. https://arxiv.org/abs/2106.07998.

[3] Trainable Calibration Measures For Neural Networks From Kernel Mean Embeddings. https://proceedings.mlr.press/v80/kumar18a.html.

---

> ### Author Response · Authors · 2022-08-01
> **Response to comments by by Reviewer 3Bsi (Part 1, continued below)**
>
> Weakness (1).  The 4% gain in conditional coverage for corrupt CIFAR-10 images (Table 1) is not as large in absolute terms as the corresponding gains in other applications, but it is not a weakness. The target coverage here is 90%, and the hybrid method with early stopping already performs well at 86%. Our 4% gain, from 86% to 90%, bridges 100% of the gap between the desired coverage and the empirical coverage achieved by our top competitor. The hybrid method works quite well on the CIFAR-10 data, but not on the credit card data; see Table 2. The latter are more interesting for our purposes and offer more room for improvement because they have more aleatoric uncertainty, see answer to reviewer Xxp1.
>
> Regarding the second part of the question, the credit card data set is both noisy (high aleatoric uncertainty) and imbalanced—only approximately 22% of the labels are equal to 1. Therefore, accuracy is not the most meaningful measure of performance: the trivial model which predicts the label ‘0’ for all samples achieves the highest accuracy. Instead of accuracy, F-scores would be more informative. If you look at NewFigure 1 in the new supplementary document “response_figures.pdf”, you will see the model trained with our method achieves the highest F-score. We will insert this figure into the paper and include a discussion.
>
> Weakness (2). Our method is more expensive to train compared to simpler alternatives, but this should not be surprising. Uncertainty is a subtle concept, and training a machine to learn how to calibrate its own uncertainty from data, without relying on any parametric model assumptions, is a fundamentally challenging task. In fact, it is already remarkable that Resnet18 models can be trained to better understand uncertainty quite successfully on data sets of the size considered in this paper using a learning algorithm as sophisticated as ours. This would not have been possible without the recent ground-breaking advances in fast differentiable sorting and ranking [73,74]. That field is still developing, so it is reasonable to anticipate that uncertainty-aware machine learning models (both our current proposal and more sophisticated future developments) will become cheaper to train with time.
>
> Having acknowledged that our learning algorithm is more expensive than simpler alternatives, the interesting question is whether its benefits can outweigh its costs. Our answer is a confident yes. First, our extensive numerical experiments demonstrate that the benefits are quite meaningful. We have already explained in the answer to the previous question how the reviewer’s concerns that our method leads only to a small 4% increase in conditional coverage for the CIFAR-10 data and a reduction in accuracy for the credit card data are due to simple misunderstandings. The increases in conditional coverage are actually significant, even for CIFAR-10, as explained above. Similarly, the reduction in accuracy for the credit card application is a fictitious artifact of the intrinsic noise and imbalance in that data. Second, the computational cost of training our method is not extraordinarily high. We have chosen to experiment with the Resnet18 in this paper mostly because it is a very common architecture. Further, the relatively small size of this model allowed us to conduct a thorough evaluation of our method with hundreds of independent experiments with relatively limited academic resources.
>
> In conclusion, training complex classification models with a well-calibrated understanding of uncertainty is an important problem, which often deserves spending some additional resources on. Will every deep classifier be trained with our method in the future? Realistically, that seems unlikely. Even better methods could be developed by others relatively soon, and some practitioners may just not care much about capturing uncertainty, either because it is not a huge concern in their field, or because they are already working on a very tight time/computational budget and thus they cannot afford to do much about it. But it is also true that many people care enough about machine learning uncertainty to be potentially willing to apply more complex algorithms to deal with it. We have discussed how there is a large and rapidly growing literature on the subject. There are also many applications of deep learning to fields in which the data are noisy and there are lots of practical, legal, or moral reasons why the problems caused by overconfident models need to be addressed urgently.

---

> > ### Author Response · Authors · 2022-08-01
> > **Response to comments by by Reviewer 3Bsi (Part 2, continued below)**
> >
> > Weakness (3). While our method splits the training set into two disjoint subsets (one for each component of our loss function), the comparison with the alternative benchmarks is fair because we apply all of them to the full training set without any data splitting, precisely as suggested by the reviewer. The only exception is the hybrid conformal benchmark, which utilizes the same data splitting strategy as our method. We recognize that this important point should have been mentioned more explicitly in Section 4. We thought this was implicitly clear, but it wasn’t. We will remove this ambiguity if given the opportunity to revise the manuscript.
> >
> > Now, an interesting follow-up question related to this reviewer’s comment is whether our novel loss function would work equally well (or perhaps even better) if we did not split the training data. Although this alternative implementation was not presented in the paper nor researched in great depth by this work, our intuition strongly suggests that the data-splitting approach is the correct one. Our guiding theoretical result (Proposition 1) says that the conformity scores of a well-calibrated model should be uniformly distributed when evaluated on independent (or hold-out) data. The “hold-out” part of this statement is crucial. Having uniformly distributed conformity scores on the training data is perfectly consistent with the model being grossly overfitted and overconfident, and thus it says nothing about whether the predictions for future test points will be more or less well calibrated. This is why we prepare the model to produce well-calibrated predictions for future test points by training it to produce approximately uniform conformity scores for hold-out data. Data splitting, in combination with a suitable training algorithm, thus becomes a strength, not as a weakness. This explains why our method can achieve better performance compared to its benchmarks despite the more limited amount of data available to the cross-entropy component of its loss function.
> >
> > Next, all models considered in our experiments have been trained using the same early stopping criteria and the same validation data sets. The lambda parameter (chosen for simplicity to be fixed) could also be tuned using the same validation data set utilized for early stopping, so that no method has access to any additional data source and all comparisons are fair.
> >
> > Finally, regarding the last reference pointed out in this question, it should be clarified that those authors aim to improve marginal calibration, and that is very different from our goal. Marginal calibration can be achieved exactly with post-hoc conformal calibration, but it is not fully satisfactory by itself because it does not rule out the possibility that the model may be very overconfident for some test data points and under-confident for others. Our goal is much more ambitious: we aim to increase the empirical conditional coverage, while we still rely on post-hoc conformal calibration to mathematically guarantee valid marginal coverage in finite samples. Regarding the choice of the focal loss as a benchmark for our experiments, we would like to refer to our answer to a related question by reviewer Xxp1. The focal-loss has been applied quite widely and it could arguably be seen as representing the state-of-the-art. In particular, it was shown by Mukhoti et al. (2020) to outperform label smoothing (Müller et al, 2019) and other benchmarks in a variety of settings. Of course, it is possible that other existing methods may in some cases perform better than our chosen benchmarks, but it would be impractical and potentially confusing to compare the performance of our method to that of all existing alternatives. This is especially true because our method already distinguishes itself for its novelty and original focus on achieving approximate conditional calibration within a conformal inference framework.

---

> > > ### Author Response · Authors · 2022-08-01
> > > **Response to comments by by Reviewer 3Bsi (Part 3, continued below)**
> > >
> > > Weakness (4). It is not accurate to say that “uncertainty estimation is particularly important for tasks in which we don't have much data (and hence base models are poorly trained).” As discussed with reviewer Xxp1, there are two types of uncertainty: epistemic and aleatory. Epistemic uncertainty can be eliminated by increasing the training sample size or the flexibility of the model. If this were the only type of uncertainty in data science, the premise of your comment would be correct. However, our method is especially designed to deal with aleatory uncertainty. Aleatoric uncertainty refers to intrinsic randomness in the outcome to be predicted that is due to unmeasured variables, which cannot be so easily eliminated.
> > >
> > > As explained in the introduction, overconfidence is an especially concerning problem in applications with significant aleatoric uncertainty, regardless of how big the data set is. Our method addresses overconfidence with a novel training algorithm and loss function that are better equipped to take advantage of the available data in order to accurately capture uncertainty. Of course our method tends to perform relatively better when more training observations are available, but this does not mean uncertainty always disappears from large data sets. Think of Figure 1: none of the practical methods considered there achieves perfect conditional coverage, even when the training data set is large, because the constant aleatoric uncertainty is hard to capture. However, the conditional coverage obtained with our method visibly increases with the sample size (which is what we would always like to see), while the conditional coverage obtained with the benchmarks does not increase as quickly (because those methods were not really designed to capture uncertainty). Same story in Figures A8 and A12. Further, the results with CIFAR-10 shown in Figure A24 are even more striking: the conditional coverages of all methods except ours visibly decrease as the training sample size increases!
> > >
> > > In conclusion, we hope to have clarified that uncertainty is not necessarily a small-sample size issue, and that data splitting is not necessarily a weakness. Of course, it would be nice to have a new method that can achieve even better results without splitting the data, but what matters here is that our method already performs better than existing alternatives which do not split the data.

---

> > > > ### Author Response · Authors · 2022-08-01
> > > > **Response to comments by by Reviewer 3Bsi (Part 4, last)**
> > > >
> > > > Question (1). There are three main reasons why we do not reach accuracy above 90% with Resnet18 on CIFAR-10: lack of data augmentation, smaller training sample size, and the presence of corrupt training images.
> > > >
> > > > The first reason is that we do not utilize, for simplicity, all the different data augmentation techniques applied in the paper mentioned by the reviewer. As prior works have demonstrated empirically, data augmentation can be a very effective strategy to boost predictive accuracy in image classification tasks. For example, Shorten and Khoshgoftaar (2019) report that the Resnet18 accuracy on CIFAR-10 without data augmentation is only about 89%. This is still a little higher than ours, but this remaining gap can be explained by reasons 2 and 3, discussed below. Here, let us just emphasize that data augmentation involves the very definition of the available observations, and thus we see it as a somewhat orthogonal issue to that of designing the loss function and learning algorithm. There is no particular reason why the conformalized training ideas discussed in this paper could not in the future be applied in combination with data-augmentation techniques, and in fact we suspect that it might help to do so. However, it would not be completely straightforward to add image augmentation in this paper. First, it would excessively lengthen the manuscript and complicate the comparison with different methods, because it would result in lots of different combinations of training algorithms and data augmentation techniques to consider. Second, the theory behind conformal inference typically assumes all data points to be exchangeable with one another, which is not necessarily true after data augmentation. This issue can likely be addressed in an effective and theoretically rigorous way, but the topic deserves sufficient attention and care that it is best left to future work. We will emphasize this exciting opportunity for further research if given the opportunity to revise this manuscript.
> > > >
> > > > The second reason for lower accuracy is that we train on a smaller sample size. We set aside a significant number of images for early-stopping validation and for post-hoc calibration. These hold-out data sets are useful within the scope of our numerical experiments and they necessarily limit the sample size available for training, but they do not point to a limitation of our method.
> > > >
> > > > The third reason is that we are training all models on data containing a fraction of heavily corrupted images. It is not too surprising that these corrupt training images end up negatively affecting the test accuracy of all methods. However, as we discussed above, corrupt images are useful to add extra aleatory uncertainty, which makes our problem more interesting. Given that there are many other real-world applications with intrinsically noisy data (aleatory uncertainty), this characteristic of our partially corrupted CIFAR-10 data set is not unrealistic and provides an informative demonstration of the importance of reliable uncertainty estimation.

---

> > > > > ### Comment · Reviewer_3Bsi · 2022-08-09
> > > > > **Thank you for your response**
> > > > >
> > > > > Dear authors,
> > > > >
> > > > > Thank you for your very complete response, which has answered many of my original questions.
> > > > >
> > > > > - I'm satisfied with the response regarding empirical improvements; thanks for clearing up my misunderstanding, as I missed that the target coverage in that table was only 90%.
> > > > >
> > > > > - I tend to agree with the authors point that higher computational cost may be worth paying for better uncertainty quantification---and it depends on the application. Hopefully, this computational overhead may be reducible in future work.
> > > > >
> > > > > - I understand that there are both epistemic and aleatoric types of uncertainty. While your experiments may have focused on aleatoric uncertainty, and this is great, there are other problems in which epistemic uncertainty may dominate, and data collection is expensive. In these cases, as a practioner, it will be hard to decide what is better: to leave our data for calibration because some of my uncertainty may be aleatoric, or use more data to try to improve my model. Other conformal methods (like jackknife+) that have better data efficiency don't make this choice as difficult. That said, there is nothing not to disagree about when you say that uncertainty is not necessarily a small sample issue, and that data splitting is not necessarily a bad thing. I think it's worthwhile to acknowledge, though, that data splitting also has its downsides, and is not a panacea.
> > > > >
> > > > > - I'm still not sure why you couldn't have taken the typical training scheme for loss $\ell_a$. After all, you do data splitting, so simply not doing data augmentation on splits $\mathcal{I}_2$ and $\mathcal{D}_2$ would preserve exchangeability, no? It's not clear why any restrictions must be placed on $\mathcal{I}_1$, or why one cannot simply follow state-of-the-art for computing the $\ell_a$ loss (including data augmentation). Data augmentation is only performed at training time, not at inference. While it may be an _orthogonal method_, it would still be good to know how much of the _performance benefit_ your method and data augmentation methods give are independent.
> > > > >
> > > > > That said, in light of some of discussion in the response, I will be happy to raise my score.

---

> > > > > > ### Author Response · Authors · 2022-08-09
> > > > > > **Thank you!**
> > > > > >
> > > > > > Dear Reviewer 3Bsi,
> > > > > >
> > > > > > It is indeed very interesting to ask whether a more data-parsimonious version of our method (e.g., using cross-validation or jackknife hold-outs instead of sample splitting) could be developed in the future. As a first approach, it seemed intuitive for us to evaluate the novel conformity loss function on data that are never processed through the cross-entropy loss, but it is true that this is not the only possible approach in theory. Our intuition is that cross-validation or the jackknife may be relatively more susceptible to overfitting in this context---due to the multi-epoch nature of the training algorithm. However, we agree that this is something that may be worth verifying empirically in a near future.
> > > > > >
> > > > > > Thank you again for the great discussion!

---

> > > > > > > ### Comment · Reviewer_3Bsi · 2022-08-09
> > > > > > > **Thanks**
> > > > > > >
> > > > > > > Yes, I think that a jackknife+ or J+aB approach would be a great followup.
> > > > > > >
> > > > > > > I"m still not sure I understand your previous comments about what makes data augmentation hard during training though per your comment on exchangeability. Could you clarify?

---

> > > > > > > > ### Author Response · Authors · 2022-08-09
> > > > > > > > **Data augmentation and exchangeability**
> > > > > > > >
> > > > > > > > Similarly to the cross-validation or jackknife idea discussed above, there is no theoretical reason why one could not use data augmentation in combination with our training method. However, it is not so obvious how to best use data augmentation for conformal calibration. The problem is that data augmentation violates the exchangeability assumption.
> > > > > > > >
> > > > > > > > Suppose we augment each of n images in a data set with a mirror image. Even if we assume the original n images were exchangeable, the new 2n augmented images clearly are not, because they are tied pairwise by mirror relationships. In other words, suppose I give you this data set: (Image 1, Mirror Image 1, Image 2, Mirror Image 2, Image 3, Mirror Image 3, Image 4, Mirror Image 4). Could you tell that this data set is statistically different from the following: (Image 4, Mirror Image 3, Mirror Image 1, Image 2, Mirror Image 2, Image 1, Image 3, Mirror Image 4)? Yes, it should be quite clear, because there was meaning in the order of the images in the first data set, but there is no meaning in the second data set.
> > > > > > > >
> > > > > > > > Given that we are trying to be as precise and rigorous as possible in this paper, it doesn't feel right to simply naively go ahead with data augmentation disregarding this subtle exchangeability issue. We think that combining data augmentation with our method is possible, but it must be done with care. We feel that this problem is sufficiently challenging and important to be best left to follow-up work.
> > > > > > > >
> > > > > > > > We hope this answers your question. Please let us know if something is unclear or if you have any more thoughts!

---

> > > > > > > > > ### Comment · Reviewer_3Bsi · 2022-08-09
> > > > > > > > > **Re: Data augmentation and exchangeability**
> > > > > > > > >
> > > > > > > > > The general case I understand. Except my question was in reference to the fact that you do data splitting---what prevents data augmentation of split $\mathcal{I}_1$ only? The exchangeability of $\mathcal{I}_2$ and $\mathcal{D}_2$ are unaffected, no? This is the only split where it would apply for training a good base model anyway.

---

> > > > > > > > > > ### Author Response · Authors · 2022-08-09
> > > > > > > > > > **Re: Re: Data augmentation and exchangeability**
> > > > > > > > > >
> > > > > > > > > > > Except my question was in reference to the fact that you do data splitting---what prevents data augmentation of split $\mathcal{I}_1$ only?
> > > > > > > > > >
> > > > > > > > > > Oh, got it now: thanks for clarifying. You're right, it is easy and it makes sense to use data augmentation in $\mathcal{I}_1$; we could have done that. We'd be happy to mention the idea in the paper. Your question previously got us thinking about data augmentation in $\mathcal{I}_2$ and $\mathcal{D}_2$ simply because that would be conceptually more interesting from the point of view of the methodological innovations proposed in this paper, but it is not straightforward.

---

> > > > > > > > > > > ### Comment · Reviewer_3Bsi · 2022-08-09
> > > > > > > > > > > **Another thought**
> > > > > > > > > > >
> > > > > > > > > > > Thanks. Not to beat a dead horse, but then as we agree that it is indeed feasible within your framework, adding in data augmentation to see how your method compares with stronger base models would be a good addition. Having such low accuracy compared to other Resnet18s only raises questions, needlessly.
> > > > > > > > > > >
> > > > > > > > > > > Another thought: $X_1, f(X_1), X_2, f(X_2)$ aren't exangeable, but $X_1, X_2$ and $f(X_1), f(X_2)$ are.  Or, even $F(X_1), F(X_2)$ should be as well, where $F$ is a random function. Would it help your method (perhaps even from a data efficiency view, requiring smaller $\mathcal{I}_2$), if you randomly sampled a single augmentation per image at each round (i.e., randomly keep only the original image, or its mirror, if the augmentation was $F \in \{ \text{original, mirror} \}$.

---

> > > > > > > > > > > > ### Author Response · Authors · 2022-08-09
> > > > > > > > > > > > **Re: Another thought**
> > > > > > > > > > > >
> > > > > > > > > > > > Sure, that might be a good idea to try.
> > > > > > > > > > > > Perhaps we should write a follow-up paper specifically focused on data augmentation.

---

> > > > > > > > > > > > > ### Comment · Reviewer_3Bsi · 2022-08-09
> > > > > > > > > > > > > **Summary**
> > > > > > > > > > > > >
> > > > > > > > > > > > > In any event, thanks for engaging. Given the amount of discussion that has arisen, I do think that others in the community will also find it interesting to discuss--and am raising my score.
> > > > > > > > > > > > >
> > > > > > > > > > > > > I do heavily encourage you to include the extra analysis in the appendix, including the discussion of the questions brought up here (and with the other reviewers).

---

> > > > > > > > > > > > > > ### Author Response · Authors · 2022-08-09
> > > > > > > > > > > > > > **Thank you!**
> > > > > > > > > > > > > >
> > > > > > > > > > > > > > Thank you for the stimulating discussion! We will incorporate these ideas in the paper, and the extra analyses in the appendix. You have also successfully convinced us to look at data augmentation more closely in the near future.

---

> ### Comment · Area_Chair_DpLh · 2022-08-08
> **Please respond to author feedback**
>
> Thank you for reviewing this paper. Could you respond to the author feedback, or at least acknowledge that you've read the reply? Does the author reply address your concerns?
>
> Best, AC

---

### Official Review · Reviewer_Xxp1 · 2022-07-11

**Rating:** 6
**Confidence:** 4
**Soundness:** 3 good
**Presentation:** 4 excellent
**Contribution:** 3 good

**Summary:**

In this paper, a new training strategy which integrates conformal prediction in training stage is proposed. The proposed training algorithm uses an additional regularization term that encourages the conformity scores to follow a uniform distribution to improve the performance of final conformal prediction.

**Questions:**

See weaknesses.

**Limitations:**

The authors have adequately addressed the limitations and potential negative societal impact.

**Strengths And Weaknesses:**

Strengths:

1. Previous training stage doesn't consider the performance of conformal prediction, and hence results in sub-optimal problem. The proposed method addresses this issue by encouraging the conformity scores to follow a uniform distribution.

2. The proposed differentiable approximation for the empirical CDF of the conformity score could be efficiently implemented. The validation dataset could be used in both training (for calculating $l_u$) and post-hoc calibration.

3. The experiments with synthetic data show some positive points of the proposed method, and also give some intuitive analyses.

Weaknesses:

1. The authors mentioned that "The idea is to mitigate overconfidence" in Abstract. Does the proposed method (i.e., adding conformal loss term $l_u$ in loss function) directly mitigate the over-confidence issue?

2. The proposed method could be a regularization term on the loss function. What about the relationship between it and other regularization methods (which have been used to address the calibration problem) like label smoothing?

3. Other real-world datasets should be used in experiment part. This paper only uses CIFAR-10 for evaluating the effectiveness of the proposed method. As it is not too hard to classify, image augmentation processes are used to generate harder samples. Why not use datasets like CIFAR-100 or (Tiny-)imagenet for better evaluation?

---

> ### Author Response · Authors · 2022-08-01
> **Response to comments by Reviewer Xxp1**
>
> Strengths (1): The hold-out data for post-hoc calibration are distinct from those used during training. We do not recycle data for these two tasks because otherwise the final prediction sets would not be guaranteed to be well-calibrated. This is explained in Section 4, but we can clarify further. That being said, our method is designed to train models that already are approximately calibrated, which tends to make post-hoc calibration less crucial. We have added NewFigures 5-9 in the supplementary file “response_figures.pdf to illustrate this. These results show our method leads to prediction sets with higher coverage compared to models trained by other means, even without post-hoc calibration. This speaks positively as to the performance of our method, but we do not believe it justifies skipping post-hoc calibration, and indeed we never advocate doing that.
>
> Weaknesses (1). Yes, our solution mitigates overconfidence. Our loss targets overconfidence by reducing the statistical deviation from uniformity of the conformity scores. We have provided theoretical justification for this solution (Proposition 1) and extensive evidence of its efficacy (Section 4 and supplement). Other reviewers found our method to be well justified and thoroughly validated, but we are happy to clarify further. The link between the uniformity of the scores and conditional coverage is in Proposition 1, while the link between overconfidence and coverage is in the introduction. We can explain this in more detail. In particular, we can add references in Section 3.1 to Figures A3-A4 and A16-A23. Recall that Figures A3-A4 and A16-A23 demonstrate how models with lower conditional coverage are associated with sub-uniform scores. By contrast, better calibrated models lead to prediction sets with higher conditional coverage and more uniform conformity scores. The reduction in overconfidence is seen in Figure A3: the histograms for the benchmarks are shifted to the left, while ours are closer to uniform. Further, the probabilities estimated by our models are more accurate (Figure A4). Finally, the results on CIFAR-10 confirm our method is less overconfident (Figure 2).
>
> Weaknesses (2). Our loss function is based on novel ideas: we take inspiration from conformal inference, training a model which can later be utilized to construct more reliable and informative prediction sets with higher conditional coverage. It is true that other methods have been proposed to mitigate overconfidence, and the richness of this literature speaks to the importance of the problem. We acknowledged the literature, and we extensively compared our method to some representative benchmarks. It is of course possible that: (a) we might have accidentally omitted a relevant reference; (b) other methods may sometimes perform better than our “state-of-the-art” benchmarks. Point (a) is easy to address: we can add missing references such as Müller et al, 2019. Point (b) is less clearly an issue, as it would be impractical and confusing to empirically compare our method to all existing alternatives. Other reviewers commented we have many benchmarks. We looked at the focal-loss because it is applied quite widely and it was shown by Mukhoti et al. (2020) to outperform label smoothing (Müller et al, 2019) and other methods.
>
> Weaknesses (3). Short answer: (1) we have already considered diverse data sets, which other reviewers found satisfactory; (2) there is relatively little information to be gained from those extra image data sets. We expand upon this below.
> Recall: our goal is to train uncertainty-aware classifiers, and that there are two types of uncertainty: epistemic and aleatory. Epistemic uncertainty may be due to insufficient data, poor training, sub-optimal architecture, or a combination of those. This is what ML has traditionally focused on, as it tends to dominate traditional image classification tasks. Training flexible networks on large data sets is effective at removing epistemic uncertainty. If this were the only type of uncertainty, we could have gained more insight from CIFAR-100 or imagenet. However, we care more about aleatory uncertainty—the intrinsic randomness due to unmeasured variables, which cannot be eliminated so easily.
> Overconfidence is especially concerning in applications with aleatoric uncertainty, and our method is meant to address that. In Section 4.1 we work on synthetic data with aleatoric and epistemic uncertainty, the proportions of which are varied (Figures A7, A11, A15). In Section 4.3 we work on credit card data with aleatoric uncertainty. But the CIFAR-10 data in Section 4.2 mostly involve epistemic uncertainty. This is why we have introduced aleatoric uncertainty by corrupting some images (Figure 2). The same could be done with CIFAR-100 or (Tiny-)imagenet, but it is unclear whether there could be much insight to be gained from such exercise. In a revision, can further expand the discussion of epistemic and aleatory uncertainty.

---

> ### Comment · Area_Chair_DpLh · 2022-08-08
> **Please respond to author feedback**
>
> Thank you for reviewing this paper. Could you respond to the author feedback, or at least acknowledge that you've read the reply? Does the author reply address your concerns?
>
> Best, AC

---

### Author Response · Authors · 2022-08-01
**Response to reviews**

Dear Reviewers Xxp1, 3Bsi, 9Dxm, and m7U1,

Thank you for reading our paper carefully and providing many insightful comments. We have responded point-by-point below, and we have conducted new numerical experiments to accompany our answers to your comments.
To facilitate the second round of review, we have kept the new empirical results separate from the original submission. NewFigures 1--10 summarize the results of these new additional experiments, and they can be found in the supplementary file "response_figures.pdf".

If our paper is accepted, we will incorporate the new figures into the manuscript (or in the supplementary material, with pointers in the main text). We will also distill the main points of our discussion into the paper.

We would be very happy to continue the discussion if you have any remaining/follow-up comments or questions!

Thank you!

The anonymous authors

---

### Meta-Review · Area_Chair_DpLh · 2022-08-20

**Recommendation:** Accept
**Confidence:** Certain

**Metareview:**

Decision: Accept

This paper extends conformal prediction techniques to multi-class classification using deep neural networks and make the training of the neural network to be aware of the conformal inference processing. The main technical contribution is a differentiable objective to approximate a CDF-based test on the conformity score. The paper provides both theoretical analysis as well as empirical evaluation results of the proposed approach.

Reviewers found the paper to be well written and the approach to be well motivated and supported. There were a few technical concerns but many of them were addressed in author feedback.

Still the main technical downside is the expensiveness of the approach, and the experiments being relatively small scale regarding network size & dataset size. Also a lot of practical issues are not discussed, e.g., data augmentation, distribution shift, etc.

However, this is indeed an early work in the conformal prediction area that tries to make neural network training adaptive to conformal inference "post-processing", and I believe the work is going to the right direction and will have good impact in "conformal prediction + DL" area.

As a side note, I'd encourage the authors to add discussions on related work that proposes regularisers for better neural network calibration, e.g., MMCE https://proceedings.mlr.press/v80/kumar18a/kumar18a.pdf.

**Award:**

No

---

### Decision · Program_Chairs · 2022-09-14

Accept